# Is Deep Learning finally better than Decision Trees on Tabular Data?

## Abstract

Tabular data is a ubiquitous data modality due to its versatility and ease of use in many real-world applications. The predominant heuristics for handling classification tasks on tabular data rely on classical machine learning techniques, as the superiority of deep learning models has not yet been demonstrated. This raises the question of whether new deep learning paradigms can surpass classical approaches. Recent studies on tabular data offer a unique perspective on the limitations of neural networks in this domain and highlight the superiority of gradient boosted decision trees (GBDTs) in terms of scalability and robustness across various datasets. However, novel foundation models have not been thoroughly assessed regarding quality or fairly compared to existing methods for tabular classification. Our study categorizes ten state-of-the-art neural models based on their underlying learning paradigm, demonstrating specifically that meta-learned foundation models outperform GBDTs in small data regimes. Although dataset-specific neural networks generally outperform LLM-based tabular classifiers, they are surpassed by an AutoML library which exhibits the best performance but at the cost of higher computational demands.

## 1 Introduction

Tabular data has long been one of the most common and widely used data formats, with applications spanning various fields such as healthcare (Johnson et al., 2016; Ulmer et al., 2020), finance (A. & E., 2022), and manufacturing (Chen et al., 2023), among others. Despite being a ubiquitous data modality, tabular data has only been marginally impacted by the deep learning revolution (Van Breugel & Van Der Schaar, 2024). A significant portion of the research community in tabular data mining continues to advocate for traditional machine learning methods, such as gradient-boosting decision trees (GBDTs) (Friedman, 2001; Chen & Guestrin, 2016; Prokhorenkova et al., 2018). Recent empirical studies agree that GBDTs are still competitive for tabular data (McElfresh et al., 2023). Nevertheless, an increasing segment of the community highlights the benefits of deep learning methods (Kadra et al., 2021; Gorishniy et al., 2021; Arik & Pfister, 2021; Somepalli et al., 2021; Kadra et al., 2024), (Holzmüller et al., 2024).

The community remains divided on whether Deep Learning approaches are the undisputed state-of-the-art methods for tabular data (Shwartz-Ziv & Armon, 2022). To resolve this debate and determine the most effective methods for tabular data, multiple recent studies have focused on empirically comparing GBDT with Deep Learning methods (Grinsztajn et al., 2022; Borisov et al., 2022; McElfresh et al., 2023). These studies suggest that tree-based models outperform deep learning models on tabular data even after tuning neural networks.

However, these recent empirical surveys only include non-meta-learned neural networks (Grinsztajn et al., 2022; Borisov et al., 2022) and do not incorporate the recent stream of methods that leverage foundation models and LLMs for tabular data (Zhu et al., 2023; Hollmann et al., 2023; Yan et al., 2024; Kim et al., 2024). Furthermore, the empirical setup of the recent empirical benchmarks is sub-optimal because no thorough hyperparameter optimization (HPO) techniques were applied to carefully tune the hyperparameters of neural networks.

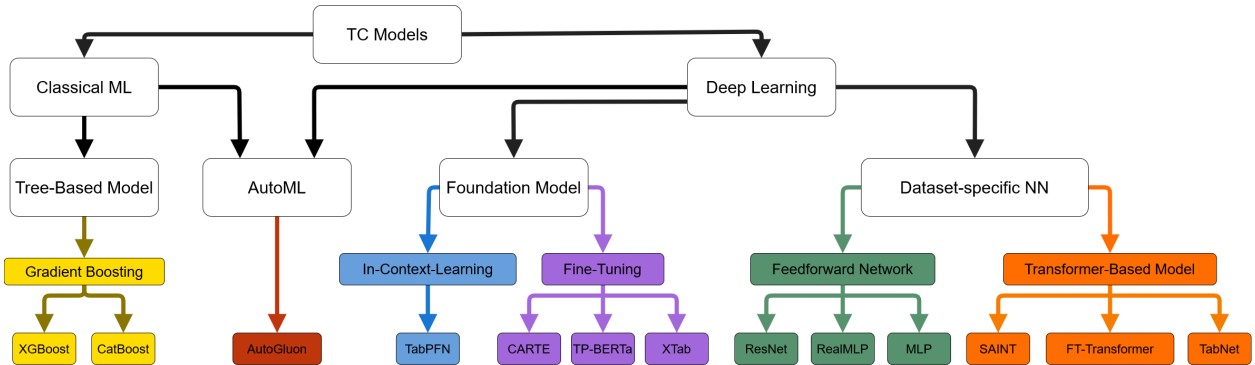

Figure 1: Taxonomy tree of algorithms applied to tabular classification (TC) models

In this empirical survey paper, we address a simple question: "Is Deep Learning now state-of-the-art on tabular data, compared to GBDTs?" Providing an unbiased and empirically justified answer to this question has a significant impact on the large community of practitioners. Therefore, we designed a large-scale experimental protocol using 68 diverse OpenML datasets and 11 recent baselines, including foundation models for tabular data. In our protocol, we use 10-fold cross-validation experiments for all the datasets and fairly tune the hyperparameters of all the baselines with an equally large HPO budget.

Moreover, to fully unlock a model's potential, after HPO we refit all models on the joined training and validation set. Hence, our study provides a fair investigation of post-hyperparameter optimization. We argue that this is a crucial oversight because training on the combined dataset can provide additional information to a model, potentially improving generalization, especially, in small data regimes. Hence, in our large-scale study, the aim is to compare classical gradient-boosted decision trees to most modern deep learning model families. We classify models according to their underlying paradigm and provide a taxonomy tree in Figure 1 including tree-based models stemming from classical machine learning, in-context learning and fine-tuned models as sub-paradigms of foundation models, and dataset-specific architectures for tabular data, which we subclassify into feedforward approaches and transformer-based models. Additionally, we include the performance of an AutoML library in our study.

In summary, we employ two well-established models from traditional machine learning, namely XGBoost (Chen & Guestrin, 2016) and CatBoost (Prokhorenkova et al., 2018), and compare them with 8 deep learning architectures (Somepalli et al., 2021; Gorishniy et al., 2021; Arik & Pfister, 2021; Hollmann et al., 2023; Kim et al., 2024; Yan et al., 2024) categorized in model families w.r.t. their learning paradigm. For a fair comparison, we evaluate all models on 68 benchmark datasets from the OpenML datasets (Bischl et al., 2021). The key findings of this study are:

- Meta-learned foundation models (Hollmann et al., 2023) outperform GBDTs in **small** datasets.

- In **mid-size and large** datasets, XGBoost and CatBoost (Chen & Guestrin, 2016; Prokhorenkova et al., 2018) continue to be competitive against Deep Learning methods in terms of performance per training cost.

- AutoML libraries, particularly AutoGluon (Erickson et al., 2020), deliver the highest **overall** performance compared to other methods but introduce a significant computational overhead.

- Dataset-specific neural networks (e.g., FT-Transformer, SAINT, ResNet) generally outperform LLM-based tabular classifiers.

## 2 Related Work

Given the prevalence of tabular data in numerous areas, including healthcare, finance, psychology, and anomaly detection, as highlighted in various studies (Chandola et al., 2009; Johnson et al., 2016; Guo et al.,

Table 1: Comparison with prior empirical survey works. In our study, we include 6 model families: Gradient Boosted Decision Trees (GBDT), AutoML, In-Context Learning (ICL), Fine-tuning (FT), Feed forward neural networks (FNN), and Transformer-based Models (Transformer).

| | Protocol | | | Model families | | | | | | |
|---|---|---|---|---|---|---|---|---|---|---|
| Study | Refitting | Model-based HPO | # Datasets | GBDT | AutoML | ICL | FT | FNN | Transformer | # Baselines |
| Ye et al. (2025) | ✗ | ✓ | 300 | ✓ | ✗ | ✓ | ✗ | ✓ | ✓ | 31 |
| Rubachev et al. (2024) | ✗ | ✓ | 8 | ✓ | ✗ | ✗ | ✗ | ✓ | ✓ | 14 |
| McElfresh et al. (2023) | ✗ | ✗ | 176 | ✓ | ✗ | ✓ | ✗ | ✓ | ✓ | 19 |
| Borisov et al. (2022) | ✗ | ✓ | 5 | ✓ | ✗ | ✗ | ✗ | ✓ | ✓ | 20 |
| Grinsztajn et al. (2022) | ✗ | ✗ | 45 | ✓ | ✗ | ✗ | ✗ | ✓ | ✓ | 7 |
| Shwartz-Ziv & Armon (2022) | ✗ | ✓ | 11 | ✓ | ✗ | ✗ | ✗ | ✗ | ✓ | 5 |
| Gorishniy et al. (2021) | ✗ | ✓ | 11 | ✓ | ✗ | ✗ | ✗ | ✓ | ✓ | 11 |
| **Ours** | ✓ | ✓ | 68 | ✓ | ✓ | ✓ | ✓ | ✓ | ✓ | 13 |

2017; Ulmer et al., 2020; Urban & Gates, 2021; A. & E., 2022; Van Breugel & Van Der Schaar, 2024), there has been significant research dedicated to developing algorithms that effectively address the challenges inherent in this domain. We summarize all algorithms evaluated in our study in a taxonomy tree shown in Figure 1.

**Classical Machine Learning.** Gradient Boosted Decision Trees (GBDTs) (Friedman, 2001), including popular implementations like XGBoost (Chen & Guestrin, 2016), LightGBM (Ke et al., 2017), and Cat-Boost (Prokhorenkova et al., 2018), are widely favored by practitioners for their robust performance on tabular datasets, and their short training times.

**Deep Learning.** In terms of neural networks, prior work shows that meticulously searching for the optimal combination of regularization techniques in simple multilayer perceptrons (MLPs) called *Regularization Cocktails* (Kadra et al., 2021) can yield impressive results. Two recent papers (Kadra et al., 2021; Gorishniy et al., 2021) propose adaptations of the ResNet architecture for tabular data, demonstrating the potential of deep learning approaches in handling tabular data. This version of ResNet, originally conceived for image processing (He et al., 2016), has been effectively repurposed for tabular datasets in their research. We demonstrate that with thorough hyperparameter tuning, a ResNet model tailored for tabular data rivals the performance of transformer-based architectures. Furthermore, recent research underscores that numerical embeddings (Gorishniy et al., 2022) for tabular data are underexplored. Incorporating these embeddings into neural network architectures, including MLPs and transformer-based models, can substantially enhance performance. Additionally, novel approaches such as RealMLP (Holzmüller et al., 2024) introduce various enhancements to the standard MLP architecture. These include using robust scaling at the preprocessing stage and experimenting with alternative numerical embedding strategies. In doing so, the authors show that RealMLP surpasses other neural network models but also remains competitive with GBDT methods.

Reflecting their success in various domains, transformers have also garnered attention in the tabular data domain. TabNet (Arik & Pfister, 2021), an innovative model in this area, employs attention mechanisms sequentially to prioritize the most significant features. SAINT (Somepalli et al., 2021), draws inspiration from the seminal transformer architecture (Vaswani et al., 2017). It addresses data challenges by applying attention both to rows and columns. They also offer a self-supervised pretraining phase, particularly beneficial when labels are scarce. The FT-Transformer (Gorishniy et al., 2021) stands out with its two-component structure: the Feature Tokenizer and the Transformer. The Feature Tokenizer is responsible for converting numerical and categorical features into embeddings. These embeddings are then fed into the Transformer, forming the basis for subsequent processing.

Recently, a new avenue of research has emerged, focusing on the use of foundation models for tabular data. XTab (Zhu et al., 2023) utilizes shared Transformer blocks, similar to those in FT-Transformer (Gorishniy et al., 2021), followed by fine-tuning dataset-specific encoders. Another notable work, TabPFN (Hollmann et al., 2023), employs in-context learning (ICL), by leveraging sequences of labeled examples provided in the input for predictions, thereby eliminating the need for additional parameter updates after training. TP-BERTa (Yan et al., 2024), a pre-trained language model for tabular data prediction, uses relative magnitude tokenization to convert scalar numerical features into discrete tokens. TP-BERTa also employs intra-feature

attention to integrate feature values with feature names. The last layer of the model is then fine-tuned on a per-dataset basis. In contrast, CARTE (Kim et al., 2024) utilizes a graph representation of tabular data and a neural network capable of capturing the context within a table. The model is then fine-tuned on a per-dataset basis.

**Empirical Studies.** Significant research has delved into understanding the contexts where neural networks (NNs) excel, and where they fall short (Shwartz-Ziv & Armon, 2022; Borisov et al., 2022; Grinsztajn et al., 2022). The recent study by (McElfresh et al., 2023) is highly related to ours in terms of research focus. However, the authors used only random search for tuning the hyperparameters of neural networks, whereas we employ Tree-structured Parzen Estimator (TPE) (Bergstra et al., 2011) as employed by (Gorishniy et al., 2021), which provides a more guided and efficient search strategy. Additionally, (McElfresh et al., 2023) study was limited to evaluating a maximum of 30 hyperparameter configurations, in contrast to our more extensive exploration of up to 100 configurations. Furthermore, despite using the validation set for hyperparameter optimization (HPO), they do not retrain the model on the combined training and validation data using the best-found configuration before evaluating the model on the test set. Our paper delineates from prior studies by applying a methodologically correct experimental protocol involving thorough HPO for neural networks. Moreover, Table 1 summarizes the model families evaluated in related empirical studies and highlights the differences in the evaluation protocol. To the best of our knowledge, we are the first to provide a thorough assessment of foundation models leveraging fine-tuning (FT) as a new player in the field of tabular classification (TC), and AutoML where - at the time of writing - both model families have been overlooked in most recent studies Ye et al. (2025); Rubachev et al. (2024), and compare them to other learning paradigms.

## 3 Experimental Protocol

In our study, we focus on binary and multi-class classification problems on tabular data. The general learning task is described in Section 3.1. A detailed description of our evaluation protocol is provided in Section 3.2.

### 3.1 Learning with Tabular Data

A tabular dataset contains $N$ samples with $d$ features defining a $N \times d$ table. A sample $x_i \in \mathbb{R}^d$ is defined by its $d$ feature values. The features can be continuous numerical values or categorical where for the latter a common heuristic is to transform the values in numerical space. Given labels $y_i \in \mathcal{Y}$ being associated with the instances (rows) in the table, the task to solve is a binary or multi-class classification problem or a regression task iff $y_i \in \mathbb{R}$. In our study, we focus on the former. Hence, given a tabular dataset $\mathcal{D} = \{(x_i, y_i)\}_{i=1}^N$, the task is to learn a prediction model $f(\cdot, \cdot)$ to minimize a classification loss function $\ell(\cdot, \cdot)$:

$$\arg\min_{\theta} \sum_{(x_i, y_i) \in \mathcal{D}} \ell(y_i, f(x_i; \theta, \lambda)), \tag{1}$$

where we use $f(x_i; \theta, \lambda)$ for denoting the predicted label by a trained model parameterized by the model weights $\theta$ and hyperparameter configuration $\lambda$.

### 3.2 Experimental Setup

**Datasets.** In our study, we assess all the methods using OpenMLCC18 (Bischl et al., 2021), a well-established tabular benchmark in the community, which comprises 72 diverse datasets[1]. The datasets contain 5 to 3073 features and 500 to 100,000 instances, covering various binary and multi-class problems. The benchmark excludes artificial datasets, subsets or binarizations of larger datasets, and any dataset solvable by a single feature or a simple decision tree. For the full list of datasets used in our study, please refer to Appendix C.

**Evaluation Protocol.** Our evaluation employs a nested cross-validation approach. Initially, we partition the data into 10 folds. Nine of these folds are then used for hyperparameter tuning. Each hyperparameter

---

[1]Due to memory issues encountered with several methods, we exclude four datasets from our analysis.

configuration is evaluated using 9-fold cross-validation. The results from the cross-validation are used to estimate the performance of the model under a specific hyperparameter configuration.

For hyperparameter optimization, we utilize Optuna (Akiba et al., 2019), a well-known HPO library with the Tree-structured Parzen Estimator (TPE) (Bergstra et al., 2011) algorithm, the default Optuna HPO method. The optimization is constrained by a budget of either 100 trials or a maximum duration of 23 hours. Upon determining the optimal hyperparameters using Optuna, we train the model on the combined training and validation splits. We provide a detailed description of our protocol in Algorithm 1. It shows the nested-cross validation with the outer folds (lines 1-16) and inner folds (lines 5-9). In each trial (lines 3-12), the mean performance across inner folds are calculated in line 10 which is used as the objective value for Optuna in line 11. After the maximal number of trials $T$ is reached or the time budget is exceeded, we select the best hyperparameter setting in line 13. In comparison to previous studies, we now refit the model in line 14 which yields for each outer fold a performance measurement in line 15. We refer to Appendix F for an ablations study on the refitting procedure. The final performance is calculated across all outer folds in line 17. To enhance efficiency, we execute every outer fold in parallel across all datasets.

All experiments are run on NVIDIA RTX2080Ti GPUs with a memory of 11 GB. Our evaluation protocol dictates that for every algorithm, up to **68K** different models will be evaluated, leading to a total of approximately **800K** individual evaluations. As our study encompasses thirteen distinct methods, this methodology culminates in a substantial total of over **8M** **evaluations**, involving more than **900K** unique models.

**Metrics.** Lastly, we report the model's performance as the average Area Under the Receiver Operating Characteristic (ROC-AUC) across 10 outer test folds. Given the prevalence of imbalanced datasets in the OpenMLCC18 benchmark, we employ ROC-AUC as our primary metric. This measure offers a more reliable assessment of model performance across varied class distributions, as it is less influenced by the imbalance in a dataset.

In our study, we adhered to the official hyperparameter search spaces from the respective papers for tuning every method. For a detailed description of the hyperparameter search spaces of all other methods included in our analysis, we refer the reader to Appendix A.

**Code:** For reproducibility, our code is available at: https://anonymous.4open.science/r/TabularStudy-0EE2.

## 4 Baselines

In our experiments, we compare a range of methods categorized into three distinct groups: Classical Machine Learning Classifiers (§ 4.1), Deep Learning Methods (§ 4.2), and AutoML frameworks (§ 4.3) as shown in Figure 1.

### 4.1 Classical Machine Learning Classifiers

**Gradient Boosted Decision Trees.** First, we consider *XGBoost* (Chen & Guestrin, 2016), a well-established GBDT library that uses asymmetric trees. The library does not natively handle categorical features, which is why we apply one-hot encoding. Moreover, we consider *CatBoost*, a well-known library for GBDT that employs oblivious trees as weak learners and natively handles categorical features with various strategies.

### 4.2 Deep Learning Methods

**Dataset-Specific Neural Networks.** Recent works have shown that MLPs featuring residual connections outperform plain MLPs and make for very strong competitors to state-of-the-art architectures (Kadra et al., 2021; Gorishniy et al., 2021). As such, in our study, we include the *ResNet* implementation provided in Gorishniy et al. (2021). Furthermore, research indicates that incorporating numerical embeddings (Gorishniy et al., 2022), such as PLR, into neural networks significantly improves performance, motivating us to include an MLP with PLR embeddings in our study Throughout this paper, any mention of MLP refers specifically

---

**Algorithm 1:** Nested Cross-Validation for Hyperparameter Optimization

---

    **Input** : Dataset $D$, Number of outer folds $K = 10$, Number of inner folds $J = 9$, Number of
                hyperparameter optimization trials $T = 100$, Search space $\Lambda$

    **Output:** Overall performance $\bar{P}_{\text{outer}}$

**1**   **for** $k \leftarrow 1$ **to** $K$ **do**

**2**      Split $D$ into training set $D_{\text{train}}^k$ and test set $D_{\text{test}}^k$;

**3**      **for** $t \leftarrow 1$ **to** $T$ **do**

**4**          Sample hyperparameter configuration $\theta_t$ from the search space $\Lambda$;

**5**          **for** $j \leftarrow 1$ **to** $J$ **do**

**6**              Split $D_{\text{train}}^k$ into inner training set $D_{\text{train}}^{k,j}$ and validation set $D_{\text{val}}^{k,j}$;

**7**              Train model $M(\lambda_t)$ on $D_{\text{train}}^{k,j}$;

**8**              Evaluate $M(\lambda_t)$ on $D_{\text{val}}^{k,j}$ to get performance $P^{k,j}(\lambda_t)$;

**9**          **end**

**10**          Compute mean performance $\bar{P}^k(\lambda_t) = \frac{1}{J} \sum_{j=1}^{J} P^{k,j}(\lambda_t)$;

**11**          Use $\bar{P}^k(\lambda_t)$ as the objective value for $\lambda_t$;

**12**      **end**

**13**      Select the best hyperparameter configuration $\lambda_k^*$ ;

**14**      Train final model $M(\lambda_k^*)$ on $D_{\text{train}}^k$;

**15**      Evaluate $M(\lambda_k^*)$ on $D_{\text{test}}^k$ to get outer performance $P_{\text{outer}}^k$;

**16**   **end**

**17** Compute overall performance $\bar{P}_{\text{outer}} = \frac{1}{K} \sum_{k=1}^{K} P_{\text{outer}}^k$;

**18** **return** $\bar{P}_{\text{outer}}$;

---

to an MLP with PLR embeddings. Additionally, RealMLP (Holzmüller et al., 2024) not only leverages numerical embeddings but also employs techniques like robust scaling, smooth clipping, and a diagonal weight layer, enabling it to surpass other neural network models. Consequently, we include RealMLP in our analysis as well.

Additionally, we consider several transformer-based architectures designed specifically for tabular data. *Tab-Net* (Arik & Pfister, 2021) employs sequential attention to selectively utilize the most pertinent features at each decision step. For the implementation of TabNet, we use a well-maintained public implementation[2]. *SAINT* (Somepalli et al., 2021) introduces a hybrid deep learning approach tailored for tabular data challenges. SAINT applies attention mechanisms across both rows and columns and integrates an advanced embedding technique. Lastly, *FT-Transformer* Gorishniy et al. (2021) is an adaptation of the Transformer architecture for tabular data. It transforms categorical and numerical features into embeddings, which are then processed through a series of Transformer layers.

**Foundation Models for Tabular Classification.** These models can be further divided into two categories based on their learning strategies: *In-Context Learning* and *Fine-Tuning*. The former eliminates the need for per-dataset finetuning, whereas the latter requires models to undergo finetuning specific to each dataset.

For **in-context learning**, we consider *TabPFN*, a meta-learned transformer architecture.

Among **fine-tuned models**, *XTab* proposes a cross-table pretraining approach that can work across multiple tables with different column types and structures. The approach utilizes independent featurizers for individual columns and federated learning to train a shared transformer component, allowing better generalization. Next, *TP-BERTa* is a variant of the BERT language model being specifically adapted for tabular prediction. It introduces a relative magnitude tokenization to transform continuous numerical values into discrete high-dimensional tokens. Its learning procedure relies on an intra-feature attention module that learns relationships between feature values and their corresponding feature names, effectively bridging the gap between numerical data and language-based feature representation. Lastly, *CARTE* utilizes a graph representation

---

[2]https://github.com/dreamquark-ai/tabnet

of tabular data to process tables with differing structures. It applies an open vocabulary setting and contextualizes the relationship between table entries and their corresponding columns by a graph-attentional network. CARTE is pre-trained on a large knowledge base enhancing generalization.

Since all the fine-tuned models were pretrained on real-world datasets, we ensure that no datasets overlapping with the OpenMLCC18 benchmark are included in the evaluation. For all baselines, we use their official implementations. We refer the readers to Appendix A for more details.

### 4.3 AutoML Frameworks

Due to the large number of AutoML frameworks available in the community (Feurer et al., 2015; Erickson et al., 2020; LeDell & Poirier, 2020; Feurer et al., 2022), it was infeasible to include all of them in our experimental study. Therefore, we selected AutoGluon (Erickson et al., 2020), which is regarded as one of the best AutoML frameworks according to the AutoML Benchmark study (Gijsbers et al., 2024). Unlike other AutoML systems, AutoGluon does not recommend performing hyperparameter optimization, instead, it relies on stacking and ensembling. In our study, we include two versions of AutoGluon: one where we perform hyperparameter optimization for all models, and a version where we follow the original author's recommendations by setting `presets="best_quality"`.

## 5 Experiments and Results

In this study, we aim to address several key research questions related to the performance of machine learning techniques and various deep learning model families on tabular data classification:

- **R1:** Do DL models outperform gradient boosting methods in tabular data classification? (§ 5.1)
  → GBDTs show robust performance while TabPFN is competitive on small datasets.
- **R2:** Do meta-learned NNs outperform data-specific NNs in tabular data classification? (§ 5.1)
  → TabPFN is superior on small datasets. Dataset-specific sota models outperform foundation models.
- **R3:** Do in-context models or fine-tuned models perform better? (§ 5.2)
  → TabPFN wins on small datasets. XTab as a fine-tuned model yields the most robust performance.
- **R4:** Do DL models outperform AutoML libraries? (§ 5.3)
  → AutoML is superior to DL models on a broad range of datasets.
- **R5:** What is the influence of hyperparameter optimization on the output quality? (§ 5.4)
  → HPO shows significant improvements besides for models reaching computational limits.
- **R6:** How do dataset characteristics relate to the performance of different model families? (§ 5.5)
  → No significant relations of meta features to performance across different model families.
- **R7:** What is the cost vs. efficiency relation of various model families? (§ 5.6)
  → GBDTs show best relations w.r.t. inference time while AutoML is competitive w.r.t. total time.

### 5.1 (R1+R2) Quality metrics

To address our first two research questions, we compare the performance of different families of methods by ranking each method on every dataset and analyzing the distribution of these ranks. Figure 2 presents a comparison between Deep Learning methods and classical ML methods. The results indicate that Classical ML methods - namely CatBoost and XGBoost - demonstrate robust and consistent performance across datasets, with CatBoost achieving a median rank of 2 and XGBoost a median rank of 2.5. Among the Deep Learning methods, TabPFN exhibits the best performance with a median rank of 2.

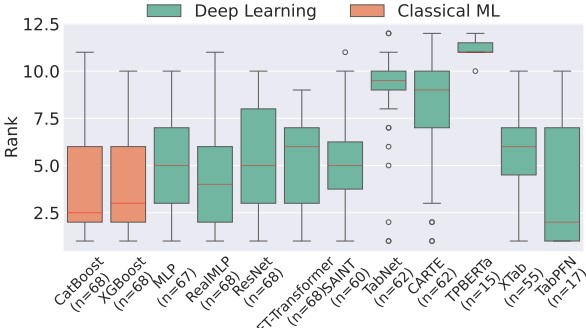 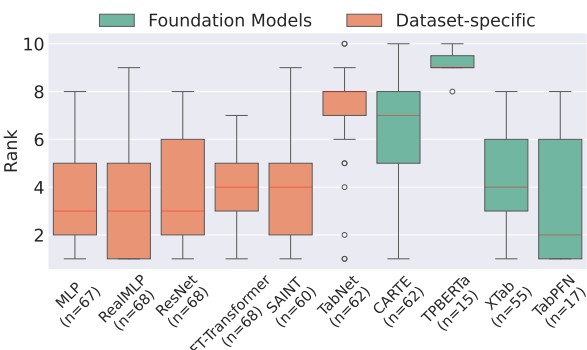

Figure 2: Distribution of ranks for the Deep Learning (10 methods) and Classical ML (2 methods) classifier families. The boxplot illustrates the rank spread, with medians represented by red lines and whiskers showing the range.

Figure 3: Distribution of ranks for the Foundation Models (4 methods) and Dataset-Specific (4 methods) classifier families. The boxplot illustrates the rank spread, with medians represented by red lines and whiskers showing the range.

To address the second research question R2, we analyzed the distribution of ranks between the two subfamilies within the Deep Learning category: Foundation Models and Dataset-Specific Neural Networks. Figure 3 illustrates that, overall, dataset-specific neural networks outperform foundation models, with the notable exception of TabPFN, which achieved a median rank of 2 across its 17 evaluated datasets. Within the dataset-specific family, RealMLP demonstrated the best performance, attaining a median rank of 3 across all 68 datasets. This is followed by MLP with PLR embeddings and ResNet, both with a median rank of 3, however, MLP exhibited a narrower interquartile range, indicating a more consistent performance. Among the foundation models, after TabPFN, XTab shows the next best performance with a median rank of 3, followed by CARTE, and finally, TP-BERTa, which displays the lowest performance within this group.

To compare foundation models with dataset-specific NNs in the small data regime, we utilized the `autorank` package Herbold (2020), performing a Friedman test followed by a Nemenyi post-hoc test at a significance level of 0.05. This statistical analysis allows for generating a critical difference (CD) diagram shown in the left plot of Figure 4.

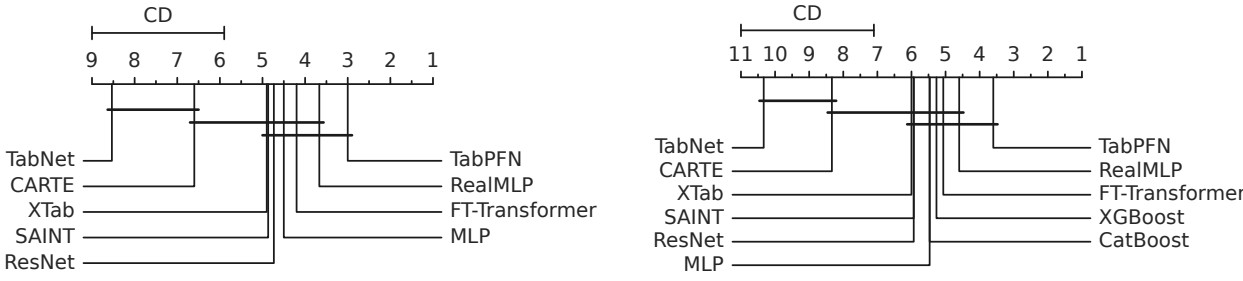

Figure 4: Statistical analysis in the small data domain (number of instances $\leq 1000$). **Left:** Critical Difference (CD) Diagram of dataset-specific neural networks against foundation models. **Right:** Critical Difference (CD) Diagram of Deep Learning models against GBDT models

Due to the limited number of datasets shared among the methods, TP-BERTa was excluded from this comparison. The black bars connecting methods indicate that there is no statistically significant difference in performance. While TabPFN outperforms RealMLP, FT-Transformer, MLP, ResNet, SAINT and XTab these results are not statistically significant. The CD diagram illustrates that in the small data domain, i.e., number of instances $\leq 1,000$, TabPFN outperforms other methods, demonstrating superior performance. We also compare the performance of deep learning models with classical machine learning models in the

small data domain. Right plot of Figure 4 presents a critical difference diagram, which indicates that TabPFN achieves the highest rank overall. A comprehensive presentation of the raw results for all methods, both after hyperparameter optimization (HPO) and with default hyperparameter configurations, is provided in Appendix B.

## 5.2 (R3) In-context learning vs. Fine-tuning

To further investigate the family of foundation models whether fine-tuning or in-context learning models yield better performance, we conducted an analysis similar to our previous research questions. We employ boxplots to display the distribution of ranks and use critical difference (CD) diagrams to evaluate the statistical significance of the results.

Figure 5 illustrates that TabPFN, categorized under in-context learning methods, achieved a median rank of 1 with no interquartile range, indicating highly consistent performance across its 17 evaluated datasets. Among the fine-tuning methods, XTab showed the best performance with a median rank of 1 but exhibited a larger interquartile range, followed by CARTE and TP-BERTa.

For a statistical comparison, we present a CD diagram in Figure 6, from which TP-BERTa is excluded due to the limited number of common datasets among the methods. The CD diagram reveals that TabPFN outperforms both XTab and CARTE, highlighting its superiority within the Foundation Models category.

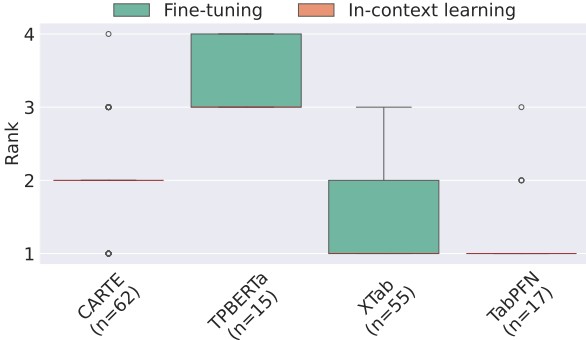

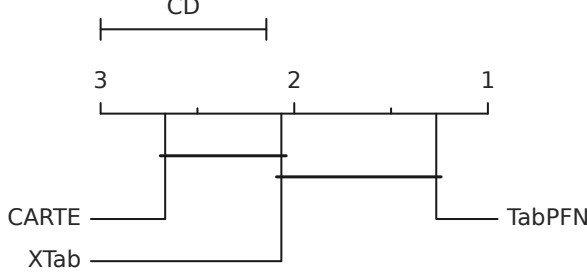

Figure 5: Distribution of ranks for the Fine-tuning (3 models) and In-context learning (1 model) classifier families. The boxplots illustrate the rank spread, with medians represented by red lines and whiskers showing the range.

Figure 6: Comparative analysis of Fine-tuning models against In-context learning models in the small data domain (number of instances $\leq$ 1000). The horizontal bar indicates the absence of statistically significant differences.

## 5.3 (R4) Deep Learning models vs. AutoML Libraries

In our study, we include AutoGluon Erickson et al. (2020), a prominent AutoML library, to compare against the Deep Learning methods. We consider two versions of AutoGluon: one where we perform hyperparameter optimization (HPO) and the officially recommended version configured with `presets=best_quality`. We compare all methods within the Deep Learning family to these versions of AutoGluon.

Figure 7 presents boxplots showing the distribution of ranks for all Deep Learning methods compared to AutoGluon. The left-hand side illustrates the comparison with the HPO version of AutoGluon. Except for TabPFN, which achieves a median rank of 2, while RealMLP follows closely with a strong median rank of 3, all other methods are outperformed by AutoGluon, which has a median rank of 3, with its interquartile range extending down to rank 1 across its 68 evaluated datasets. The right-hand side displays the recommended version of AutoGluon, which exhibits even better performance with a median rank of 1 and a very narrow interquartile range, indicating robust performance.

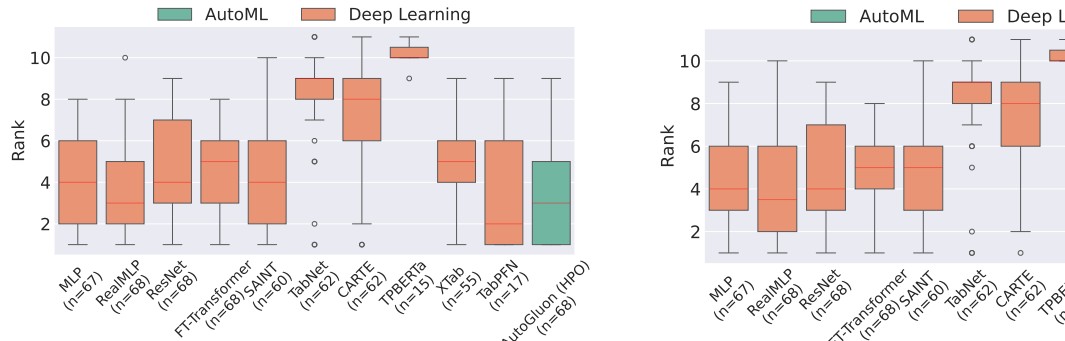

Figure 7: Distribution of ranks for the Deep Learning Models (10 methods) and AutoML (1 method) classifier families. **Left:** AutoGluon with hyperparameter optimization (HPO). **Right:** AutoGluon in its recommended configuration. The boxplot illustrates the rank spread, with medians represented by red lines and whiskers showing the range.

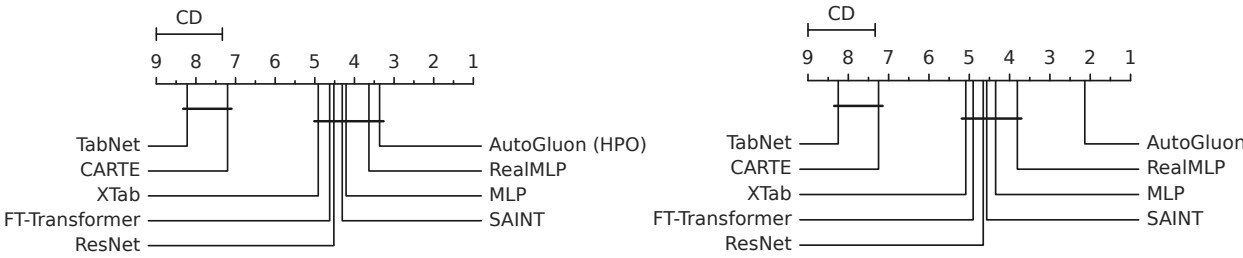

Figure 8: Comparative analysis of Deep learning models against AutoGluon. **Left:** AutoGluon with hyperparameter optimization (HPO). **Right:** AutoGluon in its recommended configuration.

Consistent with our previous analyses, we also present critical difference (CD) diagrams for this comparison. Figure 8 shows, on the left side, the CD diagram of AutoGluon with HPO and, on the right side, AutoGluon with its recommended configuration, both compared against the Deep Learning methods across all datasets. The left CD diagram indicates that while AutoGluon attains the highest rank, the differences are not statistically significant when compared to SAINT, FT-Transformer, ResNet, and XTab; however, they are statistically significant when compared to CARTE and TabNet. The right CD diagram tells a different story: AutoGluon with its recommended settings is superior to every method in the comparison with a statistically significant result.

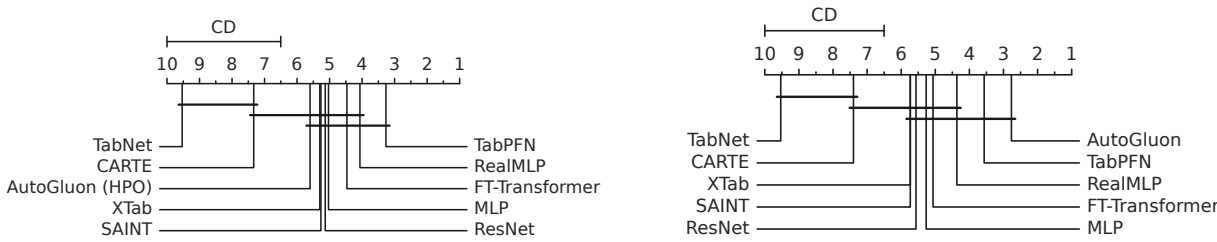

Figure 9: Comparative analysis of Deep learning models against AutoGluon in the small data domain (number of instances ≤ 1000). **Left:** AutoGluon with hyperparameter optimization (HPO). **Right:** AutoGluon in its recommended configuration.

Furthermore, we conduct the same analysis in the small data domain, where the number of instances is ≤ 1000, to include TabPFN in the analysis. Figure 9 presents these results. In the left diagram, TabPFN outperforms all other methods, including the HPO version of AutoGluon. However, in the right diagram,

AutoGluon with its recommended settings outperforms every method, including TabPFN, nevertheless, the difference is not statistically significant.

### 5.4 (R5) Analysis of HPO

In our analysis of hyperparameter optimization (HPO) versus default configurations across various machine learning methods, we observed that HPO generally led to improved performance. The analysis is depicted in Figure 10. This improvement is reflected in the average rank reductions for most methods when HPO was applied. For example, XGBoost's average rank improved significantly from 1.94 in its default configuration to 1.06 with HPO, and XTab showed a similar enhancement, moving from a rank of 1.96 down to 1.04. These findings are visually represented in the accompanying plot, which illustrates the performance gains achieved through HPO. An exception to the general trend was observed with TP-BERTa, where the default configuration slightly outperformed the HPO version (average ranks of 1.47 and 1.53, respectively). This anomaly can be attributed to the computational demands of TP-BERTa. Due to its large model size, TP-BERTa was unable to complete the full 100 hyperparameter tuning trials within the allotted 23-hour time frame, often finishing only a few trials. Consequently, the HPO process may have converged to a suboptimal configuration that did not surpass the performance of the default settings.

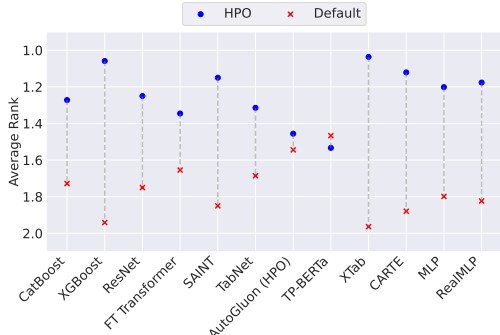

Figure 10: Comparison of average rank performance between hyperparameter-optimized (HPO) models and default models. The blue dots represent the performance of the HPO models, while the red crosses denote the default models. Lower ranks indicate better performance.

Figure 11 illustrates the importance of the individual hyperparameters tuned for every method. We calculate hyperparameter importance using the fANOVA (Hutter et al., 2014) implementation in Optuna (Akiba et al., 2019). According to our analysis, the most important hyperparameter for CatBoost is the learning rate, while for XGBoost, it is the subsample ratio of the training instances. For XTab, the learning rate is also the most important hyperparameter, closely followed by the `light_finetune` hyperparameter, which is a categorical parameter taking values `True` or `False`. When `light_finetune` is `True`, we fine-tune XTab for only 3 epochs; when it is `False`, we use the same range of epochs as for the other methods (10 to 500). Similarly, for the MLP with PLR embeddings, the learning rate proves to be the most influential hyperparameter, whereas for RealMLP, the number of units in the hidden layers. For the remaining dataset-specific neural networks in the deep learning family, as well as for CARTE, the number of training epochs is the most important hyperparameter, indicating that training duration plays a critical role in their performance.

### 5.5 (R6) Influence of meta-feature characteristics on the predictive performance

Following the methodology of McElfresh et al. (2023), we employed the PyMFE library (Alcobaça et al., 2020) to extract meta-features from the datasets used in our study. Specifically, we extracted General, Statistical, and Information-theoretical meta-features.

Figure 12 displays the mean correlation coefficients of the most significant meta-features concerning the performance of all methods, averaged across datasets. To produce this plot, we first calculate the correlation coefficients between each method's performance and each meta-feature for all datasets. For each method, we then selected the top $k$ meta-features with the highest absolute value of the correlation coefficients across all datasets, identifying them as the most important ones for that specific method. We compiled a list of significant meta-features by taking the union of these top meta-features across all methods. For each meta-feature in this combined list, we computed the mean of its correlation coefficients across datasets for all methods. Figure 12 illustrates that TabPFN and TPBERTa significantly deviate from the overall pattern observed in the other methods, exhibiting negative correlations for the meta-features `mad.mean`, `median.sd`, `t_mean.mean`, and `t_mean.sd`. To determine whether this deviation is due to the inherent properties of

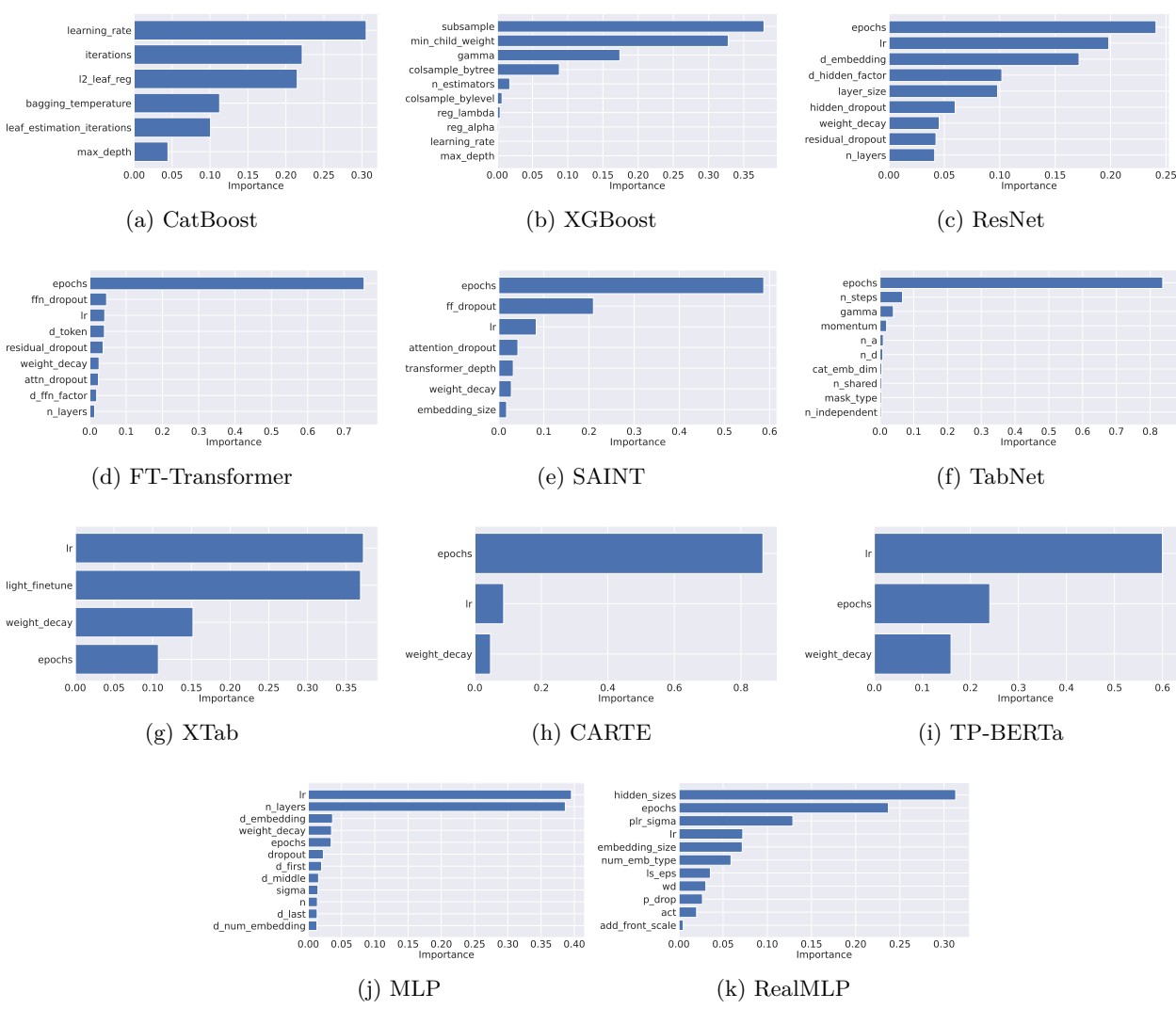

Figure 11: Hyperparameter importance for various methods

these methods or is a consequence of the limited number of datasets they were evaluated on, we repeated the analysis for all methods using only the datasets on which TabPFN and TPBERTa were run.

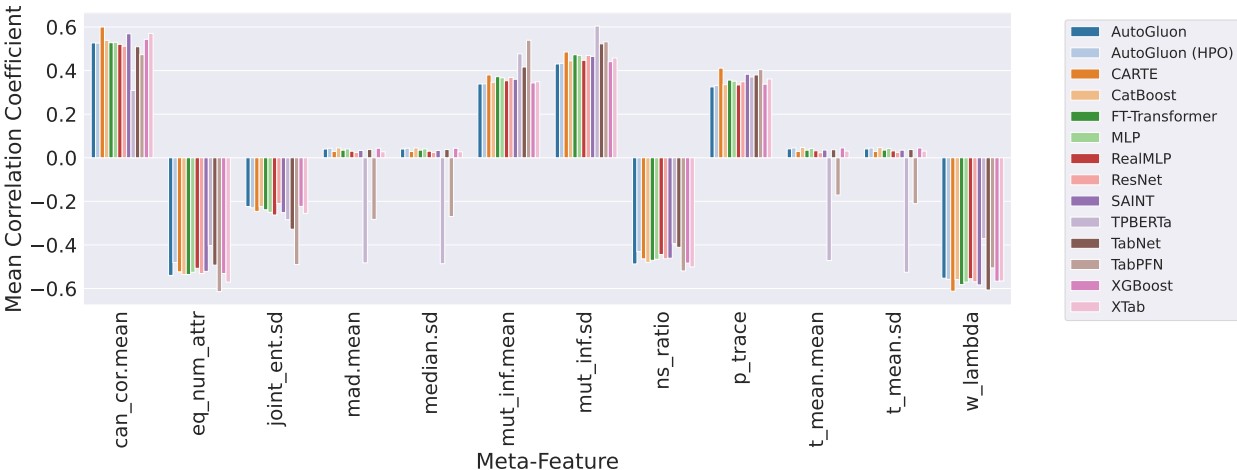

Figure 12: Mean correlation coefficient of most important meta-features with performance across all methods

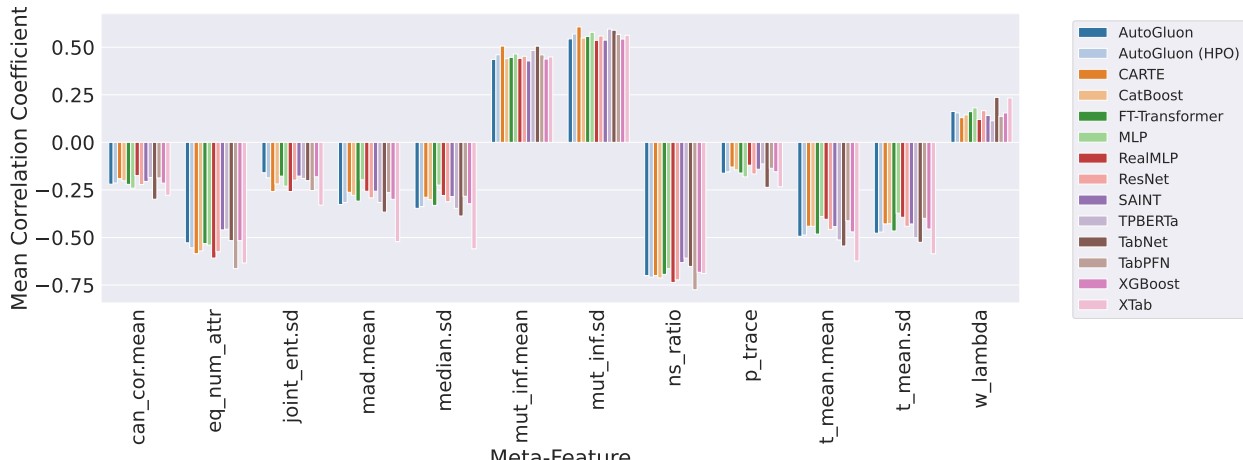

Figure 13: Mean correlation coefficient of most important meta-features with performance across all methods on datasets with results for TabPFN and TPBERTa

Figure 13 demonstrates that when the analysis is confined to only the intersection of datasets on which TabPFN and TPBERTa were evaluated, the previously observed deviation disappears. This suggests that the initial divergence was likely due to the limited number of datasets rather than the inherent properties of these methods. Therefore, it appears that all methods, regardless of their method families, are similarly influenced by the meta-features in terms of their predictive performance. In general, the strongest correlation coefficients are observed for three meta-features: `eq_num_attr`, `w_lambda`, and `can_cor.mean`.

The `eq_num_attr` meta-feature, which measures the number of attributes equivalent in information content for the predictive task, exhibits a strong negative correlation with performance across most methods. This suggests that methods generally perform worse on datasets with high feature redundancy, likely due to challenges in handling overlapping information or overfitting. Similarly, the `w_lambda` meta-feature, which computes Wilk's Lambda to quantify the separability of classes in the feature space, also shows a consistently negative correlation. This indicates that methods struggle on datasets with poor class separability, where the features do not adequately distinguish between the target classes. Conversely, the `can_cor.mean` meta-feature, representing the mean canonical correlation between features and the target, shows a positive correlation with performance. This implies that methods perform better on datasets where the features are

strongly predictive of the target variable, highlighting their reliance on well-aligned feature-target relationships.

Generally, the findings align with the common intuition of the performance of ML methods under sub-optimal class separation and further validate the empirical protocol of our study. For detailed explanations of the meta-feature abbreviations used in the plots, please refer to the official PyMFE documentation[3].

### 5.6 (R7) Cost vs. efficiency relation

To address the final research question, to see what is the cost vs. efficiency relation of various model families, we plot the intra-search space normalized Average Distance to the Maximum (ADTM) Wistuba et al. (2016) in Figure 14, illustrating how quickly each method converges to its best solution during the HPO process.

The plot shows that XGBoost is the fastest, reaching nearly optimal performance within just 5 hours. ResNet and MLP also demonstrate notable speed, followed closely by CatBoost. Overall, the gradient boosting methods (GBDT) converge faster than the deep learning models. XTab, which shares the same transformer architecture as FT-Transformer, exhibits quicker convergence, likely due to its static architecture, while the FT-Transformer's architectural components were also tuned. On the other hand, TP-BERTa is the slowest to converge, likely due to the high computational demands of its BERT-like architecture.

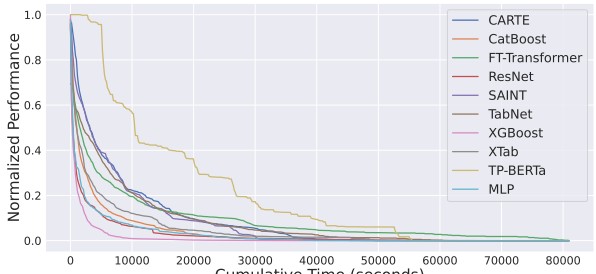

Figure 14: Intra-search space normalized average distance to the maximum over the cumulative training time (seconds).

In Figure 15, we show the performance profiles of the models considered. We first normalize the performance values and the logarithmic time values. Let $P_i^{(j)}$ and $T_i^{(j)}$ be the performance of algorithm $i$ on dataset $j$, resp., the measured time. Let $m_*^{(j)} = \max_i P_i^{(j)}$ be the model yielding the best performance and $t_\dagger^{(j)} = \max_i T_i^{(j)}$ denoting the greatest running time on dataset $j$, we compute the performance gap $\mathrm{gap}_i^{(j)} = (m_*^{(j)} - P_i(j))/m_*^{(j)}$ and the temporal gain $\mathrm{tgain}_i^{(j)} = (t_\dagger^{(j)} - T_i^{(j)})/t_\dagger^{(j)}$ being achieved for each algorithm $i$ on a dataset $j$. We define a *Performance-Time Ratio* $\mathrm{ptr}_i^{(j)} = \mathrm{gap}_i^{(j)} / \mathrm{tgain}_i^{(j)}$, where $\mathrm{ptr}_i^{(j)} > 1$ values indicate that the performance gain outweighs the time cost, $\mathrm{ptr}_i^{(j)} = 1$ yields a balanced view, whereas $\mathrm{ptr}_i^{(j)} < 1$ indicate that the time cost outweighs the performance gain. We further normalize the $\mathrm{ptr}_i^{(j)}$ to be in the range of $[0, 1]$ such that values closer to 1 (0) indicate a better (poorer) trade-off in terms of performance gain relative to time cost. To determine the cumulative proportion for a given threshold $\tau$, we count how many normalized $\mathrm{ptr'}_i^{(j)} = \mathrm{ptr}_i^{(j)} / (1 + \mathrm{ptr}_i^{(j)})$ are less than or equal to $\tau$ for each algorithm $i$. Hence, the closer a line reaches 1.0 for smaller values of $\tau$, the more often an algorithm is close to having the best performance-time-cost, i.e., lines that rise more steeply indicate better overall ratios on the datasets considered. A red star in the upper left of the illustrations indicates the best value.

On the left, the performance profiles are shown w.r.t. the measured inference time. The evaluation shows that both GBDT models yield the best performance-time ratio, followed by the dataset-specific deep learning models FT-Transformer and ResNet. Even though the AutoML framework AutoGluon shows strong performance values as discussed in Section 5.3, the hyperparameter optimization comes at the cost of higher computational cost resulting in expensive costs w.r.t. the temporal domain. Both TP-Berta and TabPFN are only evaluated on a small subset of the available datasets as indicated by the small cumulative proportion value for both approaches, where the latter shows its strong performance on the datasets it has been evaluated on by a steep increase. The approaches SAINT, TabNet, CARTE, and XTab range in-between, where CARTE and SAINT show slightly better performance-cost ratios compared to AutoGluon with increased values for the performance ratio $\tau$ considering a broader range of various datasets.

---

[3]https://pymfe.readthedocs.io/en/latest/auto_pages/meta_features_description.html

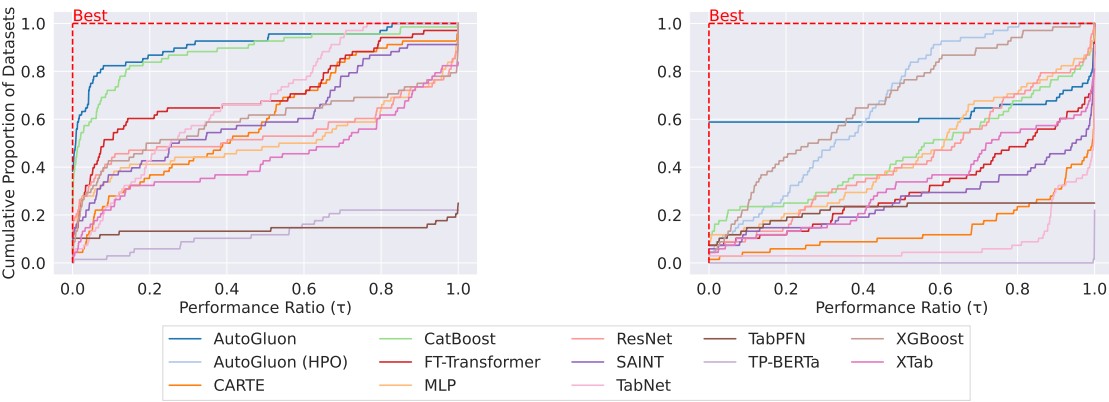

Figure 15: **Left:** Performance profiles based on inference time. **Right:** Performance profiles based on total time. Steeper curves indicate better overall performance and efficiency across datasets.

The right plot shows the equivalent performance profiles w.r.t. the measured total time. Notably, XGBoost and AutoGluon with hyperparameter optimization result in strong performance-time ratios. Transformer-based models are outperformed by more lightweight models like CatBoost and ResNet which both show competitive results. From the model family encompassing foundation models, XTab shows the strongest performance, however, is outperformed by classical GBDT approaches. TabPFN as an in-context learning model is only applicable on small data regimes and is therefore not competitive considering all datasets. In Appendix E.6, we provide a performance profiles analysis in the small-data and large-data regimes separately.

## 6 Conclusion

Our comprehensive empirical study evaluates the quality of eleven state-of-the-art tabular classification models. We categorize the approaches into model families according to their underlying learning scheme, which encompasses gradient-boosted decision trees (GBDTs) as tree-based models, foundation models with sub-paradigms of in-context learning and fine-tuning, and dataset-specific neural networks as umbrella for feedforward networks and transformer-based approaches. Our study is conducted with 68 diverse datasets from the OpenMLCC18 benchmark repository. Our study provides a rigorous assessment of state-of-the-art learning paradigms that reveals that dataset-specific neural networks, e.g., ResNet, FT-Transformer, and SAINT, generally outperform meta-learned approaches, e.g., TP-BERTa. We are the first to have a deeper investigation on foundation models, showing that TabPFN excels all other models in small data scenarios. However, classical machine learning methods, such as XGBoost and CatBoost still demonstrate robust performance while highly efficient in comparison to other model families on a broad range of datasets. Moreover, AutoGluon as an AutoML framework exhibits superior performance across diverse datasets, albeit at the cost of increased computational resources. Next to a fair comparison of model families, we provide an in-depth analysis of the influence of hyperparameter optimization on the models' performance and provide a cost-efficiency analysis that highlights that GBDT approaches outperform most modern deep learning methods. Our study contributes valuable insights into the current landscape of tabular classification data modeling and encourages further potential research directions with promising model families.

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

## A  Configuration Spaces

### A.1  CatBoost

Table 2: Search space for CatBoost.

| Parameter | Type | Range | Log Scale |
|---|---|---|---|
| max_depth | Integer | $[3, 10]$ | |
| learning_rate | Float | $[10^{-5}, 1]$ | ✓ |
| bagging_temperature | Float | $[0, 1]$ | |
| l2_leaf_reg | Float | $[1, 10]$ | ✓ |
| leaf_estimation_iterations | Integer | $[1, 10]$ | |
| iterations | Integer | $[100, 2000]$ | |

The specific search space employed for CatBoost is detailed in Table 2. Our implementation heavily relies on the framework provided by the official implementation of the FT-Transformer, as found in the following repository[4]. We do this to ensure a consistent pipeline across all methods, that we compare. The CatBoost algorithm implementation, however, is the official one[5].

For the default configuration of CatBoost, we do not modify any hyperparameter values. This approach allows the library to automatically apply its default settings, ensuring that our implementation is aligned with the most typical usage scenarios of the library.

### A.2  XGBoost

Table 3: Search space for XGBoost.

| Parameter | Type | Range | Log Scale |
|---|---|---|---|
| max_depth | Integer | $[3, 10]$ | |
| min_child_weight | Float | $[10^{-8}, 10^5]$ | ✓ |
| subsample | Float | $[0.5, 1]$ | |
| learning_rate | Float | $[10^{-5}, 1]$ | ✓ |
| colsample_bylevel | Float | $[0.5, 1]$ | |
| colsample_bytree | Float | $[0.5, 1]$ | |
| gamma | Float | $[10^{-8}, 10^2]$ | ✓ |
| reg_lambda | Float | $[10^{-8}, 10^2]$ | ✓ |
| reg_alpha | Float | $[10^{-8}, 10^2]$ | ✓ |
| n_estimators | Integer | $[100, 2000]$ | |

We utilized the official XGBoost implementation[6]. While the data preprocessing steps were consistent across all methods, a notable exception was made for XGBoost. For this method, we implemented one-hot encoding on categorical features, as XGBoost does not inherently process categorical values.

The comprehensive search space for the XGBoost hyperparameters is detailed in Table 3. In the case of default hyperparameters, our approach mirrored the CatBoost implementation where we opted not to set any hyperparameters explicitly but instead, use the library defaults.

---

[4]https://github.com/yandex-research/rtdl-revisiting-models
[5]https://catboost.ai/
[6]https://xgboost.readthedocs.io/en/stable/

Furthermore, it is important to note that XGBoost lacks native support for the ROC-AUC metric in multiclass problems. To address this, we incorporated a custom ROC-AUC evaluation function. This function first applies a softmax to the predictions and then employs the ROC-AUC scoring functionality provided by scikit-learn, which can be found at the following link[7].

### A.3 FT-Transformer

Table 4: Search space for FT-Transformer.

| Parameter | Type | Range | Log Scale |
|---|---|---|---|
| n_layers | Integer | $[1, 6]$ | |
| d_token | Integer | $[64, 512]$ | |
| residual_dropout | Float | $[0, 0.2]$ | |
| attn_dropout | Float | $[0, 0.5]$ | |
| ffn_dropout | Float | $[0, 0.5]$ | |
| d_ffn_factor | Float | $[\frac{2}{3}, \frac{8}{3}]$ | |
| lr | Float | $[10^{-5}, 10^{-3}]$ | ✓ |
| weight_decay | Float | $[10^{-6}, 10^{-3}]$ | ✓ |
| epochs | Integer | $[10, 500]$ | |

In our investigation, we adopted the official implementation of the FT-Transformer (Gorishniy et al., 2021). Diverging from the approach from the original study, we implemented a uniform search space applicable to all datasets, rather than customizing the search space for each specific dataset. This approach ensures a consistent and comparable application across various datasets. The uniform search space we employed aligns with the structure proposed in Gorishniy et al. (2021). Specifically, we consolidated the search space by integrating the upper bounds defined in the original paper with the minimum bounds identified across different datasets.

Regarding the default hyperparameters, we adhered strictly to the specifications provided in Gorishniy et al. (2021).

### A.4 SAINT

We utilize the official implementation of the method as detailed by the respective authors (Somepalli et al., 2021). The comprehensive search space employed for hyperparameter tuning is illustrated in Table 5.

Regarding the default hyperparameters, we adhere to the specifications provided by the authors in their original implementation.

Table 5: Search space for SAINT.

| Parameter | Type | Range | Log Scale |
|---|---|---|---|
| embedding_size | Categorical | $\{4, 8, 16, 32\}$ | |
| transformer_depth | Integer | $[1, 4]$ | |
| attention_dropout | Float | $[0, 1.0]$ | |
| ff_dropout | Float | $[0, 1.0]$ | |
| lr | Float | $[10^{-5}, 10^{-3}]$ | ✓ |
| weight_decay | Float | $[10^{-6}, 10^{-3}]$ | ✓ |
| epochs | Integer | $[10, 500]$ | |

---

[7]https://scikit-learn.org/stable/modules/generated/sklearn.metrics.roc_auc_score.html

### A.5 TabNet

Table 6: Search space for TabNet.

| Parameter | Type | Range | Log Scale |
|---|---|---|---|
| n_a | Integer | [8, 64] | |
| n_d | Integer | [8, 64] | |
| gamma | Float | [1.0, 2.0] | |
| n_steps | Integer | [3, 10] | |
| cat_emb_dim | Integer | [1, 3] | |
| n_independent | Integer | [1, 5] | |
| n_shared | Integer | [1, 5] | |
| momentum | Float | [0.001, 0.4] | ✓ |
| mask_type | Categorical | {entmax, sparsemax} | |
| epochs | Integer | [10, 500] | |

For TabNet's implementation, we utilized a well-maintained and publicly available version, accessible at the following link[8]. The hyperparameter tuning search space for TabNet, detailed in Table 6, was derived from McElfresh et al. (2023).

Regarding the default hyperparameters, we followed the recommendations provided by the original authors.

### A.6 ResNet

Table 7: Search space for ResNet.

| Parameter | Type | Range | Log Scale |
|---|---|---|---|
| layer_size | Integer | [64, 1024] | |
| lr | Float | $[10^{-5}, 10^{-2}]$ | ✓ |
| weight_decay | Float | $[10^{-6}, 10^{-3}]$ | ✓ |
| residual_dropout | Float | [0, 0.5] | |
| hidden_dropout | Float | [0, 0.5] | |
| n_layers | Integer | [1, 8] | |
| d_embedding | Integer | [64, 512] | |
| d_hidden_factor | Float | [1.0, 4.0] | |
| epochs | Integer | [10, 500] | |

We employed the ResNet implementation as described in prior work (Gorishniy et al., 2021). The entire range of hyperparameters explored for ResNet tuning is detailed in Table 7. Since the original study did not specify default hyperparameter values, we relied on the search space provided in a prior work (Kadra et al., 2021).

---

[8]https://github.com/dreamquark-ai/tabnet

### A.7 MLP-PLR

We employ the MLP implementation proposed by (Gorishniy et al., 2022). The search space used for hyperparameter optimization is detailed in Table 8. Default hyperparameters are adapted from (McElfresh et al., 2023), while the search space is based on the original work of (Gorishniy et al., 2022).

Table 8: Search space for MLP-PLR.

| Parameter | Type | Range | Log Scale |
|---|---|---|---|
| lr | Float | $[10^{-5}, 10^{-3}]$ | ✓ |
| weight_decay | Float | $[10^{-6}, 10^{-3}]$ | ✓ |
| dropout | Float | $[0, 0.5]$ | |
| n_layers | Integer | $[1, 16]$ | |
| d_embedding | Integer | $[64, 512]$ | |
| d_num_embedding | Integer | $[1, 128]$ | |
| d_first | Integer | $[1, 1024]$ | |
| d_middle | Integer | $[1, 1024]$ | |
| d_last | Integer | $[1, 1024]$ | |
| n | Integer | $[1, 128]$ | |
| sigma | Float | $[0.01, 100]$ | ✓ |
| epochs | Integer | $[10, 500]$ | |

### A.8 RealMLP

For our RealMLP experiments, we use the official implementation [9]. Following the authors' recommendations, we impute missing values using the mean of the training split before applying their preprocessing pipeline. We adopt the recommended default hyperparameters and search space, detailed in Table 9. Additionally, we extend the search space for the initialization standard deviation of the first embedding layer and tune the embedding dimensions, as done for the MLP, to achieve optimal performance.

Table 9: Search space for RealMLP.

| Parameter | Type | Range | Log Scale |
|---|---|---|---|
| num_emb_type | Categorical | {None, PBLD, PL, PLR} | |
| add_front_scale | Categorical | {True, False} | |
| lr | Float | [2e-2, 3e-1] | ✓ |
| p_drop | Categorical | {0.0, 0.15, 0.3} | |
| act | Categorical | {relu, selu, mish} | |
| hidden_sizes | Categorical | {[256, 256, 256], [64, 64, 64, 64, 64], [512]} | |
| wd | Categorical | {0.0, 0.02} | |
| plr_sigma | Float | [0.05, 1e1] | ✓ |
| ls_eps | Categorical | {0.0, 0.1} | |
| embedding_size | Integer | [1, 64] | |
| n_epochs | Integer | [10, 500] | |

---

[9]https://github.com/dholzmueller/pytabkit

### A.9    XTab

For XTab, we utilize the official implementation[10]. To ensure comparability with other methods, we decouple XTab from AutoGluon and apply the same preprocessing and training pipeline as used for the other models. The original work reports results for both light finetuning and heavy finetuning, so we introduce this as a categorical hyperparameter. If `light_finetuning` is set to `True`, the model is finetuned for only 3 epochs. Otherwise, we follow the same epoch range as for the other methods, i.e., $[10, 500]$. Furthermore, we use the checkpoint after 2000 iterations (`iter_2k.ckpt`), provided by the authors. Table 10 outlines the complete search space used for XTab during hyperparameter optimization (HPO).

Table 10: Search space for XTab.

| Parameter | Type | Range | Log Scale |
|---|---|---|---|
| lr | Float | $[10^{-5}, 10^{-3}]$ | ✓ |
| weight_decay | Float | $[10^{-6}, 10^{-3}]$ | ✓ |
| light_finetuning | Categorical | {True, False} | |
| epochs | Integer | 3 (if light_finetuning=True) or $[10, 500]$ (otherwise) | |

### A.10    CARTE

For CARTE, we use the official implementation[11]. Similar to XTab, since it is a pretrained model, we do not tune the architectural hyperparameters but keep them fixed and load the checkpoint provided by the authors. The search space used for CARTE during our hyperparameter optimization (HPO) process is shown in Table 11.

Table 11: Search space for CARTE.

| Parameter | Type | Range | Log Scale |
|---|---|---|---|
| lr | Float | $[10^{-5}, 10^{-3}]$ | ✓ |
| weight_decay | Float | $[10^{-6}, 10^{-3}]$ | ✓ |
| epochs | Integer | $[10, 500]$ | |

### A.11    TP-BERTa

We use the official implementation for TP-BERTa[12]. Similar to the other pretrained models, we only tune the `learning rate`, `weight decay`, and the number of finetuning `epochs`. The search space is shown in Table 12.

Table 12: Search space for TP-BERTa.

| Parameter | Type | Range | Log Scale |
|---|---|---|---|
| lr | Float | $[10^{-5}, 10^{-3}]$ | ✓ |
| weight_decay | Float | $[10^{-6}, 10^{-3}]$ | ✓ |
| epochs | Integer | $[10, 500]$ | |

---

[10]https://github.com/BingzhaoZhu/XTab
[11]https://github.com/soda-inria/carte
[12]https://github.com/jyansir/tp-berta

### A.12    TabPFN

For TabPFN, we utilized the official implementation from the authors[13]. We followed the settings suggested by the authors and we did not preprocess the numerical features as TabPFN does that natively, we ordinally encoded the categorical features and we used an ensemble size of 32 to achieve peak performance as suggested by the authors.

### A.13    AutoGluon

For our experiments, we utilize the official implementation of AutoGluon[14]. Specifically, we evaluate two configurations of AutoGluon: the HPO version and the recommended version.

- For the HPO version, we use the default search spaces for the models included in AutoGluon's ensemble.

- For the recommended version, we set `presets="best_quality"` as per the official documentation and do not perform hyperparameter optimization.

### A.14    Preprocessing

Our codebase is heavily based on that of (Gorishniy et al., 2021), and we employ the same preprocessing pipeline. Methods for which we do not apply this preprocessing are those that inherently require a different approach, such as TP-BERTa and CARTE, or those implemented within libraries where modifying the preprocessing pipeline is not trivial, such as AutoGluon and RealMLP. In these cases, we use the preprocessing strategies from the original works.

Regarding batch size, we do not tune it in our experiments due to memory constraints. Instead, we determine batch size heuristically similar to (Chen et al., 2024) based on the number of features in the dataset. While batch sizes may vary across datasets, they remain consistent across methods.

---

[13]https://github.com/automl/TabPFN
[14]https://auto.gluon.ai/stable/index.html

# B  Raw results tables

## B.1  Results after hyperparameter optimization

Table 13 shows the raw results after HPO for CatBoost and XGBoost.

Table 13: Average test ROC-AUC per dataset for XGBoost and CatBoost after hyperparameter optimization across CV folds.

| Dataset | CatBoost | XGBoost |
|---|---|---|
| adult | **0.930747** | 0.930482 |
| analcatdata_authorship | 0.999662 | **0.999816** |
| analcatdata_dmft | **0.579136** | 0.572150 |
| balance-scale | 0.972625 | **0.991268** |
| bank-marketing | **0.938831** | 0.938384 |
| banknote-authentication | **0.999935** | **0.999935** |
| Bioresponse | 0.885502 | **0.888615** |
| blood-transfusion-service-center | **0.754965** | 0.750671 |
| breast-w | 0.989162 | **0.992112** |
| car | **1.000000** | 0.999902 |
| churn | **0.922968** | 0.914432 |
| climate-model-simulation-crashes | **0.951480** | 0.947000 |
| cmc | **0.740149** | 0.735649 |
| cnae-9 | 0.996316 | **0.997454** |
| connect-4 | 0.921050 | **0.931952** |
| credit-approval | 0.934006 | **0.934692** |
| credit-g | **0.801762** | 0.798571 |
| cylinder-bands | 0.912070 | **0.928116** |
| diabetes | **0.837869** | 0.835638 |
| dna | 0.995028 | **0.995278** |
| dresses-sales | 0.595731 | **0.622414** |
| electricity | 0.980993 | **0.987790** |
| eucalyptus | **0.923334** | 0.918055 |
| first-order-theorem-proving | 0.831775 | **0.834883** |
| GesturePhaseSegmentationProcessed | 0.916674 | **0.916761** |
| har | 0.999941 | **0.999960** |
| ilpd | 0.744702 | **0.748019** |
| Internet-Advertisements | 0.979120 | **0.982276** |
| isolet | 0.999389 | **0.999488** |
| jm1 | 0.756611 | **0.759652** |
| jungle_chess_2pcs_raw_endgame_complete | **0.976349** | 0.974087 |
| kc1 | 0.825443 | **0.832007** |
| kc2 | **0.846802** | 0.843295 |
| kr-vs-kp | 0.999392 | **0.999796** |
| letter | **0.999854** | 0.999819 |
| madelon | **0.937562** | 0.932249 |
| mfeat-factors | 0.998910 | **0.999004** |
| mfeat-fourier | **0.984714** | 0.983375 |
| mfeat-karhunen | **0.999264** | 0.999211 |
| mfeat-morphological | **0.965406** | 0.963075 |
| mfeat-pixel | **0.999422** | 0.999378 |
| mfeat-zernike | **0.977986** | 0.974231 |
| MiceProtein | **1.000000** | 0.999923 |
| nomao | 0.996439 | **0.996676** |
| numerai28.6 | 0.529404 | **0.529457** |
| optdigits | 0.999844 | **0.999855** |
| ozone-level-8hr | **0.929094** | 0.922663 |
| pc1 | **0.875471** | 0.863061 |
| pc3 | 0.851122 | **0.854543** |
| pc4 | **0.953309** | 0.951037 |
| pendigits | **0.999752** | 0.999703 |
| PhishingWebsites | 0.996482 | **0.997425** |
| phoneme | **0.968024** | 0.967421 |
| qsar-biodeg | 0.930649 | **0.934875** |
| satimage | 0.991978 | **0.992114** |
| segment | **0.996231** | 0.996126 |
| semeion | **0.998687** | 0.998272 |
| sick | **0.998331** | 0.997950 |
| spambase | 0.989935 | **0.990726** |
| splice | **0.995472** | 0.995049 |
| steel-plates-fault | **0.974350** | 0.972743 |
| texture | **0.999948** | 0.999940 |
| tic-tac-toe | **1.000000** | 0.999710 |
| vehicle | **0.943460** | 0.942080 |
| vowel | 0.999259 | **0.999428** |
| wall-robot-navigation | **0.999990** | 0.999981 |
| wdbc | 0.993813 | **0.994467** |
| wilt | 0.990950 | **0.992192** |

Table 14 shows the raw results after HPO for dataset-specific neural networks.

Table 14: Average test ROC-AUC per dataset for dataset-specific neural networks after hyperparameter optimization across CV folds. Missing datasets are represented by "-".

| Dataset | FT-Transformer | ResNet | SAINT | TabNet |
|---|---|---|---|---|
| adult | 0.914869 | 0.913790 | **0.920246** | 0.882450 |
| analcatdata_authorship | 0.999985 | **1.000000** | 0.999974 | 0.999249 |
| analcatdata_dmft | 0.576947 | **0.584338** | 0.544695 | 0.515962 |
| balance-scale | **0.999735** | 0.989061 | 0.999266 | 0.979668 |
| bank-marketing | **0.938198** | 0.935740 | 0.936560 | 0.887319 |
| banknote-authentication | **1.000000** | **1.000000** | **1.000000** | **1.000000** |
| Bioresponse | 0.820159 | **0.850801** | - | - |
| blood-transfusion-service-center | 0.745975 | 0.738502 | **0.746726** | 0.660675 |
| breast-w | 0.989503 | **0.995477** | 0.988470 | 0.986694 |
| car | 0.999751 | 0.994154 | **1.000000** | **1.000000** |
| churn | 0.914596 | **0.918713** | 0.915603 | 0.891443 |
| climate-model-simulation-crashes | **0.934671** | 0.918990 | 0.925643 | 0.868204 |
| cmc | **0.739402** | 0.737829 | 0.738490 | 0.647121 |
| cnae-9 | 0.994497 | **0.997106** | - | - |
| connect-4 | 0.901170 | **0.933333** | - | - |
| credit-approval | **0.935798** | 0.933113 | 0.933493 | 0.878500 |
| credit-g | 0.783048 | 0.783524 | **0.786402** | 0.696905 |
| cylinder-bands | 0.915494 | 0.909989 | **0.923391** | 0.837792 |
| diabetes | **0.831108** | 0.821798 | 0.827285 | 0.756416 |
| dna | 0.990937 | **0.992543** | 0.992473 | 0.991448 |
| dresses-sales | 0.620033 | 0.575205 | **0.624704** | 0.555993 |
| electricity | 0.963076 | 0.960658 | **0.967012** | 0.938656 |
| eucalyptus | 0.923933 | 0.916785 | **0.925970** | 0.872365 |
| first-order-theorem-proving | 0.796707 | 0.784636 | **0.802392** | 0.774094 |
| GesturePhaseSegmentationProcessed | 0.895166 | 0.914196 | **0.919006** | 0.850596 |
| har | 0.999685 | **0.999921** | - | 0.999515 |
| ilpd | **0.751488** | 0.747491 | 0.698718 | 0.704840 |
| Internet-Advertisements | **0.974513** | 0.974187 | - | - |
| isolet | 0.998817 | **0.999401** | - | 0.998813 |
| jm1 | 0.709321 | **0.720444** | 0.719464 | 0.674043 |
| jungle_chess_2pcs_raw_endgame_complete | **0.999975** | 0.999956 | 0.999926 | 0.991981 |
| kc1 | 0.783519 | **0.806819** | 0.796918 | 0.762807 |
| kc2 | 0.832014 | 0.833248 | **0.834436** | 0.713458 |
| kr-vs-kp | 0.999777 | 0.999369 | **0.999789** | 0.998872 |
| letter | 0.999919 | **0.999926** | 0.999853 | 0.999606 |
| madelon | **0.747391** | 0.605018 | - | 0.630669 |
| mfeat-factors | 0.999015 | **0.999472** | 0.999385 | 0.998125 |
| mfeat-fourier | **0.984511** | 0.981725 | 0.980508 | 0.970539 |
| mfeat-karhunen | 0.998682 | 0.998448 | **0.999078** | 0.996960 |
| mfeat-morphological | **0.970198** | 0.968651 | 0.967681 | 0.955818 |
| mfeat-pixel | 0.997451 | 0.998690 | **0.999217** | 0.998200 |
| mfeat-zernike | 0.983479 | **0.984488** | 0.981874 | 0.968629 |
| MiceProtein | 0.999973 | 0.999973 | **1.000000** | 0.999344 |
| nomao | 0.990908 | **0.993048** | - | - |
| numerai28.6 | **0.530315** | 0.528012 | 0.525822 | - |
| optdigits | 0.999616 | **0.999927** | 0.999841 | 0.998871 |
| ozone-level-8hr | 0.919484 | **0.925416** | 0.919315 | 0.864067 |
| pc1 | **0.917591** | 0.889458 | 0.870543 | 0.804412 |
| pc3 | 0.828743 | **0.829637** | 0.827322 | 0.788151 |
| pc4 | 0.934944 | **0.944447** | 0.934528 | 0.920943 |
| pendigits | 0.999703 | 0.999638 | **0.999782** | 0.999753 |
| PhishingWebsites | 0.996760 | **0.996975** | 0.996746 | 0.996196 |
| phoneme | **0.965071** | 0.963591 | 0.960382 | 0.956279 |
| qsar-biodeg | 0.919584 | **0.932220** | 0.930632 | 0.902748 |
| satimage | **0.993516** | 0.991995 | 0.992630 | 0.987482 |
| segment | 0.994124 | 0.993581 | **0.994831** | 0.992317 |
| semeion | 0.995548 | **0.997689** | 0.997630 | 0.994019 |
| sick | 0.997937 | 0.968841 | **0.998281** | 0.981838 |
| spambase | 0.985969 | **0.987683** | 0.986263 | 0.980804 |
| splice | 0.992276 | 0.993514 | **0.995073** | 0.990441 |
| steel-plates-fault | **0.959182** | 0.949067 | 0.955379 | 0.947456 |
| texture | 0.999983 | **0.999999** | 0.999976 | 0.999763 |
| tic-tac-toe | 0.996152 | 0.999462 | **0.999725** | 0.993030 |
| vehicle | 0.963362 | **0.967212** | 0.955127 | 0.943787 |
| vowel | 0.999713 | 0.999813 | **0.999875** | 0.999686 |
| wall-robot-navigation | **0.999900** | 0.999042 | 0.999844 | 0.997585 |
| wdbc | 0.993967 | 0.995409 | **0.995546** | 0.986656 |
| wilt | 0.993047 | 0.990726 | **0.993139** | 0.991289 |

Table 15 shows the raw results after HPO for MLP with PLR embeddings and RealMLP.

Table 15: Average test ROC-AUC per dataset for MLP and RealMLP after hyperparameter optimization across CV folds. Missing datasets are represented by "-".

| Dataset | MLP | RealMLP |
|---|---|---|
| adult | **0.928689** | 0.923327 |
| analcatdata_authorship | 0.999770 | **1.000000** |
| analcatdata_dmft | **0.574532** | 0.574396 |
| balance-scale | 0.998659 | **1.000000** |
| bank-marketing | **0.937054** | 0.937031 |
| banknote-authentication | **1.000000** | **1.000000** |
| Bioresponse | 0.825631 | **0.859065** |
| blood-transfusion-service-center | **0.770627** | 0.746350 |
| breast-w | 0.992380 | **0.992882** |
| car | 0.999992 | **1.000000** |
| churn | **0.922938** | 0.913533 |
| climate-model-simulation-crashes | 0.948857 | **0.962163** |
| cmc | **0.744580** | 0.735472 |
| cnae-9 | 0.996716 | **0.997569** |
| connect-4 | 0.927373 | **0.928258** |
| credit-approval | **0.938866** | 0.917352 |
| credit-g | **0.788476** | 0.779381 |
| cylinder-bands | 0.886405 | **0.910680** |
| diabetes | 0.837342 | **0.837507** |
| dna | 0.992220 | **0.994111** |
| dresses-sales | **0.635468** | 0.537849 |
| electricity | **0.969201** | 0.961467 |
| eucalyptus | **0.921873** | 0.915693 |
| first-order-theorem-proving | **0.798812** | 0.795637 |
| GesturePhaseSegmentationProcessed | **0.911434** | 0.901441 |
| har | 0.999783 | **0.999959** |
| ilpd | 0.671938 | **0.729412** |
| Internet-Advertisements | - | **0.973810** |
| isolet | 0.998295 | **0.999635** |
| jm1 | **0.715620** | 0.713988 |
| jungle_chess_2pcs_raw_endgame_complete | **0.999965** | 0.999774 |
| kc1 | **0.805465** | 0.796117 |
| kc2 | 0.829426 | **0.845768** |
| kr-vs-kp | 0.999686 | **0.999704** |
| letter | 0.999894 | **0.999914** |
| madelon | 0.883991 | **0.930302** |
| mfeat-factors | 0.998875 | **0.999625** |
| mfeat-fourier | 0.984929 | **0.985483** |
| mfeat-karhunen | 0.998849 | **0.999019** |
| mfeat-morphological | 0.967719 | **0.969994** |
| mfeat-pixel | 0.998674 | **0.999492** |
| mfeat-zernike | **0.984610** | 0.982993 |
| MiceProtein | **0.999973** | 0.999971 |
| nomao | 0.986577 | **0.989803** |
| numerai28.6 | 0.525920 | **0.529534** |
| optdigits | 0.999794 | **0.999968** |
| ozone-level-8hr | **0.927900** | 0.923252 |
| pc1 | 0.832532 | **0.844517** |
| pc3 | **0.842511** | 0.814590 |
| pc4 | **0.945813** | 0.939257 |
| pendigits | **0.999705** | 0.999659 |
| PhishingWebsites | 0.996991 | **0.997208** |
| phoneme | **0.967617** | 0.966456 |
| qsar-biodeg | 0.924951 | **0.929226** |
| satimage | 0.992308 | **0.993034** |
| segment | **0.995046** | 0.994075 |
| semeion | 0.997350 | **0.998976** |
| sick | 0.997048 | **0.998661** |
| spambase | **0.988185** | 0.987799 |
| splice | 0.994053 | **0.994420** |
| steel-plates-fault | **0.964693** | 0.959639 |
| texture | 0.999991 | **0.999999** |
| tic-tac-toe | **1.000000** | 0.999711 |
| vehicle | 0.961813 | **0.965844** |
| vowel | 0.999638 | **0.999955** |
| wall-robot-navigation | **0.999689** | 0.998720 |
| wdbc | **0.996065** | 0.996038 |
| wilt | **0.997690** | 0.993197 |

Table 16 shows the raw results after HPO for the meta-learned neural networks.

Table 16: Average test ROC-AUC per dataset for meta-learned neural networks after hyperparameter optimization across CV folds. Missing datasets are represented by "-".

| Dataset | CARTE | TPBerta | TabPFN | XTab |
|---|---|---|---|---|
| adult | **0.902677** | - | - | - |
| analcatdata_authorship | 0.999181 | - | **1.000000** | 0.999991 |
| analcatdata_dmft | 0.586376 | - | **0.586630** | 0.556971 |
| balance-scale | **0.999413** | - | 0.997656 | 0.997420 |
| bank-marketing | **0.924664** | - | - | - |
| banknote-authentication | **1.000000** | 0.994512 | - | **1.000000** |
| Bioresponse | - | - | - | - |
| blood-transfusion-service-center | 0.739571 | 0.633041 | **0.752586** | - |
| breast-w | 0.987912 | 0.986514 | **0.994131** | 0.989666 |
| car | **0.997126** | - | - | - |
| churn | **0.923626** | - | - | - |
| climate-model-simulation-crashes | 0.938531 | - | **0.968010** | 0.944367 |
| cmc | **0.738379** | - | - | - |
| cnae-9 | **0.990151** | - | - | - |
| connect-4 | - | - | - | - |
| credit-approval | 0.909279 | 0.901989 | 0.932397 | **0.939620** |
| credit-g | **0.769619** | - | 0.768476 | - |
| cylinder-bands | 0.848539 | 0.820399 | **0.886616** | 0.881396 |
| diabetes | 0.823615 | 0.778356 | **0.836120** | 0.815847 |
| dna | 0.986120 | - | - | **0.992479** |
| dresses-sales | 0.589655 | 0.534893 | 0.538916 | **0.613136** |
| electricity | 0.909407 | - | - | **0.966899** |
| eucalyptus | 0.905245 | - | **0.928493** | 0.918317 |
| first-order-theorem-proving | 0.764092 | - | - | **0.798803** |
| GesturePhaseSegmentationProcessed | 0.798024 | - | - | **0.886960** |
| har | - | - | - | - |
| ilpd | 0.704712 | 0.586083 | **0.757892** | 0.726413 |
| Internet-Advertisements | - | - | - | - |
| isolet | - | - | - | - |
| jm1 | **0.728512** | - | - | 0.727984 |
| jungle_chess_2pcs_raw_endgame_complete | 0.973383 | - | - | **0.999950** |
| kc1 | 0.797680 | - | - | **0.803082** |
| kc2 | 0.842828 | - | **0.850065** | 0.835476 |
| kr-vs-kp | **0.999685** | 0.855273 | - | 0.999616 |
| letter | 0.999440 | - | - | **0.999859** |
| madelon | 0.836760 | - | - | **0.845746** |
| mfeat-factors | 0.996064 | - | - | **0.998443** |
| mfeat-fourier | 0.976986 | - | - | **0.982539** |
| mfeat-karhunen | 0.994814 | - | - | **0.998582** |
| mfeat-morphological | **0.967325** | - | - | 0.967136 |
| mfeat-pixel | 0.996175 | - | - | **0.998642** |
| mfeat-zernike | 0.978119 | - | - | **0.980183** |
| MiceProtein | 0.999582 | - | - | **1.000000** |
| nomao | - | - | - | **0.992727** |
| numerai28.6 | 0.514361 | - | - | **0.528062** |
| optdigits | 0.999112 | - | - | **0.999712** |
| ozone-level-8hr | 0.890063 | - | - | **0.915744** |
| pc1 | 0.835444 | - | - | **0.855741** |
| pc3 | **0.831574** | 0.625642 | - | 0.823532 |
| pc4 | 0.937337 | 0.744304 | - | **0.938455** |
| pendigits | 0.999468 | - | - | **0.999751** |
| PhishingWebsites | 0.994582 | - | - | **0.996896** |
| phoneme | 0.948702 | 0.796404 | - | **0.961749** |
| qsar-biodeg | 0.921153 | 0.833852 | - | **0.926795** |
| satimage | 0.988038 | - | - | **0.992918** |
| segment | 0.993491 | - | - | **0.994697** |
| semeion | 0.993378 | - | - | **0.997064** |
| sick | 0.995762 | - | - | **0.998232** |
| spambase | 0.983228 | - | - | **0.986044** |
| splice | 0.987950 | - | - | **0.992444** |
| steel-plates-fault | 0.943636 | - | - | **0.957088** |
| texture | 0.999541 | - | - | **0.999962** |
| tic-tac-toe | 0.984361 | 0.993803 | 0.996086 | **1.000000** |
| vehicle | 0.941691 | - | **0.970556** | 0.955838 |
| vowel | 0.998092 | - | - | **0.999630** |
| wall-robot-navigation | 0.999505 | - | - | **0.999846** |
| wdbc | 0.990612 | - | **0.996298** | 0.994317 |
| wilt | **0.994858** | 0.880733 | - | 0.994261 |

Lastly, Table 17 shows the raw results of AutoGluon using HPO and AutoGluon with its recommended settings.

Table 17: Average test ROC-AUC per dataset for AutoGluon with HPO and AutoGluon with its recommended settings across CV folds.

| Dataset | AutoGluon | AutoGluon (HPO) |
|---|---|---|
| adult | **0.931792** | 0.931658 |
| analcatdata_authorship | **1.000000** | 0.999887 |
| analcatdata_dmft | **0.577809** | 0.553672 |
| balance-scale | **0.997339** | 0.995057 |
| bank-marketing | **0.941273** | 0.940659 |
| banknote-authentication | **1.000000** | 0.999957 |
| Bioresponse | **0.888693** | 0.881238 |
| blood-transfusion-service-center | **0.741733** | 0.733305 |
| breast-w | **0.994394** | 0.993510 |
| car | 0.999861 | **0.999998** |
| churn | **0.927520** | 0.920213 |
| climate-model-simulation-crashes | **0.970051** | 0.926306 |
| cmc | **0.737077** | 0.536500 |
| cnae-9 | **0.998524** | 0.997965 |
| connect-4 | 0.934636 | **0.941976** |
| credit-approval | **0.940476** | 0.933497 |
| credit-g | **0.802381** | 0.773238 |
| cylinder-bands | **0.933320** | 0.903658 |
| diabetes | **0.833641** | 0.827171 |
| dna | **0.995385** | 0.994906 |
| dresses-sales | **0.615107** | 0.597537 |
| electricity | **0.987260** | 0.986609 |
| eucalyptus | **0.933782** | 0.925856 |
| first-order-theorem-proving | **0.835425** | 0.825561 |
| GesturePhaseSegmentationProcessed | **0.936667** | 0.917835 |
| har | **0.999958** | 0.999942 |
| ilpd | **0.765098** | 0.745564 |
| Internet-Advertisements | **0.985963** | 0.984740 |
| isolet | **0.999744** | 0.999696 |
| jm1 | **0.770272** | 0.761065 |
| jungle_chess_2pcs_raw_endgame_complete | 0.999278 | **0.999444** |
| kc1 | **0.835974** | 0.815660 |
| kc2 | **0.834913** | 0.813625 |
| kr-vs-kp | 0.999405 | **0.999412** |
| letter | **0.999934** | 0.999933 |
| madelon | **0.932817** | 0.929882 |
| mfeat-factors | **0.999350** | 0.999111 |
| mfeat-fourier | 0.986058 | **0.986717** |
| mfeat-karhunen | **0.999575** | 0.998740 |
| mfeat-morphological | **0.977508** | 0.968908 |
| mfeat-pixel | **0.999403** | 0.999139 |
| mfeat-zernike | **0.995249** | 0.985279 |
| MiceProtein | 0.999929 | **0.999981** |
| nomao | **0.996892** | 0.996441 |
| numerai28.6 | **0.530150** | 0.527692 |
| optdigits | **0.999925** | 0.999893 |
| ozone-level-8hr | **0.936029** | 0.930880 |
| pc1 | **0.888177** | 0.860825 |
| pc3 | **0.865766** | 0.845648 |
| pc4 | **0.955384** | 0.950117 |
| pendigits | **0.999725** | 0.999642 |
| PhishingWebsites | **0.997572** | 0.997102 |
| phoneme | **0.973342** | 0.964555 |
| qsar-biodeg | **0.942988** | 0.932276 |
| satimage | **0.993557** | 0.993220 |
| segment | **0.996895** | 0.996421 |
| semeion | **0.998506** | 0.998210 |
| sick | **0.998367** | 0.997357 |
| spambase | **0.991092** | 0.989781 |
| splice | **0.995941** | 0.995249 |
| steel-plates-fault | **0.973843** | 0.972323 |
| texture | **0.999998** | 0.999995 |
| tic-tac-toe | **1.000000** | 0.996585 |
| vehicle | **0.969797** | 0.965886 |
| vowel | **0.999910** | 0.999618 |
| wall-robot-navigation | **0.999993** | 0.999984 |
| wdbc | **0.995799** | 0.992456 |
| wilt | **0.995652** | 0.994495 |

## B.2 Results using default hyperparameter configurations

Table 18 shows the raw results for CatBoost and XGBoost using the default hyperparameter configurations.

Table 18: Average test ROC-AUC per dataset for XGBoost and CatBoost using the default hyperparamater configurations across CV folds.

| Dataset | CatBoost | XGBoost |
|---|---|---|
| adult | **0.930571** | 0.929316 |
| analcatdata_authorship | **0.999710** | 0.999518 |
| analcatdata_dmft | **0.549171** | 0.531850 |
| balance-scale | **0.952530** | 0.926923 |
| bank-marketing | **0.938725** | 0.934864 |
| banknote-authentication | **0.999957** | 0.999914 |
| Bioresponse | 0.879217 | **0.880176** |
| blood-transfusion-service-center | **0.729842** | 0.712258 |
| breast-w | **0.991254** | 0.990430 |
| car | **0.999509** | 0.998790 |
| churn | **0.924606** | 0.913882 |
| climate-model-simulation-crashes | **0.962296** | 0.955828 |
| cmc | **0.709590** | 0.684939 |
| cnae-9 | **0.996007** | 0.994232 |
| connect-4 | 0.893587 | **0.899588** |
| credit-approval | **0.937424** | 0.930615 |
| credit-g | **0.800667** | 0.788381 |
| cylinder-bands | 0.885160 | **0.912564** |
| diabetes | **0.835137** | 0.797009 |
| dna | 0.994641 | **0.994699** |
| dresses-sales | **0.598768** | 0.570699 |
| electricity | 0.958153 | **0.971787** |
| eucalyptus | **0.921691** | 0.902805 |
| first-order-theorem-proving | 0.826532 | **0.826895** |
| GesturePhaseSegmentationProcessed | **0.898407** | 0.892459 |
| har | 0.999899 | **0.999905** |
| ilpd | **0.741153** | 0.722052 |
| Internet-Advertisements | **0.979992** | 0.976972 |
| isolet | **0.999407** | 0.998854 |
| jm1 | **0.748060** | 0.729353 |
| jungle_chess_2pcs_raw_endgame_complete | 0.972286 | **0.974856** |
| kc1 | **0.823661** | 0.791182 |
| kc2 | **0.821163** | 0.771390 |
| kr-vs-kp | 0.999521 | **0.999720** |
| letter | **0.999740** | 0.999648 |
| madelon | **0.928172** | 0.890107 |
| mfeat-factors | **0.999031** | 0.998356 |
| mfeat-fourier | **0.984181** | 0.982669 |
| mfeat-karhunen | **0.999128** | 0.997700 |
| mfeat-morphological | **0.962489** | 0.958908 |
| mfeat-pixel | **0.999289** | 0.998703 |
| mfeat-zernike | **0.972961** | 0.966633 |
| MiceProtein | **0.999983** | 0.999680 |
| nomao | 0.995620 | **0.995690** |
| numerai28.6 | **0.518341** | 0.511976 |
| optdigits | **0.999808** | 0.999586 |
| ozone-level-8hr | **0.925485** | 0.911594 |
| pc1 | **0.891257** | 0.857895 |
| pc3 | **0.850219** | 0.816916 |
| pc4 | **0.953689** | 0.942808 |
| pendigits | **0.999764** | 0.999760 |
| PhishingWebsites | 0.995801 | **0.996764** |
| phoneme | 0.955202 | **0.957311** |
| qsar-biodeg | **0.934769** | 0.926970 |
| satimage | **0.991815** | 0.990907 |
| segment | **0.996012** | 0.995267 |
| semeion | **0.998163** | 0.996029 |
| sick | **0.998355** | 0.996943 |
| spambase | **0.989066** | 0.988888 |
| splice | **0.995198** | 0.994788 |
| steel-plates-fault | **0.972233** | 0.970148 |
| texture | **0.999908** | 0.999795 |
| tic-tac-toe | **1.000000** | 0.999181 |
| vehicle | **0.942832** | 0.935079 |
| vowel | **0.999237** | 0.996947 |
| wall-robot-navigation | **0.999989** | 0.999934 |
| wdbc | 0.994217 | **0.994471** |
| wilt | **0.991488** | 0.988659 |

Table 19 shows the raw results for dataset-specific neural networks using the default hyperparameter configurations.

Table 19: Average test ROC-AUC per dataset for dataset-specific neural networks using default hyperparameter configurations across CV folds. Missing datasets are represented by "-".

| Dataset | FT-Transformer | ResNet | SAINT | TabNet |
|---|---|---|---|---|
| adult | 0.893029 | 0.905838 | 0.870099 | **0.912781** |
| analcatdata_authorship | 0.999392 | **1.000000** | 0.999983 | 0.993186 |
| analcatdata_dmft | **0.553755** | 0.553675 | 0.526597 | 0.534271 |
| balance-scale | 0.988863 | **0.992229** | 0.991970 | 0.972816 |
| bank-marketing | 0.907667 | 0.926617 | 0.892316 | **0.927765** |
| banknote-authentication | **1.000000** | **1.000000** | **1.000000** | **1.000000** |
| Bioresponse | 0.804580 | **0.843462** | - | 0.812061 |
| blood-transfusion-service-center | 0.713181 | **0.742088** | 0.723673 | 0.728919 |
| breast-w | 0.988615 | 0.991140 | **0.992220** | 0.984383 |
| car | 0.999758 | 0.998600 | **0.999828** | 0.931659 |
| churn | **0.915966** | 0.914732 | 0.910996 | 0.905642 |
| climate-model-simulation-crashes | 0.840724 | 0.904025 | **0.937306** | 0.825571 |
| cmc | 0.686016 | 0.687757 | 0.642394 | **0.689043** |
| cnae-9 | 0.994801 | **0.996595** | - | 0.912423 |
| connect-4 | 0.922969 | **0.926041** | 0.756318 | 0.856762 |
| credit-approval | 0.915482 | **0.916769** | 0.908623 | 0.875614 |
| credit-g | 0.731714 | 0.735071 | **0.744000** | 0.632571 |
| cylinder-bands | 0.908565 | 0.891759 | **0.909314** | 0.710240 |
| diabetes | 0.755846 | **0.789923** | 0.737127 | 0.785077 |
| dna | 0.988362 | **0.992218** | 0.520670 | 0.962713 |
| dresses-sales | **0.571921** | 0.536617 | 0.568144 | 0.560591 |
| electricity | **0.963347** | 0.930924 | 0.960991 | 0.911419 |
| eucalyptus | **0.917340** | 0.897582 | 0.904708 | 0.877684 |
| first-order-theorem-proving | **0.796282** | 0.793079 | 0.772449 | 0.743350 |
| GesturePhaseSegmentationProcessed | 0.827939 | 0.853272 | **0.893255** | 0.781506 |
| har | **0.999876** | 0.999859 | - | 0.999147 |
| ilpd | 0.724591 | **0.758030** | 0.713191 | 0.715948 |
| Internet-Advertisements | **0.973465** | 0.967077 | - | 0.892480 |
| isolet | **0.999463** | 0.999307 | - | 0.997706 |
| jm1 | 0.723314 | **0.734238** | 0.652524 | 0.722615 |
| jungle_chess_2pcs_raw_endgame_complete | 0.998738 | 0.977410 | **0.999876** | 0.974173 |
| kc1 | **0.804719** | 0.795200 | 0.742990 | 0.792858 |
| kc2 | 0.805644 | 0.771497 | 0.742400 | **0.806986** |
| kr-vs-kp | **0.999792** | 0.999476 | 0.723052 | 0.987183 |
| letter | 0.999825 | **0.999864** | 0.999784 | 0.997271 |
| madelon | **0.770769** | 0.600713 | - | 0.559015 |
| mfeat-factors | 0.998765 | **0.998892** | 0.499849 | 0.993717 |
| mfeat-fourier | 0.977475 | **0.980419** | 0.971772 | 0.961111 |
| mfeat-karhunen | 0.997503 | 0.998097 | **0.998387** | 0.982592 |
| mfeat-morphological | 0.967733 | **0.969308** | 0.967478 | 0.963611 |
| mfeat-pixel | 0.997658 | **0.998676** | 0.553414 | 0.992500 |
| mfeat-zernike | 0.978039 | **0.980858** | 0.969257 | 0.966992 |
| MiceProtein | **1.000000** | 0.999963 | **1.000000** | 0.987043 |
| nomao | 0.992049 | **0.992530** | 0.499521 | 0.991441 |
| numerai28.6 | 0.507813 | 0.517071 | 0.507780 | **0.522797** |
| optdigits | 0.999631 | **0.999837** | 0.999057 | 0.998476 |
| ozone-level-8hr | **0.893747** | 0.826296 | 0.881560 | 0.869228 |
| pc1 | 0.852119 | 0.820008 | **0.866325** | 0.863233 |
| pc3 | **0.810311** | 0.771759 | 0.804479 | 0.809443 |
| pc4 | **0.944764** | 0.936765 | 0.931286 | 0.900752 |
| pendigits | 0.999740 | 0.999691 | **0.999785** | 0.999088 |
| PhishingWebsites | 0.996882 | **0.997134** | 0.996805 | 0.993856 |
| phoneme | 0.956543 | 0.938565 | **0.956949** | 0.933545 |
| qsar-biodeg | 0.916158 | 0.916804 | **0.918103** | 0.893489 |
| satimage | **0.992141** | 0.990613 | 0.985874 | 0.986280 |
| segment | **0.994709** | 0.993821 | 0.993989 | 0.992101 |
| semeion | 0.995507 | **0.996745** | 0.576269 | 0.957550 |
| sick | **0.997877** | 0.969015 | 0.991121 | 0.929353 |
| spambase | 0.983325 | **0.985056** | 0.981111 | 0.978240 |
| splice | 0.989898 | 0.990917 | **0.991932** | 0.972882 |
| steel-plates-fault | **0.959626** | 0.959356 | 0.948021 | 0.916561 |
| texture | 0.999976 | **0.999999** | 0.996944 | 0.999441 |
| tic-tac-toe | 0.998605 | **0.999375** | 0.996921 | 0.899715 |
| vehicle | 0.956404 | **0.963268** | 0.944376 | 0.923325 |
| vowel | 0.999618 | **0.999966** | 0.999888 | 0.986644 |
| wall-robot-navigation | **0.999757** | 0.998972 | 0.999104 | 0.997972 |
| wdbc | 0.994847 | 0.997080 | **0.997234** | 0.985323 |
| wilt | **0.994235** | 0.994057 | 0.988766 | 0.991840 |

Table 20 shows the raw results for MLP and RealMLP using the default hyperparameter configurations.

Table 20: Average test ROC-AUC per dataset for MLP and RealMLP using the default hyperparamater configurations across CV folds.

| Dataset | MLP | RealMLP |
|---|---|---|
| adult | 0.897504 | **0.908495** |
| analcatdata_authorship | 0.999934 | **0.999952** |
| analcatdata_dmft | 0.554240 | **0.575077** |
| balance-scale | **0.995111** | 0.980107 |
| bank-marketing | **0.904699** | 0.816579 |
| banknote-authentication | **1.000000** | **1.000000** |
| Bioresponse | 0.560952 | **0.824996** |
| blood-transfusion-service-center | **0.762080** | 0.746119 |
| breast-w | **0.994222** | 0.993411 |
| car | 0.999678 | **1.000000** |
| churn | 0.903166 | **0.917012** |
| climate-model-simulation-crashes | 0.935571 | **0.947857** |
| cmc | **0.710134** | 0.700557 |
| cnae-9 | 0.500463 | **0.992911** |
| connect-4 | **0.915051** | 0.909829 |
| credit-approval | **0.931215** | 0.914531 |
| credit-g | 0.726875 | **0.758000** |
| cylinder-bands | 0.874205 | **0.906434** |
| diabetes | **0.829853** | 0.822211 |
| dna | **0.990128** | 0.988320 |
| dresses-sales | **0.536782** | 0.526601 |
| electricity | **0.950665** | 0.950555 |
| eucalyptus | **0.922173** | 0.903885 |
| first-order-theorem-proving | **0.782461** | 0.781809 |
| GesturePhaseSegmentationProcessed | 0.819054 | **0.890444** |
| har | **0.999848** | 0.999630 |
| ilpd | **0.748217** | 0.727899 |
| Internet-Advertisements | **0.982883** | 0.961953 |
| isolet | 0.847095 | **0.999135** |
| jm1 | **0.726646** | 0.721977 |
| jungle_chess_2pcs_raw_endgame_complete | **0.998486** | 0.996257 |
| kc1 | 0.801565 | **0.806604** |
| kc2 | **0.840419** | 0.829826 |
| kr-vs-kp | **0.999765** | 0.998737 |
| letter | 0.999640 | **0.999820** |
| madelon | 0.500000 | **0.915592** |
| mfeat-factors | 0.998668 | **0.999075** |
| mfeat-fourier | **0.978653** | 0.974028 |
| mfeat-karhunen | 0.998582 | **0.999439** |
| mfeat-morphological | 0.965494 | **0.968706** |
| mfeat-pixel | 0.946632 | **0.999500** |
| mfeat-zernike | **0.980681** | 0.965872 |
| MiceProtein | 0.999963 | **1.000000** |
| nomao | **0.991436** | 0.983015 |
| numerai28.6 | 0.513601 | **0.522412** |
| optdigits | 0.999454 | **0.999927** |
| ozone-level-8hr | **0.906572** | 0.822254 |
| pc1 | **0.853077** | 0.828996 |
| pc3 | **0.784672** | 0.768438 |
| pc4 | **0.940799** | 0.906347 |
| pendigits | 0.999687 | **0.999850** |
| PhishingWebsites | **0.996479** | 0.994417 |
| phoneme | 0.948168 | **0.952913** |
| qsar-biodeg | **0.924529** | 0.911174 |
| satimage | **0.990975** | 0.986944 |
| segment | 0.993795 | **0.994189** |
| semeion | 0.968306 | **0.998289** |
| sick | **0.989590** | 0.976784 |
| spambase | **0.983168** | 0.978382 |
| splice | 0.990919 | **0.991318** |
| steel-plates-fault | **0.963250** | 0.955133 |
| texture | 0.999956 | **0.999992** |
| tic-tac-toe | **0.999145** | 0.997548 |
| vehicle | 0.944588 | **0.961117** |
| vowel | 0.997520 | **0.999641** |
| wall-robot-navigation | **0.999245** | 0.998582 |
| wdbc | 0.994219 | **0.998021** |
| wilt | **0.994105** | 0.993080 |

Table 21 shows the raw results for the meta-learned neural networks using the default hyperparameter configurations.

Table 21: Average test ROC-AUC per dataset for meta-learned neural networks using default hyperparameter configurations across CV folds. Missing datasets are represented by "-".

| Dataset | CARTE | TPBerta | TabPFN | XTab |
|---|---|---|---|---|
| adult | **0.897259** | - | - | - |
| analcatdata_authorship | 0.998103 | - | **1.000000** | 0.997620 |
| analcatdata_dmft | 0.572113 | - | **0.586630** | 0.550627 |
| balance-scale | **0.998116** | - | 0.997656 | 0.895083 |
| bank-marketing | **0.907972** | - | - | - |
| banknote-authentication | **1.000000** | 0.997535 | - | 0.996615 |
| blood-transfusion-service-center | 0.705189 | 0.659754 | **0.752586** | - |
| breast-w | 0.984775 | 0.967673 | **0.994131** | 0.988527 |
| car | **0.992862** | - | - | - |
| churn | **0.920360** | - | - | - |
| climate-model-simulation-crashes | 0.938031 | - | **0.968010** | 0.568735 |
| cmc | **0.730370** | - | - | - |
| cnae-9 | **0.986921** | - | - | - |
| connect-4 | **0.500681** | - | - | - |
| credit-approval | 0.906552 | 0.891294 | **0.932397** | 0.922447 |
| credit-g | 0.700952 | - | **0.768476** | - |
| cylinder-bands | 0.810318 | 0.814857 | **0.886616** | 0.778646 |
| diabetes | 0.755348 | 0.768974 | **0.836120** | 0.822370 |
| dna | 0.981979 | - | - | **0.992857** |
| dresses-sales | **0.591297** | 0.565189 | 0.538916 | 0.585057 |
| electricity | 0.874950 | - | - | **0.900765** |
| eucalyptus | 0.907418 | - | **0.928493** | 0.814121 |
| first-order-theorem-proving | **0.735870** | - | - | 0.721997 |
| GesturePhaseSegmentationProcessed | **0.771707** | - | - | 0.737155 |
| har | - | - | - | **0.999241** |
| ilpd | 0.729851 | 0.672431 | **0.757892** | 0.724427 |
| isolet | 0.995113 | - | - | **0.998455** |
| jm1 | 0.704730 | - | - | **0.721445** |
| jungle_chess_2pcs_raw_endgame_complete | 0.918894 | - | - | **0.965961** |
| kc1 | **0.805108** | - | - | 0.793122 |
| kc2 | 0.826925 | - | **0.850065** | 0.835398 |
| kr-vs-kp | 0.958715 | **0.999107** | - | 0.995940 |
| letter | **0.998939** | - | - | 0.989493 |
| madelon | **0.789929** | - | - | 0.689657 |
| mfeat-factors | 0.794171 | - | - | **0.997867** |
| mfeat-fourier | **0.969911** | - | - | 0.956494 |
| mfeat-karhunen | 0.978967 | - | - | **0.990728** |
| mfeat-morphological | **0.961442** | - | - | 0.948069 |
| mfeat-pixel | 0.759099 | - | - | **0.997478** |
| mfeat-zernike | 0.964453 | - | - | **0.965907** |
| MiceProtein | **0.986177** | - | - | 0.972404 |
| nomao | 0.981817 | - | - | **0.991110** |
| numerai28.6 | 0.521094 | - | - | **0.527797** |
| optdigits | 0.998452 | - | - | **0.999031** |
| ozone-level-8hr | 0.861468 | - | - | **0.915294** |
| pc1 | **0.791339** | - | - | 0.729942 |
| pc3 | 0.784448 | 0.683751 | - | **0.816464** |
| pc4 | **0.907759** | 0.699487 | - | 0.888728 |
| pendigits | **0.999522** | - | - | 0.999222 |
| PhishingWebsites | **0.991886** | - | - | 0.987949 |
| phoneme | **0.932082** | 0.798855 | - | 0.911417 |
| qsar-biodeg | 0.914703 | 0.817997 | - | **0.919134** |
| satimage | 0.982299 | - | - | **0.982955** |
| segment | **0.992163** | - | - | 0.974072 |
| semeion | 0.983218 | - | - | **0.989977** |
| sick | **0.991907** | - | - | 0.950283 |
| spambase | 0.748573 | - | - | **0.982966** |
| splice | 0.701980 | - | - | **0.991116** |
| steel-plates-fault | **0.925718** | - | - | 0.848468 |
| texture | 0.993709 | - | - | **0.999521** |
| tic-tac-toe | 0.861176 | 0.958328 | **0.996086** | 0.744202 |
| vehicle | 0.929483 | - | **0.970556** | 0.893891 |
| vowel | **0.995589** | - | - | 0.812581 |
| wall-robot-navigation | **0.998981** | - | - | 0.986489 |
| wdbc | 0.993948 | - | **0.996298** | 0.984744 |
| wilt | **0.994112** | 0.960758 | - | 0.979966 |

Lastly, Table 22 shows the raw results of AutoGluon using the default settings.

Table 22: Average test ROC-AUC per dataset for AutoGluon using default configurations across CV folds.

| Dataset | AutoGluon |
|---|---|
| adult | **0.931179** |
| analcatdata_authorship | **0.999782** |
| analcatdata_dmft | **0.584732** |
| balance-scale | **0.594936** |
| bank-marketing | **0.939889** |
| banknote-authentication | **0.999957** |
| Bioresponse | **0.884276** |
| blood-transfusion-service-center | **0.741962** |
| breast-w | **0.992231** |
| car | **0.999593** |
| churn | **0.922201** |
| climate-model-simulation-crashes | **0.957745** |
| cmc | **0.691344** |
| cnae-9 | **0.997878** |
| connect-4 | **0.936000** |
| credit-approval | **0.935450** |
| credit-g | **0.783286** |
| cylinder-bands | **0.900459** |
| diabetes | **0.821997** |
| dna | **0.994904** |
| dresses-sales | **0.586043** |
| electricity | **0.987262** |
| eucalyptus | **0.754274** |
| first-order-theorem-proving | **0.830805** |
| GesturePhaseSegmentationProcessed | **0.920355** |
| har | **0.999938** |
| ilpd | **0.737184** |
| Internet-Advertisements | **0.984077** |
| isolet | **0.999636** |
| jm1 | **0.764863** |
| jungle_chess_2pcs_raw_endgame_complete | **0.992186** |
| kc1 | **0.821507** |
| kc2 | **0.812567** |
| kr-vs-kp | **0.999619** |
| letter | **0.999901** |
| madelon | **0.925627** |
| mfeat-factors | **0.999142** |
| mfeat-fourier | **0.984642** |
| mfeat-karhunen | **0.998693** |
| mfeat-morphological | **0.969200** |
| mfeat-pixel | **0.998731** |
| mfeat-zernike | **0.982779** |
| MiceProtein | **0.899990** |
| nomao | **0.996397** |
| numerai28.6 | **0.527789** |
| optdigits | **0.999670** |
| ozone-level-8hr | **0.927357** |
| pc1 | **0.876676** |
| pc3 | **0.849770** |
| pc4 | **0.952137** |
| pendigits | **0.999684** |
| PhishingWebsites | **0.997256** |
| phoneme | **0.966521** |
| qsar-biodeg | **0.931279** |
| satimage | **0.992096** |
| segment | **0.996333** |
| semeion | **0.998341** |
| sick | **0.997864** |
| spambase | **0.989571** |
| splice | **0.995584** |
| steel-plates-fault | **0.971070** |
| texture | **0.999996** |
| tic-tac-toe | **0.999951** |
| vehicle | **0.958256** |
| vowel | **0.999641** |
| wall-robot-navigation | **0.898793** |
| wdbc | **0.992978** |
| wilt | **0.994524** |

## C  Datasets

In Table 23 we show a summary of all the OpenMLCC18 datasets used in this study.

Table 23: Summary of OpenML-CC18 Datasets with Feature and Class Frequency Statistics.

| Dataset ID | Dataset Name | Number of Instances | Number of Features | Numerical Features | Categorical Features | Binary Features | Number of Classes | Min-Max Class Freq |
|---|---|---|---|---|---|---|---|---|
| 3 | kr-vs-kp | 3196 | 37 | 0 | 37 | 35 | 2 | 0.91 |
| 6 | letter | 20000 | 17 | 16 | 1 | 0 | 26 | 0.90 |
| 11 | balance-scale | 625 | 5 | 4 | 1 | 0 | 3 | 0.17 |
| 12 | mfeat-factors | 2000 | 217 | 216 | 1 | 0 | 10 | 1.00 |
| 14 | mfeat-fourier | 2000 | 77 | 76 | 1 | 0 | 10 | 1.00 |
| 15 | breast-w | 699 | 10 | 9 | 1 | 1 | 2 | 0.53 |
| 16 | mfeat-karhunen | 2000 | 65 | 64 | 1 | 0 | 10 | 1.00 |
| 18 | mfeat-morphological | 2000 | 7 | 6 | 1 | 0 | 10 | 1.00 |
| 22 | mfeat-zernike | 2000 | 48 | 47 | 1 | 0 | 10 | 1.00 |
| 23 | cmc | 1473 | 10 | 2 | 8 | 3 | 3 | 0.53 |
| 28 | optdigits | 5620 | 65 | 64 | 1 | 0 | 10 | 0.97 |
| 29 | credit-approval | 690 | 16 | 6 | 10 | 5 | 2 | 0.80 |
| 31 | credit-g | 1000 | 21 | 7 | 14 | 3 | 2 | 0.43 |
| 32 | pendigits | 10992 | 17 | 16 | 1 | 0 | 10 | 0.92 |
| 37 | diabetes | 768 | 9 | 8 | 1 | 1 | 2 | 0.54 |
| 38 | sick | 3772 | 30 | 7 | 23 | 21 | 2 | 0.07 |
| 44 | spambase | 4601 | 58 | 57 | 1 | 1 | 2 | 0.65 |
| 46 | splice | 3190 | 61 | 0 | 61 | 0 | 3 | 0.46 |
| 50 | tic-tac-toe | 958 | 10 | 0 | 10 | 1 | 2 | 0.53 |
| 54 | vehicle | 846 | 19 | 18 | 1 | 0 | 4 | 0.91 |
| 151 | electricity | 45312 | 9 | 7 | 2 | 1 | 2 | 0.74 |
| 182 | satimage | 6430 | 37 | 36 | 1 | 0 | 6 | 0.41 |
| 188 | eucalyptus | 736 | 20 | 14 | 6 | 0 | 5 | 0.49 |
| 300 | isolet | 7797 | 618 | 617 | 1 | 0 | 26 | 0.99 |
| 307 | vowel | 990 | 13 | 10 | 3 | 1 | 11 | 1.00 |
| 458 | analcatdata_authorship | 841 | 71 | 70 | 1 | 0 | 4 | 0.17 |
| 469 | analcatdata_dmft | 797 | 5 | 0 | 5 | 1 | 6 | 0.79 |
| 1049 | pc4 | 1458 | 38 | 37 | 1 | 1 | 2 | 0.14 |
| 1050 | pc3 | 1563 | 38 | 37 | 1 | 1 | 2 | 0.11 |
| 1053 | jm1 | 10885 | 22 | 21 | 1 | 1 | 2 | 0.24 |
| 1063 | kc2 | 522 | 22 | 21 | 1 | 1 | 2 | 0.26 |
| 1067 | kc1 | 2109 | 22 | 21 | 1 | 1 | 2 | 0.18 |
| 1068 | pc1 | 1109 | 22 | 21 | 1 | 1 | 2 | 0.07 |
| 1461 | bank-marketing | 45211 | 17 | 7 | 10 | 4 | 2 | 0.13 |
| 1462 | banknote-authentication | 1372 | 5 | 4 | 1 | 1 | 2 | 0.80 |
| 1464 | blood-transfusion-service-center | 748 | 5 | 4 | 1 | 1 | 2 | 0.31 |
| 1468 | cnae-9 | 1080 | 857 | 856 | 1 | 0 | 9 | 1.00 |
| 1475 | first-order-theorem-proving | 6118 | 52 | 51 | 1 | 0 | 6 | 0.19 |
| 1478 | har | 10299 | 562 | 561 | 1 | 0 | 6 | 0.72 |
| 1480 | ilpd | 583 | 11 | 9 | 2 | 2 | 2 | 0.40 |
| 1485 | madelon | 2600 | 501 | 500 | 1 | 1 | 2 | 1.00 |
| 1486 | nomao | 34465 | 119 | 89 | 30 | 3 | 2 | 0.40 |
| 1487 | ozone-level-8hr | 2534 | 73 | 72 | 1 | 1 | 2 | 0.07 |
| 1489 | phoneme | 5404 | 6 | 5 | 1 | 1 | 2 | 0.42 |
| 1494 | qsar-biodeg | 1055 | 42 | 41 | 1 | 1 | 2 | 0.51 |
| 1497 | wall-robot-navigation | 5456 | 25 | 24 | 1 | 0 | 4 | 0.15 |
| 1501 | semeion | 1593 | 257 | 256 | 1 | 0 | 10 | 0.96 |
| 1510 | wdbc | 569 | 31 | 30 | 1 | 1 | 2 | 0.59 |
| 1590 | adult | 48842 | 15 | 6 | 9 | 2 | 2 | 0.31 |
| 4134 | Bioresponse | 3751 | 1777 | 1776 | 1 | 1 | 2 | 0.84 |
| 4534 | PhishingWebsites | 11055 | 31 | 0 | 31 | 23 | 2 | 0.80 |
| 4538 | GesturePhaseSegmentationProcessed | 9873 | 33 | 32 | 1 | 0 | 5 | 0.34 |
| 6332 | cylinder-bands | 540 | 40 | 18 | 22 | 4 | 2 | 0.73 |
| 23381 | dresses-sales | 500 | 13 | 1 | 12 | 1 | 2 | 0.72 |
| 23517 | numerai28.6 | 96320 | 22 | 21 | 1 | 1 | 2 | 0.98 |
| 40499 | texture | 5500 | 41 | 40 | 1 | 0 | 11 | 1.00 |
| 40668 | connect-4 | 67557 | 43 | 0 | 43 | 0 | 3 | 0.15 |
| 40670 | dna | 3186 | 181 | 0 | 181 | 180 | 3 | 0.46 |
| 40701 | churn | 5000 | 21 | 16 | 5 | 3 | 2 | 0.16 |
| 40966 | MiceProtein | 1080 | 82 | 77 | 5 | 3 | 8 | 0.70 |
| 40975 | car | 1728 | 7 | 0 | 7 | 0 | 4 | 0.05 |
| 40978 | Internet-Advertisements | 3279 | 1559 | 3 | 1556 | 1556 | 2 | 0.16 |
| 40979 | mfeat-pixel | 2000 | 241 | 240 | 1 | 0 | 10 | 1.00 |
| 40982 | steel-plates-fault | 1941 | 28 | 27 | 1 | 0 | 7 | 0.08 |
| 40983 | wilt | 4839 | 6 | 5 | 1 | 1 | 2 | 0.06 |
| 40984 | segment | 2310 | 20 | 19 | 1 | 0 | 7 | 1.00 |
| 40994 | climate-model-simulation-crashes | 540 | 21 | 20 | 1 | 1 | 2 | 0.09 |
| 41027 | jungle_chess_2pcs_raw_endgame_complete | 44819 | 7 | 6 | 1 | 0 | 3 | 0.19 |

# D    Hyperparameter analysis

In this section, we analyze the impact of individual hyperparameters on the performance metric. The x-axis represents the hyperparameters, while the y-axis denotes the ROC-AUC performance. These plots provide an overview of the performance landscape for each hyperparameter, illustrating their influence on model effectiveness.

## D.1    CatBoost

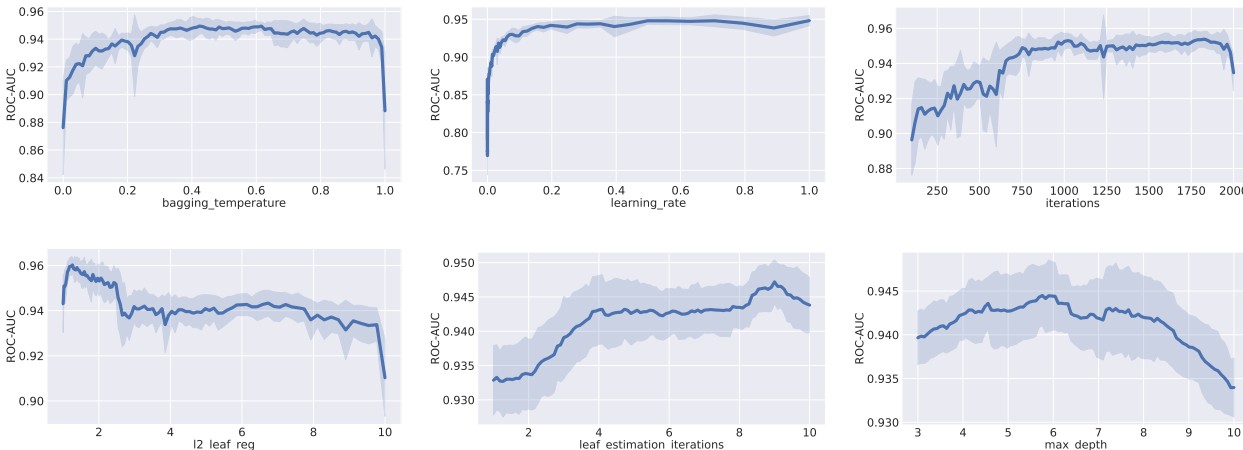

Figure 16: Effect of all the hyperparameters on model performance for CatBoost. The x-axis represents the hyperparameter values, while the y-axis shows the corresponding performance.

## D.2 ResNet

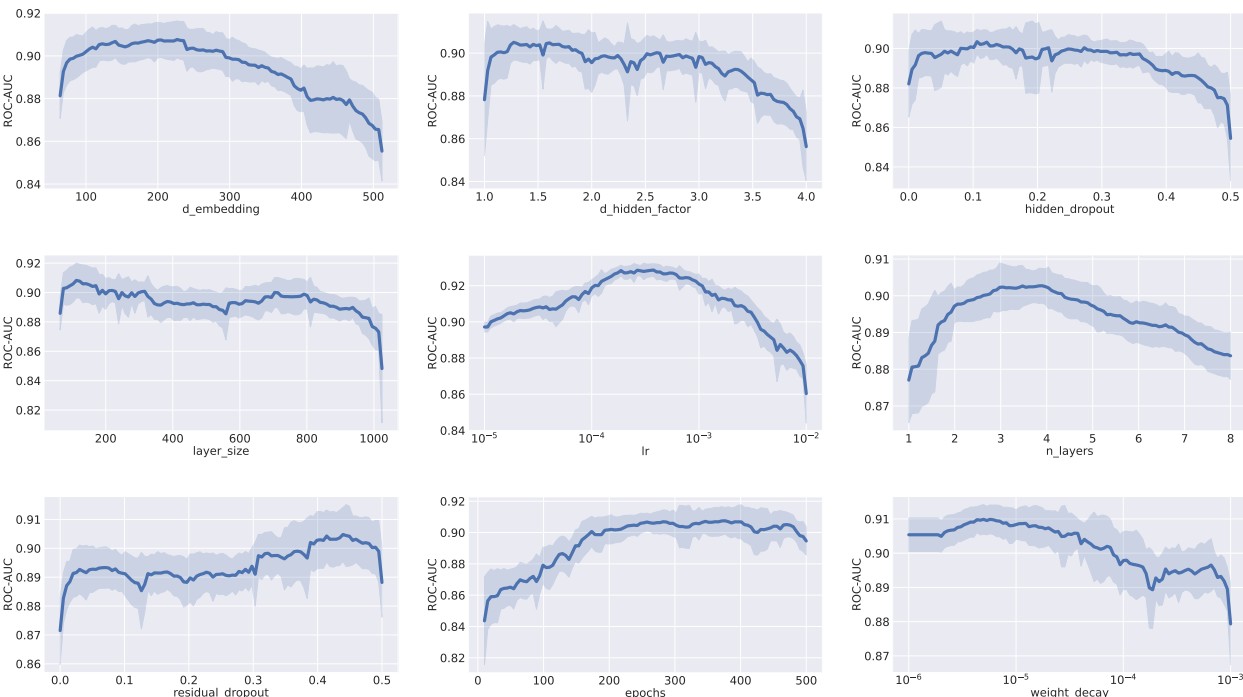

Figure 17: Effect of all the hyperparameters on model performance for ResNet. The x-axis represents the hyperparameter values, while the y-axis shows the corresponding performance.

### D.3 MLP-PLR

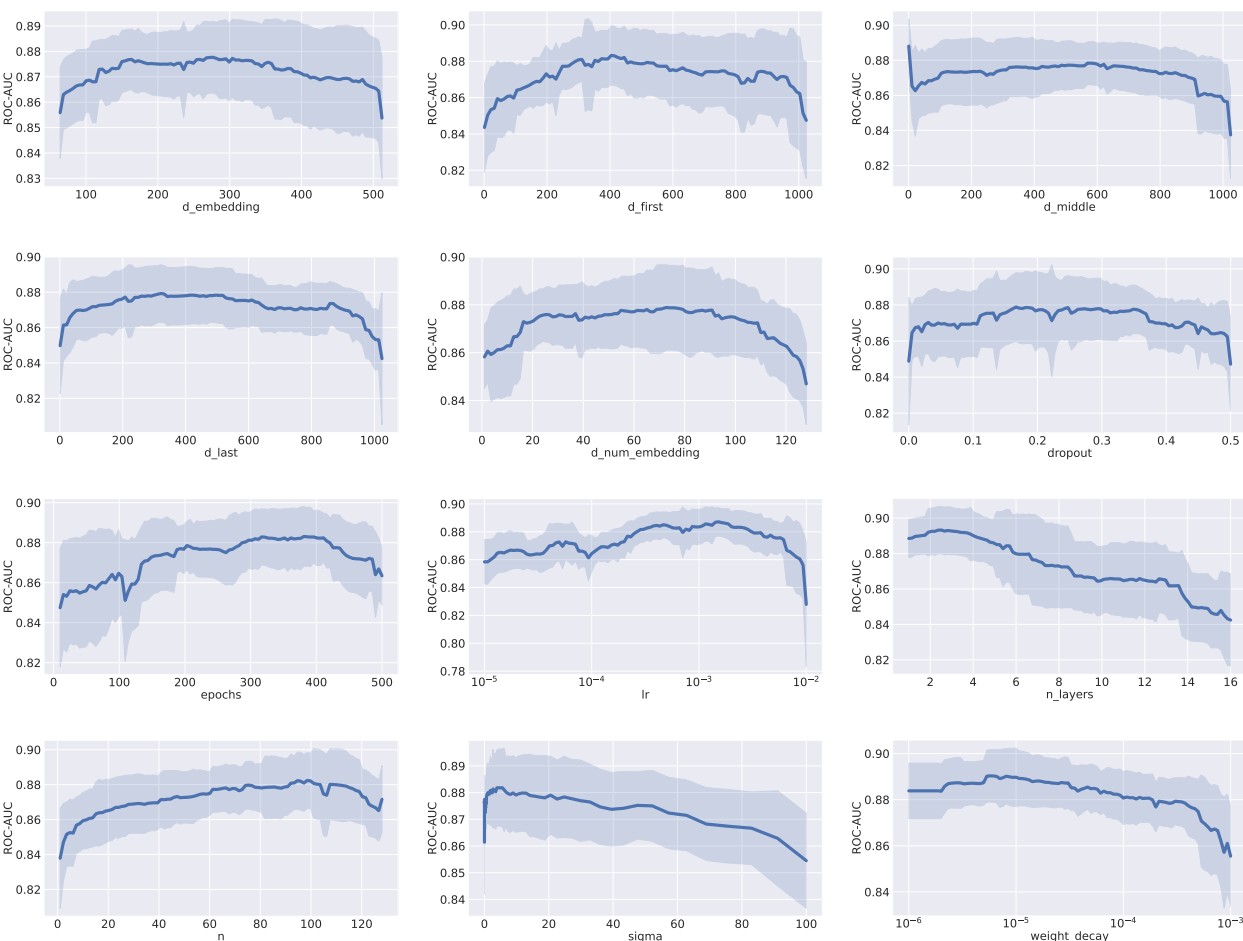

Figure 18: Effect of all the hyperparameters on model performance for MLP-PLR. The x-axis represents the hyperparameter values, while the y-axis shows the corresponding performance.

## D.4   RealMLP

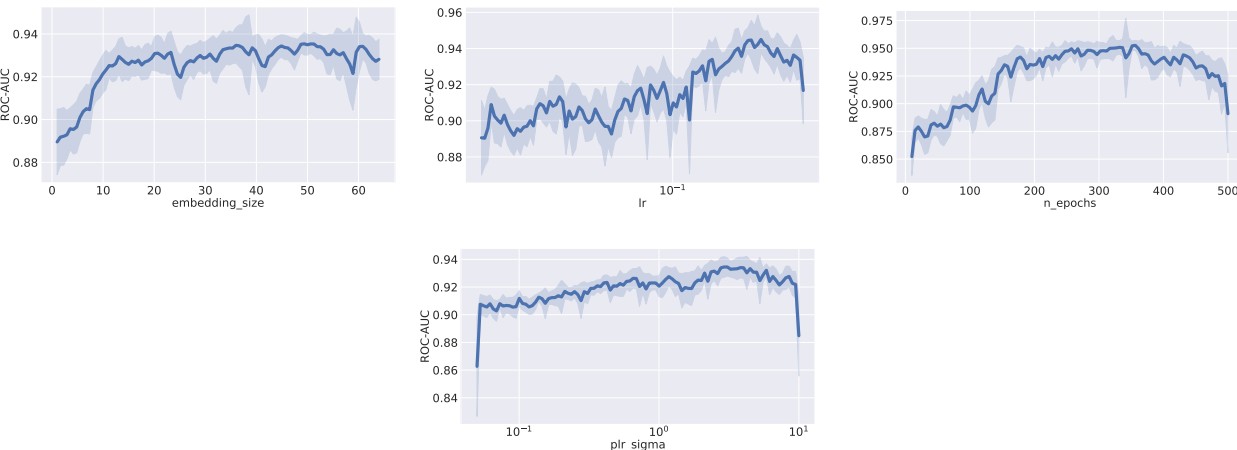

Figure 19: Effect of all the hyperparameters on model performance for RealMLP. The x-axis represents the hyperparameter values, while the y-axis shows the corresponding performance.

Since fANOVA does not support categorical hyperparameters, we exclude them from this analysis.

## D.5 XGBoost

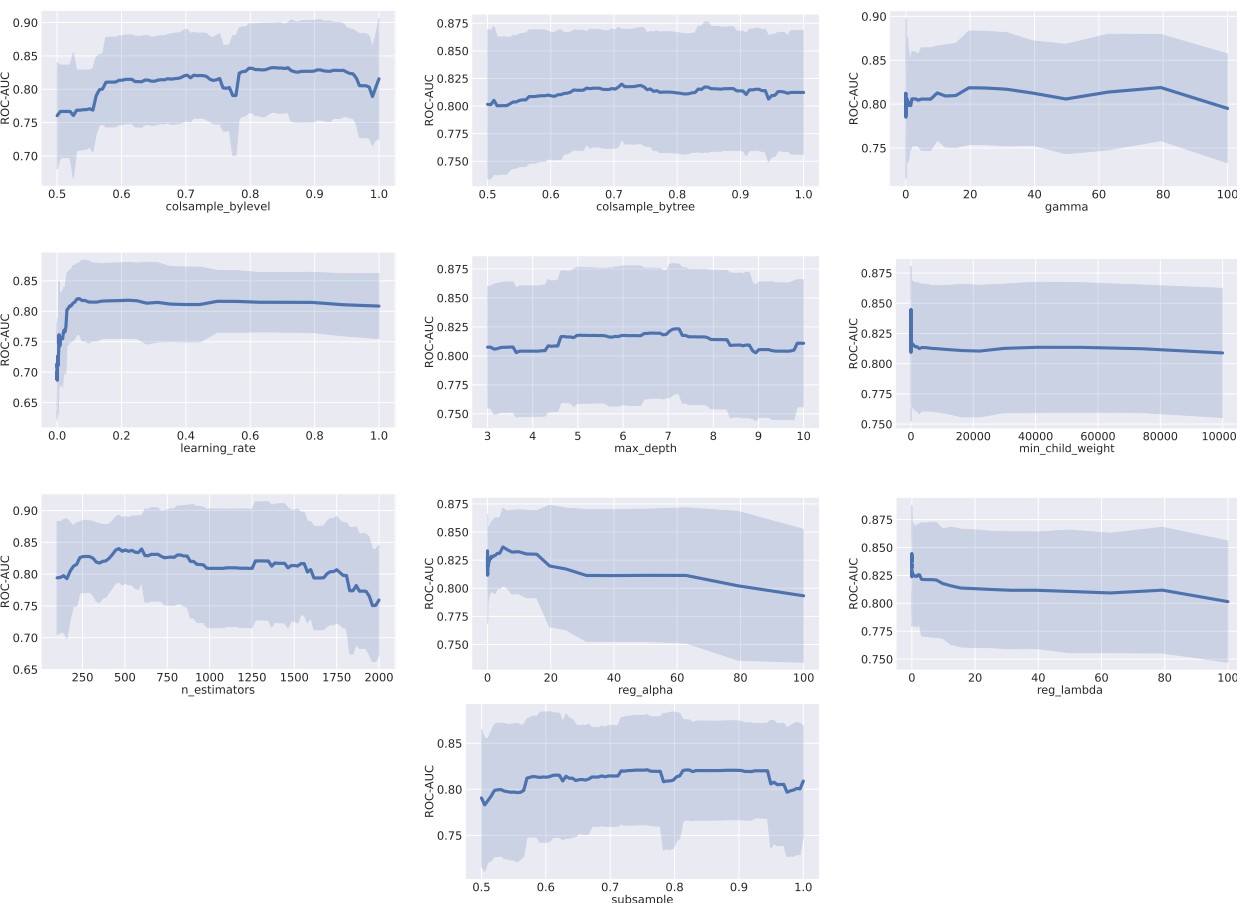

Figure 20: Effect of all the hyperparameters on model performance for XGBoost. The x-axis represents the hyperparameter values, while the y-axis shows the corresponding performance.

## D.6 FT-Transformer

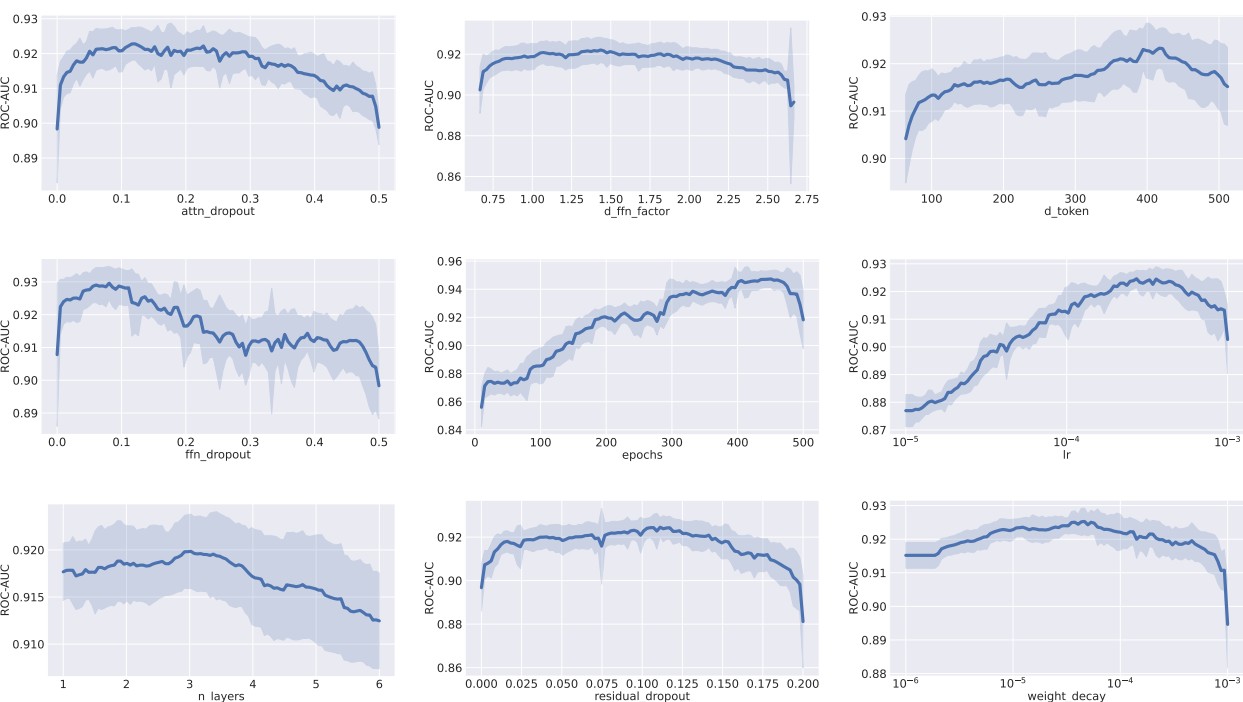

Figure 21: Effect of all the hyperparameters on model performance for FT-Transformer. The x-axis represents the hyperparameter values, while the y-axis shows the corresponding performance.

## D.7 SAINT

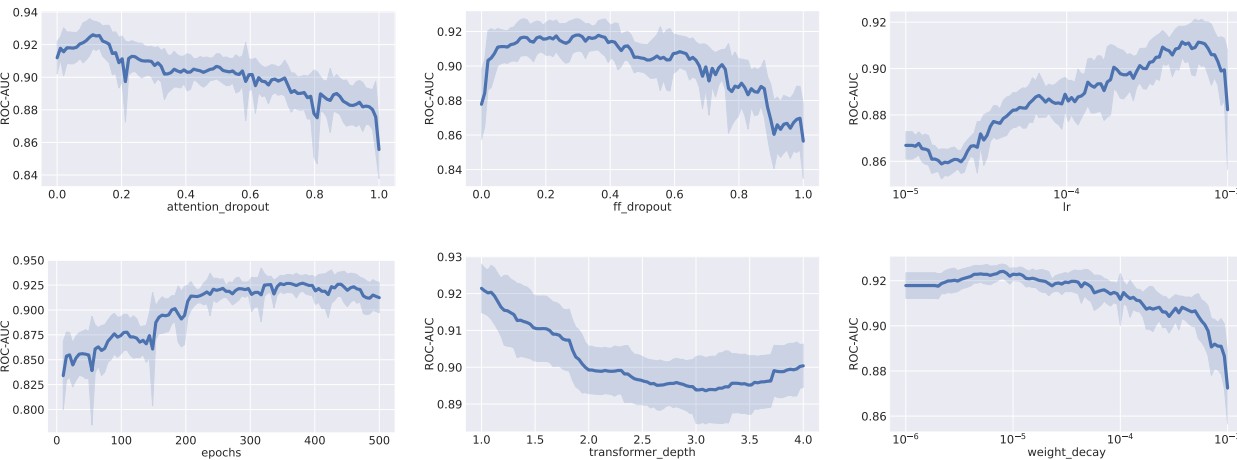

Figure 22: Effect of all the hyperparameters on model performance for SAINT. The x-axis represents the hyperparameter values, while the y-axis shows the corresponding performance.

## D.8  TabNet

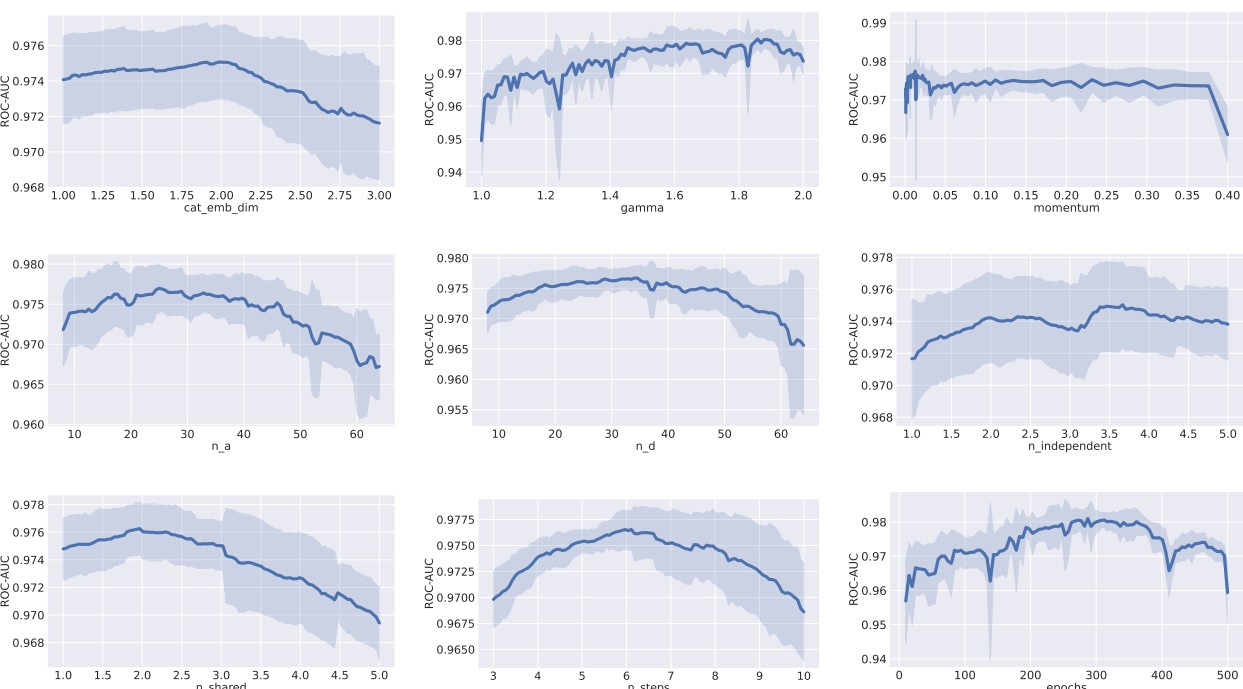

Figure 23: Effect of all the hyperparameters on model performance for TabNet. The x-axis represents the hyperparameter values, while the y-axis shows the corresponding performance.

## D.9  XTab

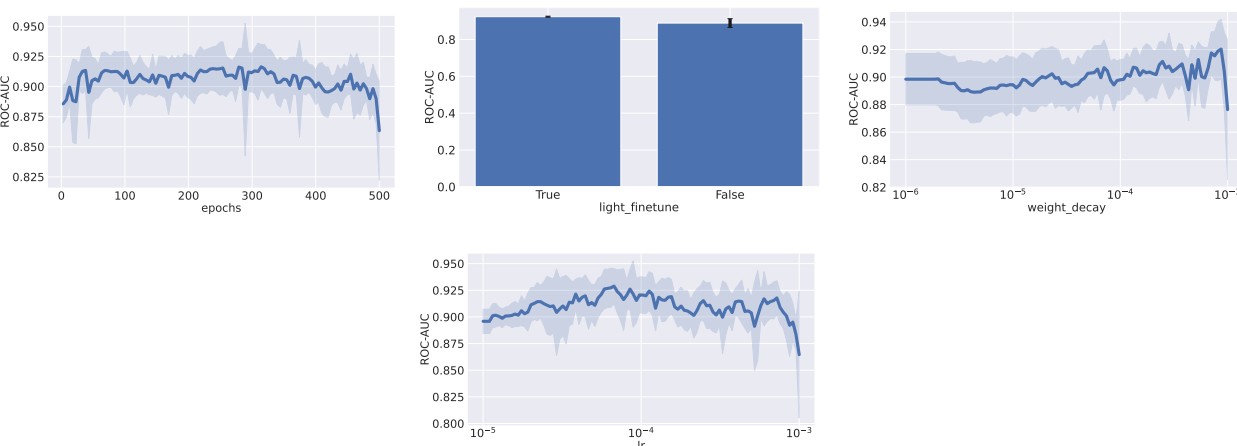

Figure 24: Effect of all the hyperparameters on model performance for XTab. The x-axis represents the hyperparameter values, while the y-axis shows the corresponding performance.

### D.10 CARTE

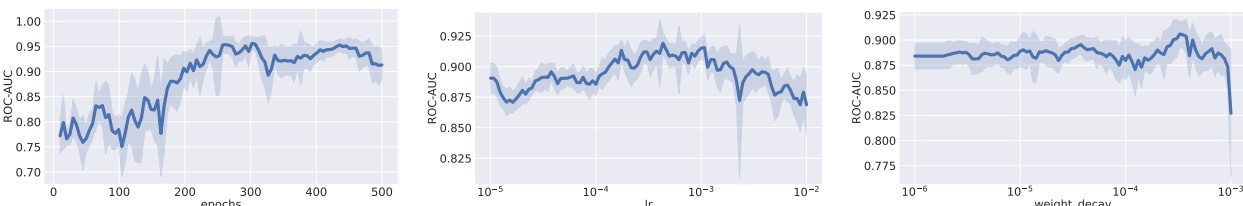

Figure 25: Effect of all the hyperparameters on model performance for CARTE. The x-axis represents the hyperparameter values, while the y-axis shows the corresponding performance.

### D.11 TP-BERTa

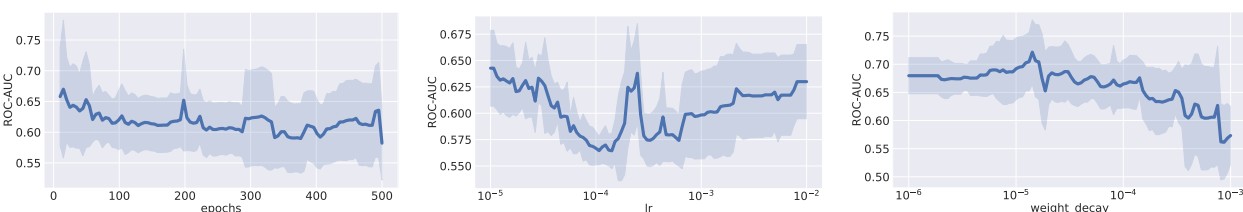

Figure 26: Effect of all the hyperparameters on model performance for TP-BERTa. The x-axis represents the hyperparameter values, while the y-axis shows the corresponding performance.

## E  Analysis of Dataset Characteristics: Instances and Features

To analyze the relationship between dataset size and the performance of different methods, we categorize datasets based on two key attributes: the number of instances and the number of features.

- **Instance-based Categorization:**
    - Datasets with 1000 or fewer instances.
    - Datasets with 1001 to 5000 instances.
    - Datasets with 5001 to 10000 instances.
    - Datasets with 10001 to 50000 instances.
    - Datasets with more than 50000 instances.

- **Feature-based Categorization:** Within each instance-based group, datasets are further divided based on the number of features:
    - Datasets with 100 or fewer features.
    - Datasets with 101 to 500 features.
    - Datasets with 501 to 1000 features.
    - Datasets with more than 1000 features.

- **Unavailable Results:** Having split the datasets into these groups, we note the ones in which no dataset belongs:
    - Datasets with instances between 5001 and 10000, and features between 100 and 500.
    - Datasets with instances between 5001 and 10000, and features greater than 1000.
    - Datasets with instances between 10001 and 50000, and features greater than 1000.

    – For datasets with more than 50000 instances, we only have results for datasets with 100 or fewer features.

    – For datasets with fewer than 1000 instances, we only have results for datasets with 100 or fewer features.

For the analysis, we present boxplots and critical difference diagrams, if the number of datasets is at least 10 for meaningful analysis. If the number of datasets in a group is fewer than 10, we use tabular results instead of boxplots or critical difference diagrams.

### E.1  Datasets with fewer than 1000 instances

In this section, we focus on datasets with fewer than 100 features and fewer than 1000 instances, resulting in a total of 18 datasets used in our study. Consequently, most methods in Figure 27 are evaluated on 18 datasets. However, there are a few exceptions: TabPFN is incompatible with one dataset, "vowel," due to it containing more than 10 classes; XTab excludes 2 datasets that were part of its pretraining phase; and TP-Berta encounters memory limitations on 10 out of the 18 datasets, reducing its coverage.

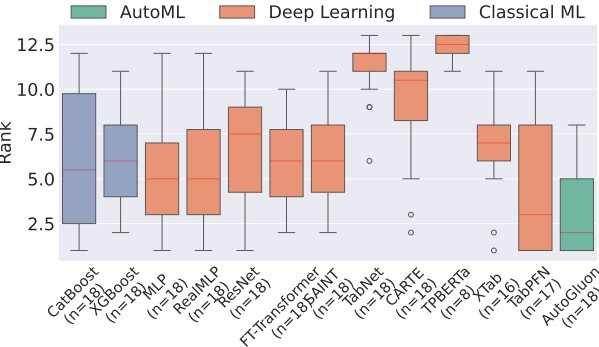

Figure 27: Distribution of ranks for all the methods in the small data domain. The boxplot illustrates the rank spread, with medians represented by red lines and whiskers showing the range.

Figure 27 reveals that AutoGluon achieves the strongest overall performance, closely followed by TabPFN. Among feedforward networks, MLP and RealMLP both rank well, though MLP shows a tighter (i.e. more robust) interquartile range. Among the other dataset-specific neural networks, FT-Transformer and SAINT perform comparably. Interestingly, MLP-like methods also show a lower median rank than the classical CatBoost and XGBoost, although CatBoost occasionally achieves ranks as low as 2.5. By contrast, TabNet and the fine-tuning–based models generally exhibit the weakest performance.

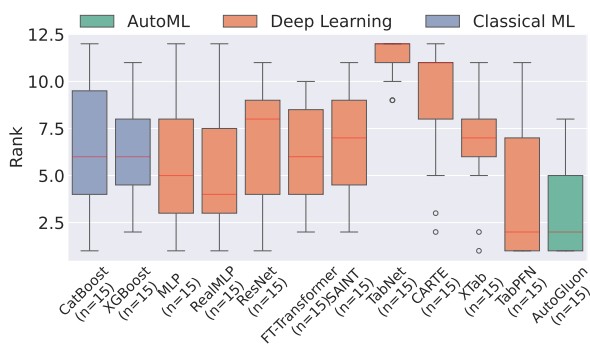
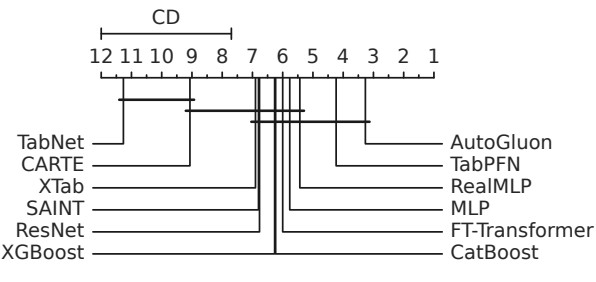

Figure 28: **Left:** Distribution of ranks for all the methods in the small data domain. The boxplot illustrates the rank spread, with medians represented by red lines and whiskers showing the range. **Right:** Comparative analysis of all the methods.

Similarly, Figure 28 shows a boxplot on the left—evaluated on the same datasets but excluding TP-BERTa—and a critical difference diagram on the right. A clear pattern emerges: in the small-data domain, dataset-specific neural architectures (e.g., MLP with PLR embeddings, RealMLP and FT-Transformer) display highly competitive performance, surpassing even CatBoost and XGBoost.

### E.2 Datasets with 1,000 to 5,000 instances

Following the previous analysis, we now focus on datasets with 1,000 to 5,000 instances and fewer than 100 features. Figure 29 illustrates the results for this subset of datasets. Similar to the small data domain, AutoGluon remains the top-performing method. However, an interesting shift occurs, CatBoost shows a significant improvement in performance, achieving the second-best overall rank, while XGBoost maintains a performance level similar to the smaller datasets. Additionally, dataset-specific neural networks continue to outperform meta-learned neural networks, with the MLP with PLR embeddings standing out due to its strong performance. It exhibits a better median rank and a narrower interquartile range compared to XGBoost.

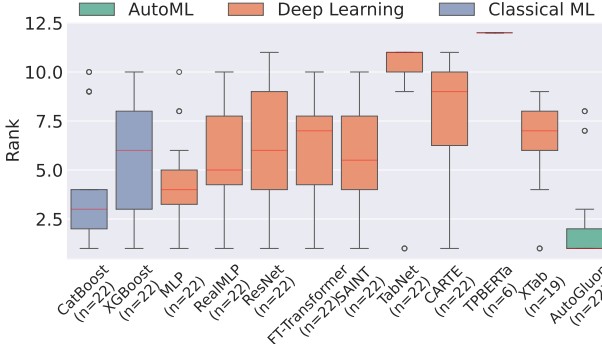

Figure 29: Distribution of ranks for all the methods in the datasets withh 1000 to 5000 instances, and less than 100 features. The boxplot illustrates the rank spread, with medians represented by red lines and whiskers showing the range.

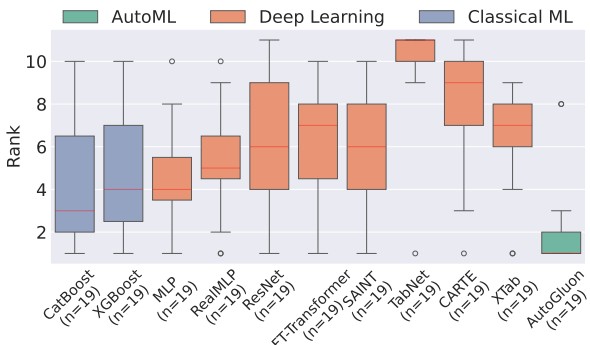 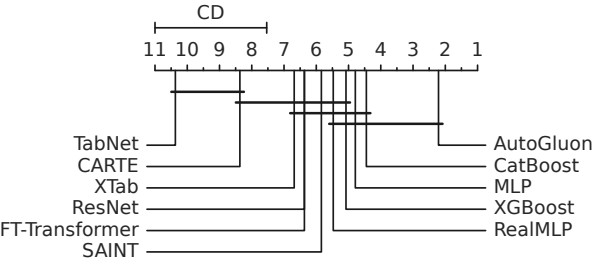

Figure 30: **Left:** Distribution of ranks for all the methods in the common datasets with instances between 1000 and 5000, and features fewer than 100. The boxplot illustrates the rank spread, with medians represented by red lines and whiskers showing the range. **Right:** Comparative analysis of all the methods.

In Figure 30, we exclude TP-BERTa again to ensure a reasonable number of common datasets, resulting in a total of 19 datasets. The left plot tells a similar story, with XGBoost now achieving the same median rank as the MLP with PLR embeddings. The right plot presents a critical difference diagram, showing AutoGluon, CatBoost, MLP, XGBoost and RealMLP as the top-performing methods. Among them, AutoGluon is statistically significantly better than all other methods, except for the aforementioned top performers.

For the remaining dataset categorization groups, we present only tabular results due to the limited number of datasets in these categories. Table 24 provides the results for datasets with 1000 to 5000 instances and 100 to 500 features. Similarly, Table 25 summarizes the performance for datasets in the 500 to 1000 features range, while Table 26 presents results for datasets with more than 1000 features. Detailed results for all other dataset categorization groups can be found below.

Table 24: Classifier Performance for Instance Range: 1000-5000 and Feature Range: 100-500

| Dataset | dna | mfeat-factors | mfeat-pixel | semeion |
|---|---|---|---|---|
| AutoGluon | **0.995385** | 0.999350 | 0.999403 | 0.998506 |
| CARTE | 0.986120 | 0.996064 | 0.996175 | 0.993378 |
| CatBoost | 0.995028 | 0.998910 | 0.999422 | 0.998687 |
| FT-Transformer | 0.990937 | 0.999015 | 0.997451 | 0.995548 |
| MLP | 0.992220 | 0.998875 | 0.998674 | 0.997350 |
| RealMLP | 0.994111 | **0.999625** | **0.999492** | **0.998976** |
| ResNet | 0.992543 | 0.999472 | 0.998690 | 0.997689 |
| SAINT | 0.992473 | 0.999385 | 0.999217 | 0.997630 |
| TabNet | 0.991448 | 0.998125 | 0.998200 | 0.994019 |
| XGBoost | 0.995278 | 0.999004 | 0.999378 | 0.998272 |
| XTab | 0.992479 | 0.998443 | 0.998642 | 0.997064 |

Table 25: Classifier Performance for Instance Range: 1000-5000 and Feature Range: 500-1000

| Dataset | cnae-9 | madelon |
|---|---|---|
| AutoGluon | **0.998524** | 0.932817 |
| CARTE | 0.990151 | 0.836760 |
| CatBoost | 0.996316 | **0.937562** |
| FT-Transformer | 0.994497 | 0.747391 |
| MLP | 0.996716 | 0.883991 |
| RealMLP | 0.997569 | 0.930302 |
| ResNet | 0.997106 | 0.605018 |
| TabNet | - | 0.630669 |
| XGBoost | 0.997454 | 0.932249 |
| XTab | - | 0.845746 |

Table 26: Classifier Performance for Instance Range: 1000-5000 and Feature Range: >1000

| Dataset | Bioresponse | Internet-Advertisements |
|---|---|---|
| AutoGluon | **0.888693** | **0.985963** |
| CatBoost | 0.885502 | 0.979120 |
| FT-Transformer | 0.820159 | 0.974513 |
| MLP | 0.825631 | - |
| RealMLP | 0.859065 | 0.973810 |
| ResNet | 0.850801 | 0.974187 |
| XGBoost | 0.888615 | 0.982276 |

### E.3 Datasets with 5,000 to 10,000 instances

Table 27: Classifier Performance for Instance Range: 5000-10000 and Feature Range: <=100

| Dataset | GPhaseSeg | first-ord-TP | optdigits | phoneme | satimage | texture | wall-rob-nav |
|---|---|---|---|---|---|---|---|
| AutoGluon | **0.936667** | **0.835425** | 0.999925 | **0.973342** | **0.993557** | 0.999998 | **0.999993** |
| CARTE | 0.798024 | 0.764092 | 0.999112 | 0.948702 | 0.988038 | 0.999541 | 0.999505 |
| CatBoost | 0.916674 | 0.831775 | 0.999844 | 0.968024 | 0.991978 | 0.999948 | 0.999990 |
| MLP | 0.911434 | 0.798811 | 0.999794 | 0.967617 | 0.992308 | 0.999990 | 0.999689 |
| RealMLP | 0.901441 | 0.795637 | **0.999968** | 0.966456 | 0.993034 | **0.999999** | 0.998720 |
| FT-Transformer | 0.895166 | 0.796707 | 0.999616 | 0.965071 | 0.993516 | 0.999983 | 0.999900 |
| XGBoost | 0.916761 | 0.834883 | 0.999855 | 0.967421 | 0.992114 | 0.999940 | 0.999981 |

Table 28: Classifier Performance for Instance Range: 5000-10000 and Feature Range: 500-1000

| Dataset | isolet |
|---|---|
| AutoGluon | **0.999744** |
| CatBoost | 0.999389 |
| FT-Transformer | 0.998817 |
| MLP | 0.998295 |
| RealMLP | 0.999634 |
| ResNet | 0.999401 |
| TabNet | 0.998813 |
| XGBoost | 0.999488 |

### E.4 Datasets with 10,000 to 50,000 instances

Table 29: Classifier Performance for Instance Range: 10000-50000 and Feature Range: <=100

| Dataset | Phishing | Adult | BankMkt | Elec | JM1 | JngChess | Letter | PenDigits |
|---|---|---|---|---|---|---|---|---|
| AutoGluon | **0.997572** | **0.931792** | **0.941273** | 0.987260 | **0.770272** | 0.999278 | **0.999934** | 0.999725 |
| CARTE | 0.994582 | 0.902677 | 0.924664 | 0.909407 | 0.728512 | 0.973383 | 0.999440 | 0.999468 |
| CatBoost | 0.996482 | 0.930747 | 0.938831 | 0.980993 | 0.756611 | 0.976349 | 0.999854 | 0.999752 |
| FT-Transformer | 0.996760 | 0.914869 | 0.938198 | 0.963076 | 0.709321 | **0.999975** | 0.999919 | 0.999703 |
| MLP | 0.996991 | 0.928689 | 0.937054 | 0.969201 | 0.715620 | 0.999965 | 0.999894 | 0.999705 |
| RealMLP | 0.997208 | 0.923327 | 0.937031 | 0.961467 | 0.713988 | 0.999774 | 0.999914 | 0.999659 |
| ResNet | 0.996975 | 0.913790 | 0.935740 | 0.960658 | 0.720444 | 0.999956 | 0.999926 | 0.999638 |
| SAINT | 0.996746 | 0.920246 | 0.936560 | 0.967012 | 0.719464 | 0.999926 | 0.999853 | **0.999782** |
| TabNet | 0.996196 | 0.882450 | 0.887319 | 0.938656 | 0.674043 | 0.991981 | 0.999606 | 0.999753 |
| XGBoost | 0.997425 | 0.930482 | 0.938384 | **0.987790** | 0.759652 | 0.974087 | 0.999819 | 0.999703 |
| XTab | 0.996896 | - | - | 0.966899 | 0.727984 | 0.999950 | 0.999859 | 0.999751 |

Table 30: Classifier Performance for Instance Range: 10000-50000 and Feature Range: 100-500

| Dataset | nomao |
|---|---|
| AutoGluon | **0.996892** |
| CatBoost | 0.996439 |
| FT-Transformer | 0.990908 |
| MLP | 0.986577 |
| RealMLP | 0.989803 |
| ResNet | 0.993048 |
| XGBoost | 0.996676 |
| XTab | 0.992727 |

Table 31: Classifier Performance for Instance Range: 10000-50000 and Feature Range: 500-1000

| Dataset | har |
|---|---|
| AutoGluon | 0.999958 |
| CatBoost | 0.999941 |
| FT-Transformer | 0.999685 |
| MLP | 0.999783 |
| RealMLP | 0.999958 |
| ResNet | 0.999921 |
| TabNet | 0.999515 |
| XGBoost | **0.999960** |

### E.5 Datasets with more than 50,000 instances

Table 32: Classifier Performance for Instance Range: >50000 and Feature Range: <=100

| Dataset | connect-4 | numerai28.6 |
|---|---|---|
| AutoGluon | **0.934636** | 0.530150 |
| CARTE | - | 0.514361 |
| CatBoost | 0.92105 | 0.529404 |
| FT-Transformer | 0.90117 | **0.530315** |
| MLP | 0.927373 | 0.525920 |
| RealMLP | 0.928258 | 0.529534 |
| ResNet | 0.933333 | 0.528012 |
| SAINT | - | 0.525822 |
| XGBoost | 0.931952 | 0.529457 |
| XTab | - | 0.528062 |

### E.6 Performance Profiles on Small and Large Data Domain

For a more fine-granular analysis of the models' performance profiles, we conducted the analysis proposed in Section 5.6 in the small- and large-data regime separately.

**Small-Data Domain.** In Figure 31, the performance profiles are shown w.r.t. the measured inference time (left) and the measured total time (right) in the small-data regime. The models CatBoost and AutoGluon yield the best performance-time ratios, with SAINT from the transformer-base models being a competitor in increasing the performance ratio $\tau$. The models FT-Transformer, ResNet, and TabNet yield similar results, where the first performs slightly better for small performance ratios, i.e., the models yield a better performance-cost ratio for a larger amount of datasets. The worst trade-off is given by TB-BERTa, where the inference time outweighs the performance.

On the right side, the performance plots are given w.r.t. the total time. In the small-data regime, as discussed in Section 5.6, the model TabPFN yields strong performance-cost ratios resulting in a superior performance followed by XGBoost, and the feedforward models MLP, and ResNet. Due to the larger training time, the fine-tuned models CARTE, XTab, and TP-BERTa are not competitive with models from other model families.

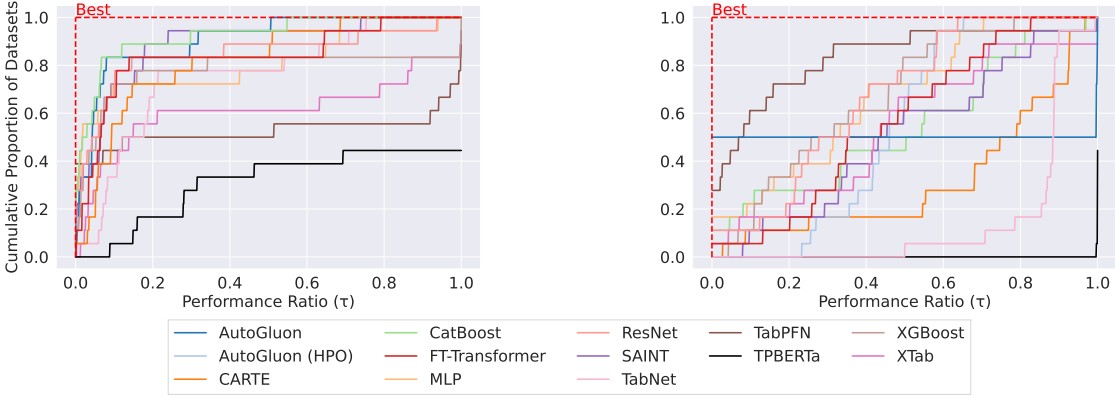

Figure 31: Performance profiles in the small data domain. **Left:** Performance profiles based on inference time. **Right:** Performance profiles based on total time. Steeper curves indicate better overall performance and efficiency across datasets.

**Large-Data Domain.** In Figure 32, the performance profiles are shown w.r.t. the measured inference time (left) and the measured total time(right) in the large-data regime. As discussed in Section 5, TabPFN is only applicable on small-data, hence, it is not included in the large-data analysis. Regarding the inference time, the models AutoGluon as an AutoML-driven approach and CatBoost from the GDBTs are superior compared to other approaches. FT-Transformer show strong results on about half the datasets used in our analysis, but cannot hold up the performance overall the whole set. The models ResNet, FT-Transformer, CARTE, and SAINT show slightly better trade-off values compared to other competitors for an increase performance ratio $\tau$. As before, TP-Berta struggles to be competitive and shows the worst performance-cost ratios.

When considering the total amount of time, the models AutoGluon (HPO) and XGBoost show the strongest performance-cost trade-offs. It is followed by CatBoost from the GDBTs family, followed by the lightweight feedforward networks, ResNet and MLP. From the fine-tuned models, XTab beats CARTE, whereas FT-Transformer wins over SAINT and Tabnet from the transformer-based approaches. Like before, TP-BERTa could not be competitive to any of the other approaches.

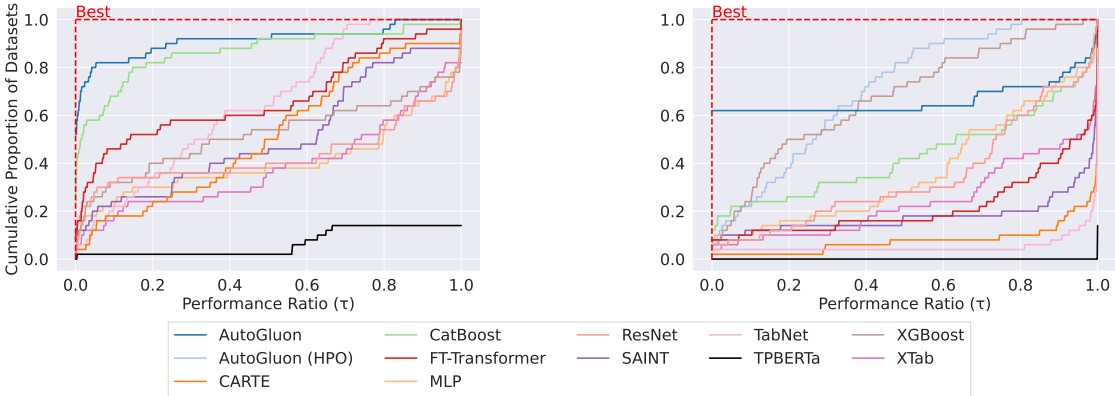

Figure 32: Performance profiles in the large data domain. **Left:** Performance profiles based on inference time. **Right:** Performance profiles based on total time. Steeper curves indicate better overall performance and efficiency across datasets.

## F    Ablating the Choice of Refitting

In this ablation, we explore whether refitting the model on the combined training and validation sets (after hyperparameter optimization, HPO) provides any measurable benefit. The standard procedure, as described in 3.2, uses a 10-fold nested cross-validation: we split the data into 10 folds, use 9 folds for inner cross-validation and HPO, then identify the best hyperparameter configuration and refit the model on all 9 folds before testing on the remaining fold.

We compare this approach to a no-refitting variant. Here, we still employ 10-fold cross-validation, but replace the inner cross-validation with a single 70/30 split of the 9 folds for training and validation. We train the model on the 70% partition, perform HPO on the 30% partition, and then save both the optimal hyperparameter configuration and the resulting trained model. Hence, when moving to the test fold, we simply load this trained model (with its fixed hyperparameters) instead of retraining on the entire 9-fold set. We repeat this for each of the 10 folds, ensuring the test set remains identical across both approaches.

The results of this ablation for CatBoost and FT-Transformer are presented below, comparing the outcomes with and without refitting.

Figure 33 presents boxplots for both considered methods: CatBoost (left) and FT-Transformer (right). In the left plot, the absence of interquartile ranges indicates that CatBoost without refitting exhibits a highly consistent rank distribution, with a median rank of 1. In contrast, CatBoost with refitting has a median rank of 2. The right plot reveals a slightly different trend for the FT-Transformer. While the refitted FT-

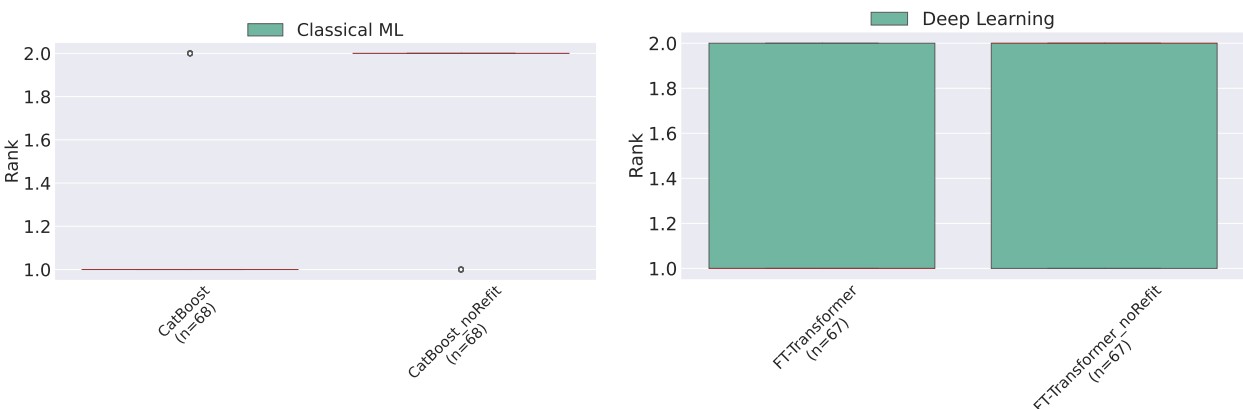

Figure 33: Distribution of ranks for CatBoost **Left** and FT-Transformer **Right**, with and without refitting. Lower ranks indicate better performance. The spread shows the variability in rankings across datasets.

Transformer also achieves a median rank of 1, its interquartile range extends up to rank 2, indicating a broader spread in rank distribution. Meanwhile, the FT-Transformer without refitting maintains a median rank of 2. Note that one dataset was excluded for the FT-Transformer without refitting due to memory constraints.

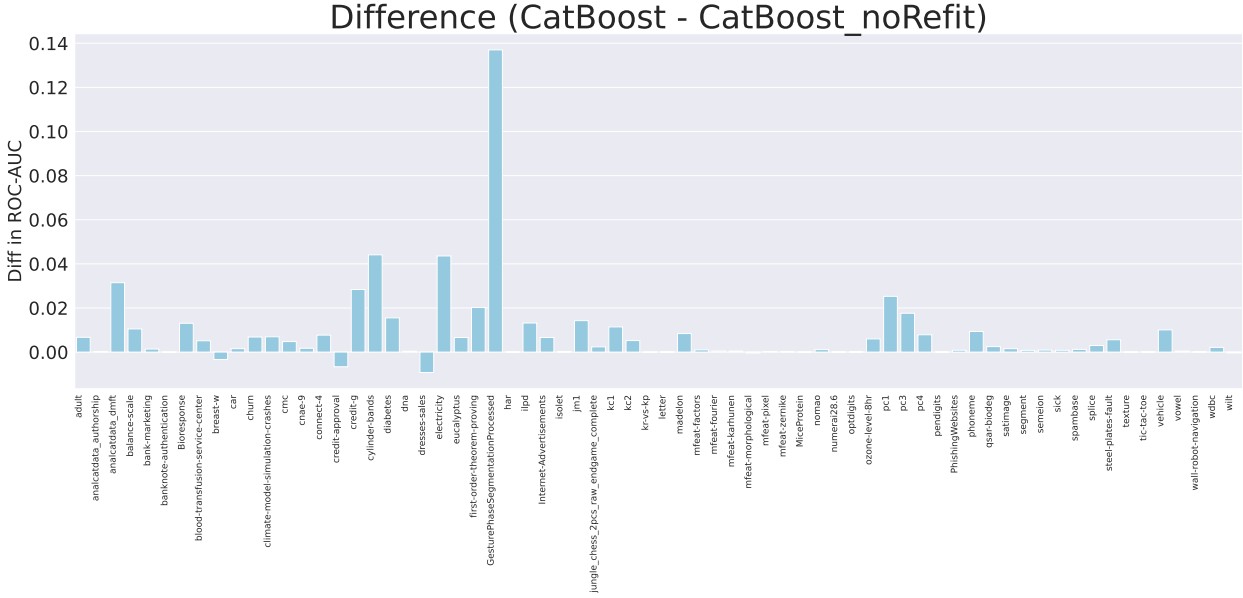

Figure 34: Performance Difference Between CatBoost with refitting and CatBoost without refitting Across Datasets. Positive values indicate an improvement in ROC-AUC when refitting is applied, while negative values indicate a performance drop.

Figure 34 illustrates the performance difference between CatBoost with and without refitting across all datasets. The results clearly show that, with only a few exceptions, CatBoost with refitting consistently outperforms its non-refitted counterpart. Similarly, Figure 35 presents the performance difference for FT-Transformer with and without refitting. Unlike CatBoost, a greater number of datasets favor the non-refitted FT-Transformer. However, overall, the majority of datasets still show improved performance with refitting.

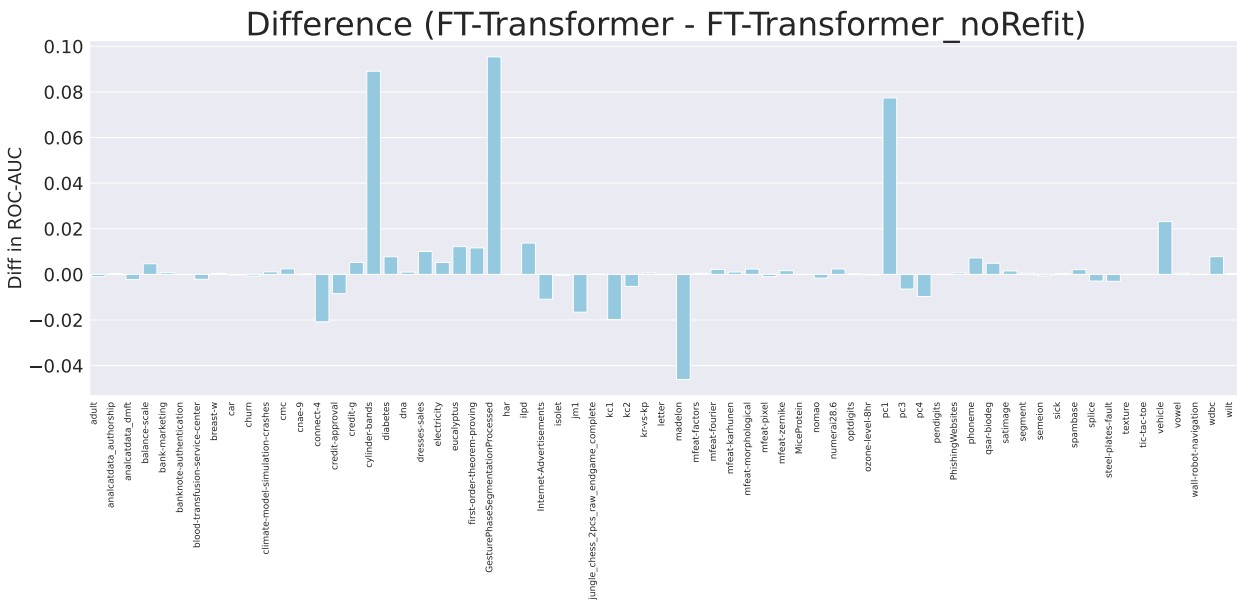

Figure 35: Performance Difference Between FT-Transformer with refitting and FT-Transformer without refitting Across Datasets. Positive values indicate an improvement in ROC-AUC when refitting is applied, while negative values indicate a performance drop.

Furthermore, we conducted a Wilcoxon signed-rank test to compare the performance of refitting versus no-refitting across multiple datasets for both CatBoost and FT-Transformer. The statistical results are summarized in Table 33. For CatBoost, we observed an average performance improvement of 0.0079 when refitting, with a median difference of 0.0016. The Wilcoxon test yielded a test statistic of 180.0000 and a highly significant p-value of $1.2985 \cdot 10^{-9}$. This strongly suggests that refitting leads to a statistically significant and consistent improvement in CatBoost's performance across datasets. Given the very low p-value $p < 0.001$, we can confidently reject the null hypothesis that refitting has no effect. In contrast, for FT-Transformer, the average improvement due to refitting was 0.0035, with a median difference of 0.0004. However, the Wilcoxon test yielded a test statistic of 843.0000 and a p-value of 0.0936, which is not statistically significant ($p > 0.05$). This suggests that while refitting improves FT-Transformer's performance on average, the improvement is not consistent or significant across datasets.

Table 33: Statistical Comparison of Refit vs. No-Refit Methods

| Method Pair | #Datasets | Avg. Diff | Median Diff | Wilcoxon Stat | p-value |
|---|---|---|---|---|---|
| CatBoost vs. CatBoost_noRefit | 68 | 0.0079 | 0.0016 | 180.0000 | **1.298511e-09** |
| FT vs. FT_noRefit | 67 | 0.0035 | 0.0004 | 843.0000 | 9.356765e-02 |

Additionally, Tables 34 and 35 present the raw results of FT-Transformer and CatBoost, respectively, in comparison to their non-refitted counterparts.

Table 34: Average test ROC-AUC per dataset for FT-Transformer using default refitting vs. no refitting across CV folds.

| Dataset | FT-Transformer | FT-Transformer_norefit |
|---|---|---|
| adult | 0.914869 | **0.915875** |
| analcatdata_authorship | **0.999985** | 0.999566 |
| analcatdata_dmft | 0.576947 | **0.579169** |
| balance-scale | **0.999735** | 0.995086 |
| bank-marketing | **0.938198** | 0.937470 |
| banknote-authentication | **1.000000** | **1.000000** |
| Bioresponse | **0.820159** | - |
| blood-transfusion-service-center | 0.745975 | **0.748119** |
| breast-w | **0.989503** | 0.989074 |
| car | 0.999751 | **0.999969** |
| churn | 0.914596 | **0.915300** |
| climate-model-simulation-crashes | **0.934671** | 0.933561 |
| cmc | **0.739402** | 0.736959 |
| cnae-9 | **0.994497** | 0.994377 |
| connect-4 | 0.901170 | **0.921978** |
| credit-approval | 0.935798 | **0.944236** |
| credit-g | **0.783048** | 0.777810 |
| cylinder-bands | **0.915494** | 0.826412 |
| diabetes | **0.831108** | 0.823379 |
| dna | **0.990937** | 0.989937 |
| dresses-sales | **0.620033** | 0.610016 |
| electricity | **0.963076** | 0.957884 |
| eucalyptus | **0.923933** | 0.911772 |
| first-order-theorem-proving | **0.796707** | 0.785106 |
| GesturePhaseSegmentationProcessed | **0.895166** | 0.799810 |
| har | 0.999685 | **0.999706** |
| ilpd | **0.751488** | 0.737753 |
| Internet-Advertisements | 0.974513 | **0.985391** |
| isolet | 0.998817 | **0.999282** |
| jm1 | 0.709321 | **0.725904** |
| jungle_chess_2pcs_raw_endgame_complete | **0.999975** | 0.999861 |
| kc1 | 0.783519 | **0.803310** |
| kc2 | 0.832014 | **0.837281** |
| kr-vs-kp | **0.999777** | 0.999173 |
| letter | **0.999919** | 0.999886 |
| madelon | 0.747391 | **0.793476** |
| mfeat-factors | **0.999015** | 0.998560 |
| mfeat-fourier | **0.984511** | 0.982372 |
| mfeat-karhunen | **0.998682** | 0.997649 |
| mfeat-morphological | **0.970198** | 0.967869 |
| mfeat-pixel | 0.997451 | **0.998448** |
| mfeat-zernike | **0.983479** | 0.981858 |
| MiceProtein | 0.999973 | **1.000000** |
| nomao | 0.990908 | **0.992552** |
| numerai28.6 | **0.530315** | 0.527963 |
| optdigits | **0.999616** | 0.999487 |
| ozone-level-8hr | 0.919484 | **0.919689** |
| pc1 | **0.917591** | 0.840223 |
| pc3 | 0.828743 | **0.835171** |
| pc4 | 0.934944 | **0.944674** |
| pendigits | **0.999703** | 0.999668 |
| PhishingWebsites | **0.996760** | 0.996105 |
| phoneme | **0.965071** | 0.957862 |
| qsar-biodeg | **0.919584** | 0.914716 |
| satimage | **0.993516** | 0.992003 |
| segment | **0.994124** | 0.993598 |
| semeion | 0.995548 | **0.996208** |
| sick | **0.997937** | 0.997762 |
| spambase | **0.985969** | 0.983881 |
| splice | 0.992276 | **0.995195** |
| steel-plates-fault | 0.959182 | **0.962215** |
| texture | **0.999983** | 0.999973 |
| tic-tac-toe | 0.996152 | **0.996209** |
| vehicle | **0.963362** | 0.940233 |
| vowel | **0.999713** | 0.999198 |
| wall-robot-navigation | **0.999900** | 0.999870 |
| wdbc | **0.993967** | 0.986203 |
| wilt | **0.993047** | 0.992642 |

Table 35: Average test ROC-AUC per dataset for CatBoost using default refitting vs. no refitting across CV folds.

| Dataset | CatBoost | CatBoost_norefit |
|---|---|---|
| adult | **0.930747** | 0.924052 |
| analcatdata_authorship | **0.999662** | 0.999470 |
| analcatdata_dmft | **0.579136** | 0.547691 |
| balance-scale | **0.972625** | 0.962132 |
| bank-marketing | **0.938831** | 0.937464 |
| banknote-authentication | 0.999935 | **0.999979** |
| Bioresponse | **0.885502** | 0.872449 |
| blood-transfusion-service-center | **0.754965** | 0.749848 |
| breast-w | 0.989162 | **0.992507** |
| car | **1.000000** | 0.998453 |
| churn | **0.922968** | 0.916146 |
| climate-model-simulation-crashes | **0.951480** | 0.944551 |
| cmc | **0.740149** | 0.735398 |
| cnae-9 | **0.996316** | 0.994599 |
| connect-4 | **0.921050** | 0.913372 |
| credit-approval | 0.934006 | **0.940661** |
| credit-g | **0.801762** | 0.773381 |
| cylinder-bands | **0.912070** | 0.867995 |
| diabetes | **0.837869** | 0.822365 |
| dna | **0.995028** | 0.994658 |
| dresses-sales | 0.595731 | **0.605008** |
| electricity | **0.980993** | 0.937421 |
| eucalyptus | **0.923334** | 0.916719 |
| first-order-theorem-proving | **0.831775** | 0.811589 |
| GesturePhaseSegmentationProcessed | **0.916674** | 0.779683 |
| har | **0.999941** | 0.999887 |
| ilpd | **0.744702** | 0.731536 |
| Internet-Advertisements | **0.979120** | 0.972513 |
| isolet | **0.999389** | 0.999282 |
| jm1 | **0.756611** | 0.742362 |
| jungle_chess_2pcs_raw_endgame_complete | **0.976349** | 0.973983 |
| kc1 | **0.825443** | 0.814042 |
| kc2 | **0.846802** | 0.841593 |
| kr-vs-kp | 0.999392 | **0.999419** |
| letter | **0.999854** | 0.999802 |
| madelon | **0.937562** | 0.929178 |
| mfeat-factors | **0.998910** | 0.997917 |
| mfeat-fourier | **0.984714** | 0.984229 |
| mfeat-karhunen | **0.999264** | 0.998802 |
| mfeat-morphological | 0.965406 | **0.965867** |
| mfeat-pixel | **0.999422** | 0.999183 |
| mfeat-zernike | **0.977986** | 0.977831 |
| MiceProtein | **1.000000** | 0.999991 |
| nomao | **0.996439** | 0.995329 |
| numerai28.6 | **0.529404** | 0.529350 |
| optdigits | **0.999844** | 0.999780 |
| ozone-level-8hr | **0.929094** | 0.923125 |
| pc1 | **0.875471** | 0.850199 |
| pc3 | **0.851122** | 0.833527 |
| pc4 | **0.953309** | 0.945471 |
| pendigits | **0.999752** | 0.999728 |
| PhishingWebsites | **0.996482** | 0.995649 |
| phoneme | **0.968024** | 0.958699 |
| qsar-biodeg | **0.930649** | 0.928167 |
| satimage | **0.991978** | 0.990444 |
| segment | **0.996231** | 0.995441 |
| semeion | **0.998687** | 0.997784 |
| sick | **0.998331** | 0.997520 |
| spambase | **0.989935** | 0.988718 |
| splice | **0.995472** | 0.992511 |
| steel-plates-fault | **0.974350** | 0.968766 |
| texture | **0.999948** | 0.999946 |
| tic-tac-toe | **1.000000** | 0.999952 |
| vehicle | **0.943460** | 0.933394 |
| vowel | **0.999259** | 0.998833 |
| wall-robot-navigation | **0.999990** | 0.999910 |
| wdbc | **0.993813** | 0.991693 |
| wilt | 0.990950 | **0.991393** |

