# OpenReview forum: "Is Deep Learning finally better than Decision Trees on Tabular Data?"
_TMLR — Rejected by TMLR_

### Review · Reviewer_MUe8 · 2024-12-15

**Summary Of Contributions:**

The paper presents and extensive tabular benchmark for classification tasks, comparing GBDTs with DL methods, prior fitted networks and Tabular foundation models. With that, it seamlessly fits in well with other benchmarks performed for tabular DL [1, 2, 3].
The findings are consistent with other results [4] - TabPFN and CatBoost are very strong contenders - and results are well reported.
The HPO scheme is well documented and differs from other benchmarks, filling a small but existing gap in tabular benchmark literature.


---
[1] Gorishniy, Yury, et al. "Revisiting deep learning models for tabular data." NeurIPS (2021).
[2] Borisov, Vadim, et al. "Deep neural networks and tabular data: A survey." IEEE transactions on neural networks and learning systems (2022).
[3] Grinsztajn, Léo, Edouard Oyallon, and Gaël Varoquaux. "Why do tree-based models still outperform deep learning on typical tabular data?." NeurIPS (2022).
[4] McElfresh, Duncan, et al. "When do neural nets outperform boosted trees on tabular data?." NeurIPS (2024).

**Audience:**

Yes

**Claims And Evidence:**

Yes

**Requested Changes:**

**Disclaimer**
Most of the "possible changes" refer to very recent studies, most of which are preprints. I do not think it is reasonable to ask the authors to include them, as these studies likely came out after the manuscript was finalized. However, I will present them here, and if possible, the authors can consider including them in their manuscript.

**Possible Changes**
- There are some very recent benchmark studies [1] and [2]. It would be interesting to include them in Table 1. Additionally, a comparison regarding the datasets from [2] could be of interest.
- Recent models such as [3] and [4] claim superiority over existing models. While the computational cost of including these could be significant, if it is feasible, it would be interesting to include them. The same holds for [5] and [6].
    - Additionally, a simple MLP leveraging embeddings (PLR) or PLE encodings + embeddings [7] would be very interesting in the comparison. Given your extensive HPO scheme, this would further verify/test whether simple, well-tuned networks actually excel on tabular data [8].

**Requested Changes**
- It would be interesting if you could include more details regarding results for different datasets. For example, results (including/excluding TabPFN) for datasets with ≤1000 observations, between 1000 and 5000 observations, and so on. Similarly, for the number of features: Is there a model that performs better when more features are available compared to the other models?
- Please release the code and the results for this such that other researchers can benchmark their newly developed model against your results.
    - It also would be nice, if you would publish all results, also for the different hpo steps if possible.

---

[1] Ye, Han-Jia, et al. "A closer look at deep learning on tabular data." arXiv preprint arXiv:2407.00956 (2024).
[2] Rubachev, Ivan, et al. "TabReD: Analyzing Pitfalls and Filling the Gaps in Tabular Deep Learning Benchmarks." arXiv preprint arXiv:2406.19380 (2024).
[3] Gorishniy, Yury, Akim Kotelnikov, and Artem Babenko. "TabM: Advancing Tabular Deep Learning with Parameter-Efficient Ensembling." arXiv preprint arXiv:2410.24210 (2024).
[4] Thielmann, Anton Frederik, et al. "Mambular: A sequential model for tabular deep learning." arXiv preprint arXiv:2408.06291 (2024).
[5] Gorishniy, Yury, et al. "TabR: Tabular Deep Learning Meets Nearest Neighbors." ICLR (2024).
[6] Müller, Andreas, Carlo Curino, and Raghu Ramakrishnan. "MotherNet: A Foundational Hypernetwork for Tabular Classification." arXiv preprint arXiv:2312.08598 (2023).
[7] Gorishniy, Yury, Ivan Rubachev, and Artem Babenko. "On embeddings for numerical features in tabular deep learning." NeurIPS (2022).
[8] Kadra, Arlind, et al. "Well-tuned simple nets excel on tabular datasets." NeurIPS (2021).

**Strengths And Weaknesses:**

**Strengths**
- Overall, the paper is well-written, and the benchmarks are very extensive, with significant HPO. While the findings are not surprising, they strengthen the insights of the field and confirm that TabPFNs are very strong models for small datasets. For medium/larger datasets, CatBoost remains the model to beat.
- It is always beneficial to have an additional benchmark comparing models with different HPO schemes.
- The described methodology is well-grounded and explores a slightly different realm than previous studies [1, 2, 3, 4].
- It fits well within the scope of TMLR.

**Questions**
- How do you perform the ranking? Figure 10 indicates that you are not just computing the rank for each task and that each dataset has a rank of 1-11.
    - Figure 10: Since these are relative ranks, how is it possible that every model's rank improved (except for TP-Berta)? Shouldn't the average ranks between models at least be consistent? I guess I am missing the point, could you explain the figure?
- What preprocessing was performed? I might have missed it, but I only found that it was consistent across models. If so, did you leverage PLE for DL methods? Also, which embeddings did you use for FT-Transformer/SAINT? PLR, linear + ReLU, just linear [5]? Did you also leverage embeddings for the ResNet architecture?
- Why use 100 trials or 23 hours? Wouldn't it be more sensible to use a fixed time budget as [1]? GBDTs are much more efficient, so 100 fits can be performed in a fraction of the time, e.g., a FT-Transformer would require more time for this.
- I did not find any information regarding batch sizes for the DL models. Maybe I have overlooked it, but I think you have not optimized it, right? How was the batch size set, and was it different for different datasets? Additionally, what was the reasoning for not optimizing it?
- Figures 2-3: Why is \(n\) different for each model? More specifically, how does it change when only the datasets used for TabPFN are compared for all models and hence \(n=17\)?
- What is n_ensembles in the TabPFN default configuration?
- Why do you optimize the number of epochs? Isn't it more sensible to optimize the patience while setting a fixed maximum number of epochs that is never reached? Since the number of epochs has the strongest effect on DL models, having a parameter space starting from 10 epochs seems to be the main cause here. Do you have any further insights regarding this?

**Minor**
- Section 3.1 typo: "iff."
- Figures 12 and 13 are not very readable. I suggest rotating them and making each a full page (maybe in the appendix).


---
[1] Gorishniy, Yury, et al. "Revisiting deep learning models for tabular data." NeurIPS (2021).
[2] Borisov, Vadim, et al. "Deep neural networks and tabular data: A survey." IEEE transactions on neural networks and learning systems (2022).
[3] Grinsztajn, Léo, Edouard Oyallon, and Gaël Varoquaux. "Why do tree-based models still outperform deep learning on typical tabular data?." NeurIPS (2022).
[4] McElfresh, Duncan, et al. "When do neural nets outperform boosted trees on tabular data?." NeurIPS (2024).
[5] Gorishniy, Yury, Ivan Rubachev, and Artem Babenko. "On embeddings for numerical features in tabular deep learning." NeurIPS (2022).

---

> ### Author Response · Authors · 2025-02-01
>
> We sincerely appreciate your thorough and constructive feedback. Below, we address your concerns in detail and highlight the additions and improvements we have made to the manuscript.
>
> ---
> ### Answers to Questions (1/2)
> ---
>
> **How do you perform the ranking? Figure 10 indicates that you are not just computing the rank for each task and that each dataset has a rank of 1-11.**
>
> **Figure 10: Since these are relative ranks, how is it possible that every model's rank improved (except for TP-Berta)? Shouldn't the average ranks between models at least be consistent? I guess I am missing the point, could you explain the figure?**
>
> In Figure 10, the "rank" we report is not a global rank among all methods on every dataset. Instead, each method’s HPO version is only compared to its own default version, a simple two-model comparison. Concretely, for a given dataset *D*, we compare "Method X (HPO)" vs. "Method X (Default)." Whichever achieves higher performance on *D* gets rank 1, and the other gets rank 2 (if there is a tie, both receive rank 1.5). We then average these ranks across all datasets for that method, producing one value for the HPO version and one value for the default version of each method.
>
> Because each method is effectively in a "2-player race,” HPO vs. default, there is no direct competition among different methods (e.g., CatBoost vs. XGBoost) in this figure. As a result, multiple methods can all "improve" via HPO relative to their own default settings, which explains why there is an improvement for nearly every method except TP-Berta. Every method’s improvement is measured only relative to itself. The purpose of Figure 10 is thus to highlight how much each method gains from hyperparameter optimization compared to its own default hyperparameters, rather than comparing all methods directly on every dataset.
>
> In contrast, for the boxplots in the paper, we use a different ranking approach that ranks all available methods on each dataset. Here, we compare all methods on each dataset and assign a rank of 1 to the highest-performing method, rank 2 to the second-highest, and so on. This yields a complete head-to-head competition among all participating methods for every dataset. We then plot these ranks for every method across datasets as boxplots. A method that outperforms others on many datasets will show a lower (better) median rank in the boxplot, whereas methods that perform worse or more variably will have higher (worse) median ranks or broader distributions.
>
> ---
>
> **What preprocessing was performed? I might have missed it, but I only found that it was consistent across models. If so, did you leverage PLE for DL methods? Also, which embeddings did you use for FT-Transformer/SAINT? PLR, linear + ReLU, just linear [5]? Did you also leverage embeddings for the ResNet architecture?**
>
> Our codebase is heavily based on the implementation from [1] and follows their preprocessing pipeline without adding new embedding techniques such as PLE or custom PLR embeddings. Every deep learning model (FT-Transformer, SAINT, ResNet) relies on the simple linear embedding layers that come with its original implementation, and we do not introduce additional embedding structures. The data preprocessing is kept consistent across all models to ensure a fair comparison. For numerical features, we apply quantile transformation fitted on the training set only; missing values are typically imputed with the mean. For categorical features, we handle missing values by replacing them with a special token, and then encode them using an OrdinalEncoder or (if specified) one-hot or target encoding, in a manner that is again consistent with [1]. We have updated the manuscript accordingly to reflect the missing information as pointed out by the reviewer.
>
> ---
>
> **Why use 100 trials or 23 hours? Wouldn't it be more sensible to use a fixed time budget as [1]? GBDTs are much more efficient, so 100 fits can be performed in a fraction of the time, e.g., a FT-Transformer would require more time for this.**
>
> We employed a "100 trials or 23 hours" budget to ensure every method had both a maximum number of hyperparameter evaluations and a maximum wall-clock time. Although a purely time-based budget is also valid, having both trial and time caps prevents faster methods (e.g., GBDTs) from overfitting the validation data with a very large number of trials, while still giving more computationally demanding architectures (like FT-Transformer) enough time to complete a sufficient number of optimization runs. This approach is similar to what [2] used, where for every algorithm-dataset pair they allowed up to 10 hours with a maximum of 30 hyperparameter evaluations, and also limited every train/evaluation cycle to two hours.
>
> ---

---

> > ### Author Response · Authors · 2025-02-01
> >
> > ---
> > ### Answers to Questions (2/2)
> > ---
> >
> >
> > **I did not find any information regarding batch sizes for the DL models. Maybe I have overlooked it, but I think you have not optimized it, right? How was the batch size set, and was it different for different datasets? Additionally, what was the reasoning for not optimizing it?**
> >
> > We did not optimize batch size in our experiments due to memory constraints on our compute system. Instead, we used a simple heuristic as in [3] determined by the number of features: for example, a batch size of 512 was used if a dataset had 32 or fewer features, dropping to 128 up to 100 features, and to 32 for even higher dimensional inputs. Although this policy meant that the batch size could vary across different datasets, it was applied consistently across all methods on a given dataset to ensure a fair comparison. Training with a large batch size on high-dimensional datasets would often raise out-of-memory errors, so this strategy was adopted to maintain uniformity and feasibility. Ideally, we would have also tuned the batch size, but resource limitations made the above heuristic both necessary and practical.
> >
> > ---
> >
> > **Figures 2-3: Why is (n) different for each model? More specifically, how does it change when only the datasets used for TabPFN are compared for all models and hence (n=17)?**
> >
> > The value of n can differ across methods because we could not always run every model on every dataset. Certain methods ran into memory issues (e.g., SAINT, TP-BERTa), and in the case of XTab, we excluded datasets that had been used during its pretraining. TabPFN was restricted to 17 datasets from the OpenMLCC18 benchmark because it cannot handle more than 1000 instances or 100 features, so we only ran it on that subset.
> >
> > In order to provide a fair comparison where every method is evaluated on exactly the same datasets, we employed critical difference (CD) diagrams. We also added supplementary boxplots in Appendix E, where n is the same for all datasets included in those analyses.
> >
> > ---
> >
> > **What is n_ensembles in the TabPFN default configuration?**
> >
> > In the default TabPFN configuration, n_ensembles (or N_ensemble_configurations) is set to 32. This parameter controls how many slightly different predictions the model will generate and ensemble. Specifically, TabPFN creates multiple rotations (permutations) of the features and/or labels and then averages their predictions, which is intended to improve robustness and accuracy.
> >
> > ---
> >
> > **Why do you optimize the number of epochs? Isn't it more sensible to optimize the patience while setting a fixed maximum number of epochs that is never reached? Since the number of epochs has the strongest effect on DL models, having a parameter space starting from 10 epochs seems to be the main cause here. Do you have any further insights regarding this?**
> >
> > We acknowledge that using a patience-based early stopping (with a sufficiently large maximum epoch count) is another valid approach, conceptually, it also determines how long a model trains. However, we chose to optimize the number of epochs explicitly for two primary reasons. First, we found it simpler and more consistent across our entire pipeline to set an upper bound of epochs and tune that within our hyperparameter space, rather than relying on every method’s sometimes-unique early-stopping implementation. Second, during our nested cross-validation routine, we wanted a direct handle on the model’s training duration rather than an indirect measure like patience, which can interact differently with different validation criteria for the different models.
> >
> > In practice, patience tuning and epoch tuning can yield very similar outcomes since both control how long the model trains before stopping. We found that explicitly searching over a range of epochs (starting from 10) gave enough flexibility for the hyperparameter optimizer to discover the appropriate training length. While it is true that the number of epochs can heavily impact deep learning models, we aimed to give them a sufficiently large search space so that if more training was beneficial, the tuning process could discover that. Hence, we do not believe our choice to optimize epochs directly was a limiting factor; rather, it was a straightforward and uniform way to manage the training duration across all deep learning methods in our study. Furthermore, even though we tuned the number of epochs for the two newly added baselines (MLP-PLR and RealMLP), our hyperparameter analysis revealed that number of epochs was not the most influential hyperparameter, instead, learning rate was the key parameter for MLP-PLR, while hidden sizes had the greatest impact on RealMLP.
> >
> > ---

---

> ### Author Response · Authors · 2025-02-01
>
> ---
> ### Requested Changes
> ---
>
> **It would be interesting if you could include more details regarding results for different datasets. For example, results (including/excluding TabPFN) for datasets with ≤1000 observations, between 1000 and 5000 observations, and so on. Similarly, for the number of features: Is there a model that performs better when more features are available compared to the other models?**
>
> **Please release the code and the results for this such that other researchers can benchmark their newly developed model against your results. It also would be nice, if you would publish all results, also for the different HPO steps if possible.**
>
> We appreciate the reviewer’s interest in seeing more granular results for different dataset sizes and feature counts. We have incorporated the proposed analysis into Appendix E of our manuscript, where we provide an extended analysis that categorizes datasets by instance count (e.g., ≤1000, 1001–5000, etc.) and further subdivides them by the number of features (≤100, 101–500, etc.). Additionally, our code is now anonymously available at: [https://anonymous.4open.science/r/TabularStudy-0EE2/README.md](https://anonymous.4open.science/r/TabularStudy-0EE2/README.md).
>
> ---
> ### Possible Changes
> ---
>
> **There are some very recent benchmark studies [1] and [2]. It would be interesting to include them in Table 1. Additionally, a comparison regarding the datasets from [2] could be of interest.**
>
> **Recent models such as [3] and [4] claim superiority over existing models. While the computational cost of including these could be significant, if it is feasible, it would be interesting to include them. The same holds for [5] and [6].**
>
> **Additionally, a simple MLP leveraging embeddings (PLR) or PLE encodings + embeddings [7] would be very interesting in the comparison. Given your extensive HPO scheme, this would further verify/test whether simple, well-tuned networks actually excel on tabular data [8].**
>
> We appreciate the pointers to these recent studies and the suggestions for additional methods. Due to the significant computational cost of running every new model and new datasets, as well as the complexity of integrating specialized components into our existing pipeline, we were not able to include all of them in the current study. Nonetheless, because an MLP with PLR encodings is relatively simple to implement and run, we conducted the proposed experiment and updated our manuscript to include these results. Furthermore, we have added both [4] and [5] to Table 1.
>
> ---
>
> [1] Gorishniy, Y., Rubachev, I., Khrulkov, V., & Babenko, A. (2021). Revisiting deep learning models for tabular data. Advances in Neural Information Processing Systems, 34, 18932-18943.
>
> [2] McElfresh, D., Khandagale, S., Valverde, J., Prasad C, V., Ramakrishnan, G., Goldblum, M., & White, C. (2024). When do neural nets outperform boosted trees on tabular data?. Advances in Neural Information Processing Systems, 36.
>
> [3] Chen, J., Yan, J., Chen, Q., Chen, D. Z., Wu, J., & Sun, J. (2024, August). Can a Deep Learning Model be a Sure Bet for Tabular Prediction?. In Proceedings of the 30th ACM SIGKDD Conference on Knowledge Discovery and Data Mining (pp. 288-296).
>
> [4] Ye, H. J., Liu, S. Y., Cai, H. R., Zhou, Q. L., & Zhan, D. C. (2024). A closer look at deep learning on tabular data. arXiv preprint arXiv:2407.00956.
>
> [5] Gorishniy, Y., Rubachev, I., Kartashev, N., Shlenskii, D., Kotelnikov, A., & Babenko, A. (2024). TabR: Tabular Deep Learning Meets Nearest Neighbors. In The Twelfth International Conference on Learning Representations.
>
> ---

---

> > ### Comment · Reviewer_MUe8 · 2025-02-01
> > **Acknowledgement of Changes**
> >
> > Dear Authors,
> >
> > Thank you for thoroughly addressing all questions and for the thoughtful revisions you’ve made. I truly appreciate your efforts, and I believe the additional results significantly strengthen the paper.
> >
> > I already found the paper compelling before the revision, and after reading the other reviewers' comments alongside your comprehensive response, I appreciate it even more.
> >
> > I fully agree that incorporating the suggested methods [1, 2, 3, 4] into the benchmark is beyond the scope of this work. I also genuinely appreciate the inclusion of the MLP-PLR [5]. While you might consider briefly mentioning the existence of the other methods [1, 2, 3, 4] not covered, this is entirely optional and not necessary for the current version.
> >
> > Given the quality of your revisions and the strengthened contribution, I have no further questions or suggestions for changes, and I fully support the paper’s acceptance.
> >
> > ---
> > [1] Gorishniy, Yury, Akim Kotelnikov, and Artem Babenko. "TabM: Advancing Tabular Deep Learning with Parameter-Efficient Ensembling." arXiv preprint arXiv:2410.24210 (2024).
> >
> > [2] Thielmann, Anton Frederik, et al. "Mambular: A sequential model for tabular deep learning." arXiv preprint arXiv:2408.06291 (2024).
> >
> > [3] Gorishniy, Yury, et al. "TabR: Tabular Deep Learning Meets Nearest Neighbors." ICLR (2024).
> >
> > [4] Müller, Andreas, Carlo Curino, and Raghu Ramakrishnan. "MotherNet: A Foundational Hypernetwork for Tabular Classification." arXiv preprint arXiv:2312.08598 (2023).
> >
> > [5] Gorishniy, Yury, Ivan Rubachev, and Artem Babenko. "On embeddings for numerical features in tabular deep learning." NeurIPS (2022). [8] Kadra, Arlind, et al. "Well-tuned simple nets excel on tabular datasets." NeurIPS (2021).

---

### Review · Reviewer_tLh9 · 2025-01-09

**Summary Of Contributions:**

This paper is a benchmarking & reproducibility study on tabular classification. It evaluates 11 different methods, 8 of which are recent deep learning-based methods, on OpenML-CC18. Notably, compared to McElfresh et al. (2023), this paper evaluates AutoGluon and some recent finetuning-based methods.

**Audience:**

Yes

**Claims And Evidence:**

Yes

**Requested Changes:**

See W1 - W4. Note that while I've marked "Claims And Evidence" as "no", I think that, related to W1-W4, explaining the reasoning behind your choices, clarifying the manuscript, or making changes are fairly straightforward.

UPDATE 2024-02-01: My requested changes above have been addressed in the revision and rebuttal.

**Strengths And Weaknesses:**

Strengths

(S1) Having an evaluation that includes AutoGluon and recent finetuning-based methods (CARTE, TP-BERTa, and XTab) is helpful, as these have tended to be omitted previously.
(S2) The overall protocol of the study (especially with hypertuning) is thorough and appears to be well-executed. And the additional analyses on meta-features at the end is insightful.

Weaknesses

(W1) The paper claims that refitting on the train+validation data is important for evaluating methods, but doesn't back up this claim. Can you please provide results that show that refitting on both train+validation makes a difference. Especially for the size of datasets in this benchmark, my priors are that this wouldn't make a difference, but I'll be happy to update my priors if data is presented on this.

(W2) Re: "After the maximal number of trials T is reached or the time budget is exceeded, we select the best hyperparameter setting in line 13." Having a maximum number of trials does not make sense to me. It seems to unfairly penalize methods that are faster. And, even if one were to do this, it would be better to have separate results for maximum trials and maximum time: the current setup makes the results hard to interpret.

(W3) There is no anonymized code (a la https://anonymous.4open.science/). For reproducibility studies, reproducibility is of course critical, so the absence of code at review time is an issue.

(W4) "Both TP-Berta and TabPFN are only evaluated on a small subset of the available datasets as indicated by the small cumulative proportion value for both approaches, where the latter shows its strong performance on the datasets it has been evaluated on by a steep increase." This highlights the weakness in this kind of plot. Would rather see subsetting with TabPFN as suggested above, or separate plots for small datasets (with TabPFN included) and for large datasets (without TabPFN shown).

Minor questions / concerns / comments:

(M1) To be clear, "there are additional analyses I would have liked to see" does not appear to be an appropriate reason to reject a paper from TMLR. Nevertheless, I am mentioning these, as I think these may be worthwhile to share with the authors. First, it would be nice to include at least one of the recent hypernetwork-based methods [eg 1, 2], since they're allegedly useful for fast inference. Second, it would be nice to include at least one of the LLM feature engineering methods [eg 3, 4]. Third, it would be nice to include at least one of the follow-up works to TabPFN [eg 5]. Fourth, previous benchmarking papers have been strangely focused on classification vs regression, so evaluating on regression would be interesting.

(M2) I appreciated the plots in Appendix D of AUROC vs hyperparameter. But in addition to the marginal distributions which were provided, I'd be interested in knowing the optimal combination of hyperparameters. (Note that Figure 18 has wide marginal error bars for XGBoost, suggesting that the right combination is what matters most.) Including a table of the optimal hyperparameter combination for each dataset would be very nice to see.

(M3) For XGBoost, why did you use one hot encoding of categorical -- why not ordinal encoding? Also, it's pretty popular to use early stopping with XGBoost -- why not do this?

(M4) For TabPFN, why not run it on large datasets by subset features and/or samples as necessary (as easily supported by TabPFN and used previously in [eg 6, 7])?

(M5) How did you handle missingness? Or does this not occur at all in the benchmark datasets?

(M6) What does "Min-Max Class Freq" mean in Table 19, and how can it sometimes be 1.00?

[1] Bonet, David, Daniel Mas Montserrat, Xavier Giró-i-Nieto, and Alexander G. Ioannidis. "HyperFast: Instant Classification for Tabular Data." In Proceedings of the AAAI Conference on Artificial Intelligence, vol. 38, no. 10, pp. 11114-11123. 2024.

[2] Müller, Andreas, Carlo Curino, and Raghu Ramakrishnan. "MotherNet: A Foundational Hypernetwork for Tabular Classification." arXiv preprint arXiv:2312.08598 (2023).

[3] Hollmann, Noah, Samuel Müller, and Frank Hutter. "Large language models for automated data science: Introducing caafe for context-aware automated feature engineering." Advances in Neural Information Processing Systems 36 (2024).

[4] Nam, Jaehyun, Kyuyoung Kim, Seunghyuk Oh, Jihoon Tack, Jaehyung Kim, and Jinwoo Shin. "Optimized Feature Generation for Tabular Data via LLMs with Decision Tree Reasoning." arXiv preprint arXiv:2406.08527 (2024).

[5] Ma, Junwei, Valentin Thomas, Rasa Hosseinzadeh, Hamidreza Kamkari, Alex Labach, Jesse C. Cresswell, Keyvan Golestan, Guangwei Yu, Maksims Volkovs, and Anthony L. Caterini. "TabDPT: Scaling Tabular Foundation Models." arXiv preprint arXiv:2410.18164 (2024).

[6] Calvin McCarter. What exactly has TabPFN learned to do? In ICLR Blogposts 2024, 2024.

[7] Feuer, Benjamin, Chinmay Hegde, and Niv Cohen. "Scaling tabpfn: Sketching and feature selection for tabular prior-data fitted networks." arXiv preprint arXiv:2311.10609 (2023).

UPDATE 2024-02-01: My weaknesses above have been addressed in the revision and rebuttal.

---

> ### Author Response · Authors · 2025-02-01
>
> We would like to thank the reviewer for evaluating our work. In the following, we provide a detailed account of all the changes that we have made in the revised version of the paper to address the reviewer’s concerns.
>
> ---
>
> **(W1) The paper claims that refitting on the train+validation data is important for evaluating methods, but doesn't back up this claim.**
>
> We agree with the reviewer’s point that the performance and the benefits of the refitting procedure relate to a dataset’s size. Following the standard error \(\sigma/\sqrt{n}\), where \(\sigma\) is the standard deviation and \(n\) refers to the sample size, with an increasing number of data samples used as input for training our model, the benefits of refitting are marginal. However, in our experimental setup, we include the refitting in our evaluation pipeline as it has been applied in literature [1] and is an established procedure in common libraries where refitting is done per default (scikit-learn; [https://scikit-learn.org/stable/modules/generated/sklearn.model_selection.GridSearchCV.html](https://scikit-learn.org/stable/modules/generated/sklearn.model_selection.GridSearchCV.html), refit=True).
>
> To validate our claim, in our revised draft, in Appendix F, we show the benefits of refitting on two models (CatBoost and FT-Transformer) to empirically substantiate the refitting procedure applied in our evaluation protocol. As observed in the provided analysis, refitting provides better test performance, and in some cases, the difference between refitting and not refitting is statistically significant.
>
> ---
>
> **(W2) Having a maximum number of trials does not make sense to me. It seems to unfairly penalize methods that are faster. And, even if one were to do this, it would be better to have separate results for maximum trials and maximum time: the current setup makes the results hard to interpret.**
>
> Our primary aim was to provide an upper bound on the hyperparameter optimization (HPO) search space to avoid overfitting while still allowing sufficiently large exploration. In practice, it is quite common to cap both the number of trials and the total time. [2], for example, allocate a maximum of 10 hours and up to 30 hyperparameter evaluations for each method-dataset pair, with each individual train/evaluation cycle limited to two hours. Setting both trial and time limits ensures that no single method can run indefinitely, while still offering every method enough time to optimize its hyperparameters.
>
> ---
>
> **(W3) There is no anonymized code.**
>
> Thank you very much for this important remark. For reproducibility, our code is now anonymously available at: [https://anonymous.4open.science/r/TabularStudy-0EE2/README.md](https://anonymous.4open.science/r/TabularStudy-0EE2/README.md).
>
> ---
>
> **(W4) Would rather see subsetting with TabPFN as suggested above, or separate plots for small datasets (with TabPFN included) and for large datasets (without TabPFN shown).**
>
> We appreciate the reviewer’s feedback on the limitations of our original plots. In Appendix E, we address these concerns by categorizing the datasets according to the number of instances and features, which enables separate analyses for smaller and larger datasets. For each category, we provide both boxplots and critical difference diagrams, comparing all available datasets as well as only those subsets shared across methods for a fair comparison. When a category contains fewer than 10 datasets, we present the results in tabular form to maintain clarity and completeness. This approach allows us to include TabPFN on the smaller datasets where it is applicable and to exclude it on larger datasets where it is not.
>
> Additionally, in Appendix E.6, we added performance plots only on the small data regime, where the strength of TabPFN is more clearly illustrated.
>
> ---

---

> ### Author Response · Authors · 2025-02-01
>
> ---
>
> **(M1) First, it would be nice to include at least one of the recent hypernetwork-based methods [eg 1, 2], since they're allegedly useful for fast inference. Second, it would be nice to include at least one of the LLM feature engineering methods [eg 3, 4]. Third, it would be nice to include at least one of the follow-up works to TabPFN [eg 5]. Fourth, previous benchmarking papers have been strangely focused on classification vs regression, so evaluating on regression would be interesting.**
>
> We sincerely appreciate the related work reference in your question. We fully agree that all models mentioned are relevant for comparison. However, many of these models were developed concurrently with our research efforts.
> As authors, we can evaluate only the body of published work available at the time of our study, focusing on models that demonstrated competitive performance up to that point. The most recent works included in our survey is CARTE, TP-Berta, which were published in 2024, and reflect the insights available during the preparation of our manuscript.
>
> ---
>
> **(M2) But in addition to the marginal distributions which were provided, I'd be interested in knowing the optimal combination of hyperparameters.**
>
> In our evaluation protocol, we apply a nested cross-validation step. Hence, for every model-dataset pair, there are 10 best hyperparameter configurations. By taking the number of datasets into account which were used in our empirical study, we end up with 680 optimal configurations, which exceeds a presentable manuscript. However, we present a more thorough hyperparameter analysis in Appendix D where the impact of each individual hyperparameter on the performance metric is illustrated.
>
> ---
>
> **(M3) For XGBoost, why did you use one hot encoding of categorical -- why not ordinal encoding? Also, it's pretty popular to use early stopping with XGBoost -- why not do this?**
>
> Ordinal encoding is indeed another common approach, but we used one-hot encoding for XGBoost to maintain consistency with the codebase from [4], on which our entire pipeline is based. Concerning early stopping in XGBoost, the number of boosting rounds is not a static value, but instead a hyperparameter which is selected based on the task and as such does not necessitate early stopping.
>
> We do the same for e.g. the MLP/ResNet model, to keep the experimental protocol consistent across all models. This way, no single method gains an advantage (or disadvantage) by employing a unique stopping mechanism.
>
> ---
>
> **(M4) For TabPFN, why not run it on large datasets by subset features and/or samples as necessary (as easily supported by TabPFN and used previously in [eg 6, 7])?**
>
> We would like to thank the reviewer for raising an interesting point. Recent work has shown that TabPFN does not achieve competitive results in the aforementioned scenarios which surpass its limitations [3].
>
> ---
>
> **(M5) How did you handle missingness? Or does this not occur at all in the benchmark datasets?**
>
> We do encounter missingness in several of the OpenML datasets we use. For numerical features, missing values are imputed with the mean of the training split. For categorical features, we insert a special token to indicate missing values. We will update the manuscript with the missing information.
>
> ---
>
> **(M6) What does "Min-Max Class Freq" mean in Table 19, and how can it sometimes be 1.00?**
>
> The min-max class frequency measures the proportion of the number of data samples in the least-represented class (lrc) relative to the number of samples in the most-represented class (mrc). This metric is calculated using the following formula: \(cf = lrc/mrc\).
>
> According to this definition, if the classes in a dataset are perfectly balanced, meaning each class has the same number of instances, the min-max class frequency equals 1.00. Conversely, in a highly imbalanced dataset, where the class with the fewest samples contains 100 instances while the class with the most samples contains 1,000 samples, the min-max class frequency would be: \(100/1000 = 0.1\).
>
> This metric provides a quantitative measure of class imbalance, with values closer to 1.00 indicating well-balanced datasets and smaller values highlighting greater imbalance.
>
> ---
> [1] Goodfellow, I., Warde-Farley, D., Mirza, M., Courville, A., & Bengio, Y. (2013). Maxout networks. In International conference on machine learning.
>
> [2] McElfresh, D., Khandagale, S., Valverde, J., Prasad C, V., Ramakrishnan, G., Goldblum, M., & White, C. (2024). When do neural nets outperform boosted trees on tabular data?. Advances in Neural Information Processing Systems.
>
> [3] Gorishniy, Y., Kotelnikov, A., & Babenko, A. (2025). TabM: Advancing Tabular Deep Learning with Parameter-Efficient Ensembling.  International Conference on Learning Representations.
>
> [4] Gorishniy, Y., Rubachev, I., Khrulkov, V., & Babenko, A. (2021). Revisiting deep learning models for tabular data. Advances in Neural Information Processing Systems.
>
> ---

---

> > ### Comment · Reviewer_tLh9 · 2025-02-02
> > **Re: revision and rebuttal**
> >
> > I have reviewed the revision and rebuttal, and am delighted that all my concerns have been fundamentally addressed, and I will shortly update my review accordingly. Re (M2) raw hyperparameter data, thanks for the clarification; I agree that it would be unsuitable to present this even in the supplemental. However, I think it would be nice to upload these to the Github repo, either now or upon acceptance. It's strangely hard to find such data on the internet, so making these broadly available would be useful.

---

### Review · Reviewer_wftX · 2025-01-19

**Summary Of Contributions:**

This paper provides a comprehensive exploration of classification models for tabular data, particularly designing a fairer evaluation protocol and a more thorough hyperparameter search process. The paper covers various families of algorithms for tabular data and derives several valuable conclusions based on thorough experiments.

**Audience:**

Yes

**Broader Impact Concerns:**

None.

**Claims And Evidence:**

Yes

**Requested Changes:**

Could the paper provide a clearer explanation of the following questions:
- In the hyperparameter search process, how is the search space defined to ensure fairness?
- Some models in the hyperparameter search process are significantly slower than more efficient models like XGBoost. How can a reasonable total time budget for the search process be defined?
- The authors claim to be the first to conduct a "deeper investigation" on foundation models. Could the authors clarify what this "deeper investigation" specifically refers to?

**Strengths And Weaknesses:**

**Strengths.**
- The exploration process in the paper involves extensive hyperparameter tuning, including efficient search strategies and sufficient search rounds.
- The paper features a well-designed protocol, using 10-fold cross-validation for thorough evaluation on each dataset, and retraining models on combined training and validation data.

**Weaknesses.**
- The conclusions drawn from the investigation are not very convincing. This can be attributed to two main reasons:
    - The study lacks many important methods in terms of both quantity and variety. For instance, classical machine learning methods such as SVM and Naive Bayes are missing. Besides, dataset-specific neural networks that show comparable performance to XGBoost and other gradient-boosting models, such as TabR[1], ModernNCA[2], and RealMLP[3], are not explored. The ResNet and SAINT models analyzed in the paper are no longer the most representative state-of-the-art deep learning models. As a result, the experimental findings do not strongly support conclusions like "Do DL models outperform gradient boosting methods in tabular data classification?"
    - Recent tabular data benchmarks typically contain hundreds of datasets, such as Tabzilla[4] (176 datasets). The limited number of datasets used in this paper may lead to biased conclusions. For example, the deviation observed in Figure 12 may not be due to the inherent properties of TabPFN and TP-BERTa as suggested, but rather due to the limited datasets available.
- The paper mainly compares the performance of different methods, lacking deeper, more insightful analysis. For instance, further investigation into why certain models perform better in specific scenarios could provide more value.
- Many of the algorithms mentioned in the paper are also applicable to regression tasks, which are also a significant problem in tabular data. It would be valuable for the paper to explore regression tasks as well.

[1] TabR: Tabular Deep Learning Meets Nearest Neighbors in 2023. [https://arxiv.org/abs/2307.14338]

[2] Modern Neighborhood Components Analysis: A Deep Tabular Baseline Two Decades Later. [https://arxiv.org/abs/2407.03257]

[3] Better by Default: Strong Pre-Tuned MLPs and Boosted Trees on Tabular Data. [https://arxiv.org/abs/2407.04491]

[4] When Do Neural Nets Outperform Boosted Trees on Tabular Data? [https://arxiv.org/abs/2305.02997]

---

> ### Author Response · Authors · 2025-02-01
>
> We thank the reviewer for investing the time in reviewing our work. Below, we address the reviewer’s main concerns:
>
> ---
> ### Requested Changes
> ---
>
> ---
>
> **In the hyperparameter search process, how is the search space defined to ensure fairness?**
>
> We use the hyperparameter search space as specified in the respective papers where the models were introduced if a hyperparameter search space was made available; if not, we use search spaces from prior related work. A comprehensive overview of the hyperparameters and their corresponding ranges, as provided in the original implementations, is included in “Appendix A - Configuration Space”.
>
> ---
>
> **Some models in the hyperparameter search process are significantly slower than more efficient models like XGBoost. How can a reasonable total time budget for the search process be defined?**
>
> Unlike related studies that employ random search for hyperparameter optimization, we utilize Optuna's TPE algorithm. This approach generally reduces the time required to identify an optimal set of hyperparameters in high-dimensional/large search spaces (especially when parallel resources are not available). This is particularly beneficial for deep learning models that require a longer time to train, allowing for better test-time performance in a limited time frame. Additionally, we make use of two different criteria to terminate the hyperparameter optimization process: the number of maximal trials and the maximal amount of time. The intuition behind our two criteria is that early stopping based on the maximal amount of trials should benefit fast methods that can overfit the validation data, while stopping based on time keeps the comparison fair between deep learning methods and traditional methods, given that deep learning methods take more time to train and consequently to achieve a specific number of HPO trials.
>
> ---
>
> **The authors claim to be the first to conduct a "deeper investigation" on foundation models. Could the authors clarify what this "deeper investigation" specifically refers to?**
>
> We believe our work is the first to include foundation models (In-context Learning and Fine-tuning) as baselines, which were mostly overlooked in previous studies such as those by [1], [2], [3] (cf. Table 1). Consequently, our survey introduces a classification framework that organizes models into distinct families. This classification provides a comprehensive understanding of the performance dynamics within and across these families, offering deeper insights into their behavior on various scaled datasets. The scope of our study is to explore the capabilities of these families, highlight their potential in tabular data classification, and rank them in both small-scale and large-scale data regimes based on the benchmark datasets provided within the OpenMLCC18 benchmark. By ranking different model families, we gain an understanding of which learning paradigm is superior to others.
>
> ---

---

> > ### Author Response · Authors · 2025-02-01
> >
> > ---
> > ### Weaknesses (1/2)
> > ---
> > ---
> > **The study lacks many important methods in terms of both quantity and variety. For instance, classical machine learning methods such as SVM and Naive Bayes are missing. Besides, dataset-specific neural networks that show comparable performance to XGBoost and other gradient-boosting models, such as TabR[1], ModernNCA[2], and RealMLP[3], are not explored. The ResNet and SAINT models analyzed in the paper are no longer the most representative state-of-the-art deep learning models. As a result, the experimental findings do not strongly support conclusions like "Do DL models outperform gradient boosting methods in tabular data classification?"**
> >
> > We believe that in our study we have included top-performing representatives from all model families. As such, we believe that SVM and Naive Bayes are methods that do not impact the validity of our results, given that both methods are outperformed by gradient boosting methods as also shown in prior work [1]. Secondly, we would like to point out to the reviewer that TabR, ModernNCA, and RealMLP were developed concurrently with our research efforts.
> > Given the limited time, in our revised draft, we included RealMLP [4] as another dataset-specific baseline as suggested by the reviewer. Notably, we use the models as proposed in the original papers. However, we want to highlight that a subset of components applied in the models could potentially boost the performance of competitors. An analysis of the impact of the components on a baseline (e.g., numerical embeddings as proposed in RealMLP applied on any other baseline) is a worthwhile investigation on its own, and we leave this study for future work as it is out of scope of our empirical assessment.
> >
> > ---
> >
> > **Recent tabular data benchmarks typically contain hundreds of datasets, such as Tabzilla[4] (176 datasets). The limited number of datasets used in this paper may lead to biased conclusions. For example, the deviation observed in Figure 12 may not be due to the inherent properties of TabPFN and TP-BERTa as suggested, but rather due to the limited datasets available.**
> >
> > We would like to direct the reviewer's attention to Table 1, where we have listed all the recent work regarding tabular data benchmarks according to our knowledge. From 5 out of 7 listed related works, we have the highest number of included datasets. Furthermore, we believe we have clearly listed the differences between Tabzilla, the other baselines, and our work.
> >
> > We would kindly invite the reviewer to page 13, where we have written “This suggests that the initial divergence was likely due to the limited number of datasets rather than the inherent properties of these methods.” In Figure 13, we have provided an analysis where we have included only the intersection of datasets where all methods can be run, and we have observed that all methods are impacted similarly by the meta-features.
> >
> > With limited datasets, we do not imply a limited number of datasets considered in our study, since, as shown in Table 1, we believe our work is well-positioned in the community compared to prior work. What we imply is that both TabPFN and TP-BERTa have limitations regarding the datasets they can be applied to.
> >
> > ---

---

> ### Author Response · Authors · 2025-02-01
>
> ---
> ### Weaknesses (2/2)
> ---
> ---
> **The paper mainly compares the performance of different methods, lacking deeper, more insightful analysis. For instance, further investigation into why certain models perform better in specific scenarios could provide more value.**
>
> We believe that we have performed an investigation regarding a possible correlation between various considered meta-features and the predictive performance of different models. However, our analysis implies that all the meta-features impact the different models similarly. We believe that we have shared valuable insights in our work, for example, the impact of the different hyperparameters on the predictive performance of specific models, the impact of model-based HPO on the predictive performance of individual models, a detailed comparison of different foundation models, AutoML methods, etc. Furthermore, in our revised draft, we added in Appendix E an analysis of the relationship between dataset size and the performance of different methods.
>
> ---
>
> **Many of the algorithms mentioned in the paper are also applicable to regression tasks, which are also a significant problem in tabular data. It would be valuable for the paper to explore regression tasks as well.**
>
> We agree with the reviewer that regression tasks are equally important to classification tasks in the benchmark. However, at the time the experiments were conducted, there was no established regression benchmark in the community compared to classification, which has been available for a few years and cleaned multiple times over the passing years. As such, we mostly focused our study on classification tasks. Based on this, we believe the aforementioned point, while interesting, should be tackled in future work from our perspective.
>
> ---
>
> [1] McElfresh, D., Khandagale, S., Valverde, J., Prasad C, V., Ramakrishnan, G., Goldblum, M., & White, C. (2024). When do neural nets outperform boosted trees on tabular data?. Advances in Neural Information Processing Systems, 36.
>
> [2] Gorishniy, Y., Rubachev, I., Khrulkov, V., & Babenko, A. (2021). Revisiting deep learning models for tabular data. Advances in Neural Information Processing Systems, 34, 18932-18943.
>
> [3] Borisov, V., Leemann, T., Seßler, K., Haug, J., Pawelczyk, M., & Kasneci, G. (2022). Deep neural networks and tabular data: A survey. IEEE transactions on neural networks and learning systems.
>
> [4] Holzmüller, D., Grinsztajn, L., & Steinwart, I. (2024). Better by default: Strong pre-tuned mlps and boosted trees on tabular data. The Thirty-eighth Annual Conference on Neural Information Processing Systems.
>
> ---

---

### Decision · Action_Editor_2xqy · 2025-02-27

**Recommendation:** Reject

**Comment:**

# Summary of Paper

The paper presents a comprehensive benchmarking study on tabular data classification, comparing various methods, including gradient boosting decision trees, dataset-specific deep learning methods, AutoML, and tabular foundation models. The study aims to provide a fair evaluation protocol and a thorough hyperparameter search process. The authors evaluate 13 methods on the OpenML-CC18 benchmark. The paper also introduces a classification framework to organize models into distinct families and provides insights into their performance across different dataset scales.

## Advantages

- **Thorough Hyperparameter Tuning and Evaluation**: The paper employs an extensive hyperparameter search process, including the use of Optuna's TPE algorithm. Additionally, the evaluation employs a nested cross-validation approach, partitioning the data into 10 folds. Nine of these folds are used for hyperparameter tuning, with each hyperparameter evaluated using 9-fold cross-validation. This rigorous approach ensures a robust evaluation of model performance.

- **Inclusion of Various Methods**: The study includes AutoML and recent fine-tuning-based methods like CARTE, TP-BERTa, and XTab, which were previously overlooked in similar benchmarks. This inclusion broadens the scope of the study.

# Summary of Discussion

The reviewers generally appreciated the thoroughness of the benchmarking study. However, several concerns were raised regarding the methodology, scope, and depth of the investigation. Key points of discussion include:

## Limited Dataset Coverage
- While the study covers 68 datasets, ranking third among related works, this coverage is still considered limited compared to other studies. Additionally, the largest dataset in the study contains fewer than 100,000 instances, which raises concerns about the generalizability of the findings to larger-scale datasets.

## Regression Omission
- Both Reviewer tLh9 and Reviewer wftX noted the omission of regression tasks, which are important in tabular data analysis. Regression tasks have been explored in multiple related works, including [1-4], and their exclusion limits the scope and generalizability of the study's findings.

## Insufficient Method Coverage
- Reviewer tLh9 suggested that recent hypernetwork-based methods and LLM feature engineering methods should be included in the study to provide a more diverse and up-to-date comparison. Reviewer wftX criticized the study for lacking recent state-of-the-art models like TabR, which was publicly released in July 2023, prior to the submission of this paper. The absence of such models raises concerns about the validity of the conclusions, particularly regarding the comparative performance of DL models versus gradient boosting methods. Additionally, the limited number of methods (13) and their division into 6 families were seen as insufficient to draw robust conclusions about model performance.

## Refitting
- Reviewer tLh9 questioned the necessity of refitting models on combined training and validation data. While the authors provided empirical evidence in the revision to support their refitting approach, the ablation study only examined two methods: FT-Transformer and CatBoost. The results showed that refitting significantly improved CatBoost's performance but did not consistently benefit FT-Transformer. Given that refitting increases the amount of training data and the benchmark includes many small datasets, the claim that "refitting outperforms its non-refitted counterpart" is not sufficient to support the importance of refitting.

## Time Budget Definition
- The justification for the time budget (23 hours and up to 100 hyperparameter evaluations) was questioned by both Reviewer tLh9 and Reviewer wftX. The authors did not fully address the concern of how a reasonable total time budget for the search process should be defined. Reviewers noted that the time budget should be explicitly justified, especially given the variability in dataset sizes and computational resources.

## Limited Insights
- The findings presented in the paper, such as "GBDTs show robust performance" and "TabPFN performs well on small datasets," are consistent with conclusions drawn in recent works. For example, the strong performance of TabPFN on small datasets was also observed in [3], which evaluated 57 small datasets.

# Conclusion

While the paper presents a valuable contribution to the field of tabular data classification, it falls short in several critical areas. Given the above concerns, the paper, in its current form, does not meet the standards required for acceptance. We therefore recommend that the authors undertake a major revision and resubmit the manuscript.

# Improvements
To strengthen the study, the following improvements are also recommended:

1. **Expand the Benchmark Scope**:
   - Incorporate **regression tasks** into the benchmark.
   - Include **state-of-the-art tabular data methods**, such as TabR, which have demonstrated strong performance in related works.
   - Extend the evaluation to **large-scale datasets**.

2. **Clarify Key Methodological Choices**:
   - Provide a **detailed justification** for the **23-hour time budget** for hyperparameter search. This should include an analysis of how this budget was determined in relation to the computational resources required for different methods, the variability in dataset sizes.
   - Clearly explain why **refitting** is considered a **fair and reasonable protocol** for model evaluation. Address potential biases, such as the observed performance boost in GBDT methods (e.g., CatBoost) compared to others (e.g., FT-Transformer), and provide empirical evidence to support the claim that refitting is universally beneficial or at least unbiased across different methods.


## References

[1] Revisiting deep learning models for tabular data. In NeurIPS 2021.

[2] Why do tree-based models still outperform deep learning on typical tabular data? In NeurIPS 2022.

[3] When do neural nets outperform boosted trees on tabular data?. In NeurIPS 2024.

[4] A closer look at deep learning methods on tabular datasets. ArXiv:2407.00956. 2025.

**Audience:**

Yes.

**Claims And Evidence:**

The claims made in the submission are not fully supported by accurate, convincing, and clear evidence, as highlighted by the following issues:

1. **From the perspective of convincing experimental evidence**:
   - The study lacks regression tasks.
   - The dataset coverage is still considered limited compared to other works, and the largest dataset contains fewer than 100,000 instances.
   - The method coverage is insufficient.

2. **From the perspective of the explanation of specific settings**:
   - The importance of refitting models on combined training and validation data is not convincingly supported.
   - The justification for the time budget of HPO is unclear and not fully addressed.

Details of each part and the recommended improvements can be found in the following comments.

**Resubmission Of Major Revision:**

The authors may consider submitting a major revision at a later time.